# Dirichlet-based Per-Sample Weighting by Transition Matrix for Noisy Label Learning

**HeeSun Bae[1], Seungjae Shin[1], Byeonghu Na[1] & Il-Chul Moon[1,2]**
[1]Department of Industrial and Systems Engineering, KAIST, [2]summary.ai
{cat2507,tmdwo0910,wp03052,icmoon}@kaist.ac.kr

## Abstract

For learning with noisy labels, the transition matrix, which explicitly models the relation between noisy label distribution and clean label distribution, has been utilized to achieve the statistical consistency of either the classifier or the risk. Previous researches have focused more on how to estimate this transition matrix well, rather than how to utilize it. We propose good utilization of the transition matrix is crucial and suggest a new utilization method based on resampling, coined RENT. Specifically, we first demonstrate current utilizations can have potential limitations for implementation. As an extension to Reweighting, we suggest the Dirichlet distribution-based per-sample Weight Sampling (DWS) framework, and compare reweighting and resampling under DWS framework. With the analyses from DWS, we propose RENT, a REsampling method with Noise Transition matrix. Empirically, RENT consistently outperforms existing transition matrix utilization methods, which includes reweighting, on various benchmark datasets. Our code is available at https://github.com/BaeHeeSun/RENT.

## 1 Introduction

The success of deep neural networks heavily depends on a large-sized dataset with accurate annotations (Daniely & Granot, 2019; Berthon et al., 2021). However, creating such a large dataset is arduous and inevitably affected by human errors in annotations, referred to as *noisy labels*. It causes model performance degradation (Arpit et al., 2017; Zhang et al., 2021a;b), and studies have been suggested to solve this degradation (Zhang & Sabuncu, 2018; Li et al., 2020; Wang et al., 2021; Wei et al., 2021b; Bae et al., 2022; Na et al., 2024). Among various treatments, one prominent approach is to estimate the transition matrix from true labels to noisy labels (Patrini et al., 2017).

Transition matrix explicitly models the relation between noisy labels and the latent clean labels (Yao et al., 2020; Li et al., 2021). It means that the transition matrix provides a probability that a given true label is transitioned to another noisy label, where the true label is unknown in our setting. With this transition matrix, a trainer can ensure statistical consistency either to the true classifier (Patrini et al., 2017) or to the true risk (Liu & Tao, 2015) ideally. Since this information is unknown when learning with noisy label, previous transition matrix related studies have focused on the accurate estimation of transition matrix (Li et al., 2021; Cheng et al., 2022).

Even if we assume that the transition matrix is accurately estimated, how we utilize the transition matrix can also impact the performance. Forward (Patrini et al., 2017) is one of the general risk structures for utilizing the transition matrix (Zhu et al., 2021; Yang et al., 2022). It trains a classifier by minimizing the divergence between the noisy label distribution and the classifier output weighted by the transition matrix. Other ways of transition matrix utilization include Reweighting (Liu & Tao, 2015; Xia et al., 2019). Reweighting employs an importance-sampling technique, ensuring statistical consistency of the empirical risk to the true risk. However, in practice, the empirical risk of Reweighting can also deviate from the true risk because the estimation of per-sample weights relies on the imperfect classifier's output. In other words, when the classifier's output cannot accurately estimate per-sample weights, it can lead to a potential mismatch between the estimated empirical risk and the true risk, compromising the effectiveness of reweighting as a transition matrix utilization.

Recently, An et al. (2020) suggested that resampling outperforms reweighting for correcting dataset sampling bias. Motivated by the potential benefit of resampling over reweighting, we introduce an

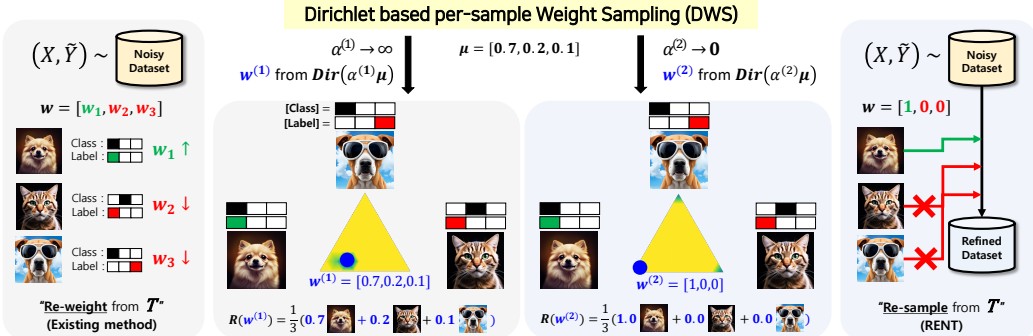

Figure 1: Dirichlet distribution-based per-sample Weight Sampling with shape parameter $\alpha$ and the mean vector $\boldsymbol{\mu}$. Image at the vertices of yellow triangles represents data instance. Blocks above the images represent **true Class**, **noisy Label**. Sides are implementation example of sampled $\boldsymbol{w}$. $\boldsymbol{w}^{(1)}$ assigns weights to all data (Reweighting), while $\boldsymbol{w}^{(2)}$ simulates resampling refined dataset (RENT).

extended framework, Dirichlet distribution-based per-sample Weights Sampling (DWS), to encompass both resampling and reweighting. Figure 1 shows an implementation example of our framework. $\boldsymbol{w}^{(1)}$ and $\boldsymbol{w}^{(2)}$ are sampled weight vectors from different Dirichlet distributions with the same mean. They represent different properties by the shape parameter, $\alpha$. These weights are represented as reweighting and resampling in implementation, as each side of the figure.

We then analyze the impact of $\alpha$ on DWS. First, $\alpha$ affects the variance of risk and smaller $\alpha$ means larger variance. When variance of the risk increases, it showed performance improvement according to Lin et al. (2022) empirically, so we expect better performance with small $\alpha$. Second, the Mahalanobis distance between the true weight vector and the mean of per-sample weights sampling distribution is proportional to the square root of $\alpha$, emphasizing the merit of small $\alpha$. Finally, $\alpha$ is related to label perturbation. This label perturbation aligns with Chen et al. (2020), who proposes label perturbation during training can reproduce better performance for learning with noisy label.

This analysis on the impact of $\alpha$ under DWS framework finally explains the differences between resampling and reweighting. It provides theoretical rationale behind the superior performance of resampling over reweighting for learning with noisy label. It leads us to introduce RENT, the first REsampling method to utilize the Noise Transition matrix. RENT empirically shows better performance for $T$ utilization when combined with various $T$ estimation methods.

In summary, the contributions of this paper are as follows.

1) We suggest DWS, which samples per-sample weight vectors from the Dirichlet distribution.

2) With DWS, we can express reweighting and resampling in a unified framework and demonstrate resampling can be better than reweighting for learning with noisy label.

3) Under this situation, we suggest RENT, which resamples dataset with the transition matrix. RENT empirically shows good performance, suggesting a new transition matrix utilization method.

## 2 TRANSITION MATRIX FOR LEARNING WITH NOISY LABEL

### 2.1 PROBLEM DEFINITION: LEARNING WITH NOISY LABEL

This paper considers a classification task with $C$ classes. Let the uppercase, e.g. $(X, Y)$, be a random variable and the corresponding lowercase, e.g. $(x, y)$, denote a realized instance. We define an input as $X \in \mathcal{X}$ and a true label as $Y \in \mathcal{Y}$, where $\mathcal{X} \subset \mathbb{R}^d$ and $\mathcal{Y} = \{1, ..., C\}$, respectively. $\tilde{Y} \in \mathcal{Y}$ represents a noisy label. $\tilde{\mathcal{D}} = \{(x_i, \tilde{y}_i)\}_{i=1}^N$ is the dataset with noisy labels. The model is $f_\theta(x) = \sigma(g(x; \theta))$, parameterized by $\theta$. $g : \mathbb{R}^d \to \mathbb{R}^C$ is a mapping function and $\sigma(\cdot)$ is a softmax function. The objective is to find the optimal classifier $f_\theta^*$ which minimizes the true risk as:

$$f_\theta^* = \operatorname{argmin}_{f_\theta} R_l(f_\theta), \text{ with } R_l(f_\theta) := \mathbb{E}_{(x,y) \sim p(X,Y)} [l(f_\theta(x), y)] \tag{1}$$

$l(\cdot, \cdot)$ is a loss function that measures the prediction quality of the label. For simplicity, we denote $R_l(f_\theta)$ as $R_l$, omitting $f_\theta$ unless otherwise specified. Since $\tilde{\mathcal{D}}$ contains noisy labels, the empirical risk function using $\tilde{\mathcal{D}}$, denoted as $\tilde{R}_l^{emp} := \frac{1}{N} \sum_{i=1}^N l(f_\theta(x_i), \tilde{y}_i)$, does not converge to the true

risk $R_l$. It implies that $f_\theta^*$ cannot be accurately approximated by minimizing $\tilde{R}_l^{emp}$. Therefore, our objective is to minimize $R_l$, or training $f_\theta$ to approximate $p(Y|X = x)$, by learning with $\tilde{D}$. Please refer to Appendix B for more studies related to learning with noisy label task.

## 2.2 Transition Matrix for Learning with Noisy Label

When there are $C$ classes, the noisy label generation process can be explained with a matrix $T \in [0, 1]^{C \times C}$, whose entry $T_{jk}$ is the probability of a clean label $k$ being flipped to a noisy label $j$. In other words, a noisy label $\tilde{y}_i$ is the sampling result from clean label $y_i$ with its flip probability as $p(\tilde{Y} = \tilde{y}_i | Y = y_i)$. Here, this matrix, $T$, has been referred to as the *transition matrix* in noisy label community (Patrini et al., 2017; Yao et al., 2021).[1] Then, the noisy label distribution, from which the noisy labelled dataset are sampled, can be expressed as Eq. 2.

$$\boldsymbol{p}(\tilde{Y}|x) = T(x)\boldsymbol{p}(Y|x) \text{ with } T_{jk}(x) = p(\tilde{Y} = j|Y = k, x) \ \forall j, k = 1, ..., C \tag{2}$$

Here, $\boldsymbol{p}(\cdot|\cdot)$ is vector and $p(\cdot|\cdot)$ is scalar. Either $\boldsymbol{p}(Y|x)$ or $\boldsymbol{p}(\tilde{Y}|x)$ can be calculated from the other if $T$ is given.[2] With this property, previous studies utilizing the transition matrix have been able to explain the statistical consistency of their classifier (Li et al., 2021; Cheng et al., 2022) or risk (Liu & Tao, 2015; Patrini et al., 2017; Liu et al., 2023) to their true counterpart. The problem is that true $T$ is unknown, and previous studies have focused on estimating the good transition matrix (Patrini et al., 2017; Xia et al., 2020; Zhang et al., 2021b; Zhu et al., 2021; 2022).

While we acknowledge the importance of estimating $T$, this paper emphasizes the utilization phase of $T$ in its research scope. Modifying $\tilde{R}_l^{emp}$ is essential for training $f_\theta$ to minimize Eq. 1 when only $T$ and $\tilde{D}$ are available. We explicitly refer to this as $T$ utilization, which means, we divide the transition matrix based $f_\theta$ training into two phases: 1) $T$ estimation and 2) $T$ utilization. Empirically, $T$ utilization impacts the model performance significantly, underscoring the importance of utilization phase. The following section analyzes the previous researches on $T$ utilization, and we demonstrate that current practices on $T$ utilization inherits significant drawbacks in actual deployments.

## 2.3 Utilizing Transition Matrix for Learning with Noisy Label

In this section, assume that true $T$ is accessible for the sake of analyzing $T$ utilization. Until now, three directions have been proposed for $T$ utilization (Patrini et al., 2017; Liu & Tao, 2015). Each methodology claims that it guarantees a specific type of statistical consistency, either to the classifier or the risk. However, there are cases when this ideal consistency is empirically hard to be achieved, and this section analyzes such practical situations without ideal consistency. We consider $l$ as Cross Entropy loss, which is generally used for classification. $R_{l,\cdot}^{emp}$ denotes the empirical risk of $R_{l,\cdot}$.

1) **Forward** (Patrini et al., 2017) risk minimizes the gap between $\tilde{Y}$ and $Tf_\theta(X)$ as $R_{l,\mathrm{F}} = \mathbb{E}_{p(X,\tilde{Y})}\left[l\left(Tf_\theta(X), \tilde{Y}\right)\right]$. The learned $f_\theta$ is statistically consistent to the optimal classifier $f_\theta^*$. However, $f_\theta$ trained with $R_{l,\mathrm{F}}^{emp}$ can be different from $f_\theta^*$. According to Zhang et al. (2021b), the gap between $\boldsymbol{p}(\tilde{Y}|x)$ and the noisy label probability distribution approximated from $\tilde{D}$ can be high. We specify the impact of this gap, $\epsilon(\neq 0)$, to the classifier as $f_\theta^*(x) - f_\theta(x) = T^{-1}\epsilon \neq 0$. It means that if $\boldsymbol{p}(\tilde{Y}|x)$ is not estimated accurately, the deviation of $f_\theta$ from $f_\theta^*$ is inevitable for classifier consistency. Please check Appendix C also for more discussions regarding this issue.

2) **Backward** (Patrini et al., 2017) risk is $R_{l,\mathrm{B}} = \mathbb{E}_{p(X,\tilde{Y})}\left[\boldsymbol{l}(X)T^{-1}\right]$ with $\boldsymbol{l}(X) = [l(f_\theta(X), 1), ..., l(f_\theta(X), C)]$. As $\boldsymbol{p}(Y|X) = T^{-1}\boldsymbol{p}(\tilde{Y}|X)$, $R_{l,\mathrm{B}}$ is statistically consistent to $R_l$. However, optimizing $R_{l,\mathrm{B}}^{emp}$ can lead to unstable performances as reported in Patrini et al. (2017).

3) **Reweighting** (Liu & Tao, 2015; Xia et al., 2019) risk computes per-sample weights based on the likelihood ratio (Kahn & Marshall, 1953), for the noisy label classification as: $R_{l,\mathrm{RW}} = \mathbb{E}_{p(X,\tilde{Y})}\left[\frac{p(Y=\tilde{Y}|X)}{(T\boldsymbol{p}(Y|X))_{\tilde{Y}}}l\left(f_\theta(X), \tilde{Y}\right)\right]$. Here, $(\cdot)_c$ means the $c$−th cell value of the vector. $R_{l,\mathrm{RW}}^{emp}$ would be expressed as $\sum_{i=1}^N \frac{1}{N}\frac{f_\theta(x_i)_{\tilde{y}_i}}{(Tf_\theta(x_i))_{\tilde{y}_i}}l(f_\theta(x_i), \tilde{y}_i)$.

---

[1]We define the transition matrix as Eq. 2 for mathematical correctness. Appendix B.2 for more explanations.

[2]Since it is natural assuming the dependency of $T$ on the input, we consider the transition matrix of $x$ as $T(x)$. In this paper, we omit $x$ from $T(x)$, denoting as $T$ for convenience.

By applying importance sampling (Kahn & Marshall, 1953; Katharopoulos & Fleuret, 2018) to $T$ utilization, **Reweighting** does not suffer from unstable optimization as **Backward**. However, the problem is that $p(Y|X)$ is required as a component for per-sample weight, where estimating $p(Y|X)$ serves as the final objective. While previous studies (Liu & Tao, 2015; Berthon et al., 2021) have estimated $p(Y|X)$ from the on-training classifier's output, this estimation can be inaccurate.

# 3 DWS: DIRICHLET-BASED PER-SAMPLE WEIGHT SAMPLING

In this section, we suggest a new framework, DWS, which incorporates per-sample Weight Sampling based on the Dirichlet distribution. Through this incorporation, we interpret sample reweighting and resampling by a single framework, facilitating the direct comparison of both methods. Then, we analyze the impact of the shape parameter in the Dirichlet distribution, from which the per-sample weights are generated, to empirical risk. With these analyses, we propose resampling can be a better choice than reweighting based on two characteristics: the closeness between the mean of the per-sample weight distribution and the true weight; and the label perturbation nature of resampling when it is explained by DWS. Finally, we introduce our method, RENT, which becomes a resampling-based approach for learning with noisy label.

## 3.1 DIRICHLET-BASED WEIGHT SAMPLING

Let $\mathrm{Dir}(\alpha\boldsymbol{\mu})$ be the Dirichlet distribution with parameters $\alpha \in \mathbb{R}$ and $\boldsymbol{\mu} \in \mathbb{R}^N$. $\alpha$ is a scalar concentration parameter and $\boldsymbol{\mu}$ is a base measure with $\sum_{i=1}^{N} \mu_i$ is equal to 1. $\alpha, \mu_1, ..., \mu_N$ are positive values by the definition. Figure 2 illustrates properties of the Dirichlet distribution. As depicted in the figure, instances drawn from the distribution with small $\alpha$ tend to cluster around the vertices of a simplex (the leftmost). The sampling frequencies for each dimension converge to the mean value of that dimension.

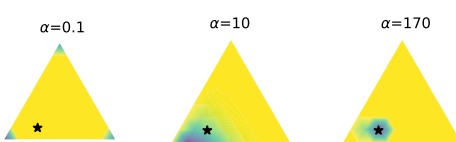

Figure 2: Density plot of $\mathrm{Dir}(\alpha\boldsymbol{\mu})$ with different $\alpha$. $\boldsymbol{\mu}$ is set as $[\mathbf{0.7}, \mathbf{0.2}, \mathbf{0.1}]$ for this illustration. Star ($\star$) denotes the mean ($\boldsymbol{\mu}$). Note that this value is invariant to $\alpha$. Yellow denotes lower density, while it becomes denser progressively with violet.

In contrast, as $\alpha$ increases, a greater proportion of samples from the Dirichlet distribution will exhibit vectors that are in proximity to the mean vector (the rightmost).

Now, consider the property of reweighting and resampling. Reweighting multiplies pre-defined per-sample weight values to the loss of each sample, indicating the importance of each sample. On the other hand, resampling alters the composition of the dataset by assigning an importance value to each sample via its sampling ratio. By interpreting *not selecting* as *assigning 0 weight value to* a specific sample, resampling can be perceived as a process of per-sample weight vector sampling that exhibits concentration toward a specific dimension. Consequently, we suggest that both reweighting and resampling can be interpreted in the Dirichlet-based per-sample weight sampling (DWS) framework.

$$R_{l,\mathrm{DWS}}^{emp} := \frac{1}{M} \sum_{j=1}^{M} \sum_{i=1}^{N} w_i^j l(f_\theta(x_i), \tilde{y}_i), \text{ with } \boldsymbol{w^j} \sim \mathrm{Dir}(\alpha\boldsymbol{\mu}) \tag{3}$$

Here, $\boldsymbol{w^j} \in \mathbb{R}^N$ is $j$-th sample following $\mathrm{Dir}(\alpha\boldsymbol{\mu})$ and $\boldsymbol{w^j} = [w_1^j, ..., w_N^j]$. $M$ is the number of sampling $\boldsymbol{w}$. With Eq. 3, per-sample weight parameters of existing **Reweighting** can be considered as the sampled weights from the Dirichlet distribution whose $\alpha \to +\infty$. Also, resampling can be interpreted as the reweighting method with sampled weights from the distribution with its $\alpha \to 0$. In other words, we integrate both reweighting and resampling as Eq. 3, providing a way to compare these two distinct importance sampling based techniques comprehensively.

## 3.2 ANALYZING DWS FOR LEARNING WITH NOISY LABEL

Following Section 3.1, $\alpha$ controls the property of $\boldsymbol{w}$. Therefore, analyzing the impact of $\alpha$ to $R_{l,\mathrm{DWS}}^{emp}$ is necessary for comparing several DWS cases with different $\alpha$ (i.e. reweighting, resampling).

$V(w_i^j)$ **and** $V(R_{l,\mathbf{DWS}}^{emp})$   We start from the variance of $w_i^j$ as $V(w_i^j) = \frac{\mu_i(1-\mu_i)}{\sum_{l=1}^{N} \alpha\mu_l + 1} = \frac{\mu_i(1-\mu_i)}{\alpha+1}$ for all $i = 1, ..., N$ by definition. Here, $V(w_i^j)$ converges to 0 as $\alpha \to +\infty$ (reweighting) and becomes

larger with smaller $\alpha$ (if $\alpha \to 0$, it represents resampling). According to Lin et al. (2022), increasing the variance of the risk function can improve robustness when learning with noisy label empirically. This study supports the robustness improvement of our framework, DWS with smaller $\alpha$ (or with larger $V(w_i^j)$) as Eq. 4. Let $l_i = l(f_\theta(x_i), \tilde{y}_i), \ \forall i = 1, ..., N$ for convenience.

$$V(R_{l,\text{DWS}}^{emp}) = \frac{1}{M^2} \sum_{j=1}^{M} \left( \sum_{i=1}^{N} l_i^2 V(w_i^j) + \sum_{k \neq i} l_i l_k Cov(w_i^j, w_k^j) \right) \tag{4}$$

Note that our study is different from Lin et al. (2022) in that they focused on the impact of increasing the variance of the risk function as a regularization to overfitting to noisy labels, while our objective is to suggest a unified framework to explain reweighting and resampling.

**Distance from the true weight**   Mean of per-sample weights vectors, $\boldsymbol{\mu}$, is approximated from the on-training classifier. Therefore, it may be different from the true per-sample weight vector. Let $\tilde{\mu}_i^* = \frac{p(Y = \tilde{y}_i | x_i)}{p(\tilde{Y} = \tilde{y}_i | x_i)}$ and $\boldsymbol{\mu}^* = \left[ \frac{\tilde{\mu}_i^*}{\sum_{l=1}^{N} \tilde{\mu}_l^*}, ..., \frac{\tilde{\mu}_N^*}{\sum_{l=1}^{N} \tilde{\mu}_l^*} \right]$. Note that the mean of per-sample weights distribution can be approximated as the Gaussian distribution following central limit theorem (Douglas C. Montgomery, 2013), since $\boldsymbol{w^j}$ are i.i.d. sampled. It means $\bar{\boldsymbol{w}} = \frac{1}{M} \sum_{j=1}^{M} \boldsymbol{w^j}$ will follow $\mathcal{N}(\boldsymbol{\mu}, \Sigma/M)$. $\Sigma$ is the covariance matrix of $\text{Dir}(\alpha\boldsymbol{\mu})$. We decompose $\Sigma = S/(\alpha + 1)$ as $\alpha$ and $\alpha$-invariant term $S$. Then we get Mahalanobis distance (Mahalanobis, 1936) between $\boldsymbol{\mu}^*$ and $\bar{\boldsymbol{w}}$ as:

$$d_M(\boldsymbol{\mu}^*, \bar{\boldsymbol{w}}) = \sqrt{(\boldsymbol{\mu}^* - \boldsymbol{\mu})^T \left( \frac{\Sigma}{M} \right)^{-1} (\boldsymbol{\mu}^* - \boldsymbol{\mu})} = \sqrt{M(\alpha + 1)(\boldsymbol{\mu}^* - \boldsymbol{\mu})^T S^{-1} (\boldsymbol{\mu}^* - \boldsymbol{\mu})} \tag{5}$$

The distance between $\boldsymbol{\mu}^*$ and per-sample weight distribution is proportional to the square root of $\alpha$. It means if $\boldsymbol{\mu^*}$ (true) and $\boldsymbol{\mu}$ (estimated) is not the same, $d_M(\boldsymbol{\mu}^*, \bar{\boldsymbol{w}})$ becomes smaller as $\alpha \to 0$.

**Noise injection of $R_{l,\textbf{DWS}}^{emp}$**   Recent researches including Neelakantan et al. (2015) suggest that injecting random noise to the training procedure can improve generalization, and it also works for learning with noisy label (Chen et al., 2020; Wei et al., 2021a). Following the concept, we interpret per-sample weights sampling as normally distributed noise injection during training, as Eq. 6. The intensity of noise can be controlled with $\alpha$ (Full derivation of Eq. 6 is in Appendix D.1).

$$\lim_{N \to \infty} R_{l,\text{DWS}}^{emp} = \sum_{i=1}^{N} \mu_i l(f_\theta(x_i), \tilde{y}_i) + \sum_{i=1}^{N} z_i l(f_\theta(x_i), \tilde{y}_i), \text{ with } z_i \sim \mathcal{N}\left( 0, \frac{\mu_i(1 - \mu_i)}{M(\alpha + 1)} \right) \tag{6}$$

**Comparison to Previous Work**   Eq. 6 denotes $R_{l,\text{DWS}}^{emp}$ with $N \to \infty$ can be similar to the risk of SNL (Chen et al., 2020), who suggested label perturbation enhances the robustness for noisy label learning. We compare details of DWS and SNL. Specifically, the empirical risk function of SNL is:

$$R_{l,\text{SNL}}^{emp} = \sum_{i=1}^{N} l(f_\theta(x_i)\tilde{y}_i) + \sigma \sum_{i=1}^{N} \sum_{k=1}^{C} z_{ik} l(f_\theta(x_i), k), \ z_{ik} \sim \mathcal{N}(0, 1) \tag{7}$$

In the above, $\sigma$ is a hyperparameter and $z_{ik}$ is a perturbation noise from standard Normal distribution. Eq. 6 and Eq. 7 are similar in three points. First, risks are decomposed into two parts: the static risk (the former) and the stochastic noise (the latter). Second, the mean of noise ($\mathbb{E}[z]$) are 0; and third, considering the noise parts (the latter part), they are composed as the multiplication of $l(\cdot, \cdot)$ and $z$.

On the other hand, $R_{l,\text{DWS}}^{emp}$ and $R_{l,\text{SNL}}^{emp}$ differ as follows. First, $R_{l,\text{Dir}}^{emp}$ reflects distribution shift between noisy and true label through $\mu_i$, while $R_{l,\text{SNL}}^{emp}$ does not. This means that $\mathbb{E}[R_{l,\text{SNL}}^{emp}]$ can differ from $R_l$. Second, the distributions of $z_i$ for $R_{l,\text{DWS}}^{emp}$ are not identical, unlike $R_{l,\text{SNL}}^{emp}$. Since the variance term in Eq. 6 increases with an increase in $\mu_i (\leq 0.5)$, it introduces instance-wise adaptive perturbations. This results in larger perturbations applied to confident samples, mitigating overfitting, while smaller perturbations to less confident samples, thereby preserving the training process. To empirically assess the difference between $R_{l,\text{DWS}}^{emp}$ and $R_{l,\text{SNL}}^{emp}$, we compare the performance of the two risks and the noise injection to **Reweighting** in Section 4.4 and Appendix F.4.

Analyzing the impact of $\alpha$ to $\mathcal{R}_{l,\text{DWS}}^{emp}$, smaller $\alpha$ can be beneficial. This also aligns with the experimental results in Section 4.3 and Appendix F.3, showing good performance with $\alpha \to 0$ empirically. Based on these analyses, we suggest a resampling to utilize transition matrix in the following section.

## 3.3 RENT: RESAMPLE FROM NOISE TRANSITION

This section proposes REsampling method to utilize Noise Transition matrix in this section. To our knowledge, this is the first resampling study on noisy label classifications. Inspired by the concept of Sampling-Importance-Resampling (SIR) (Rubin, 1988; Smith & Gelfand, 1992), RENT involves resampling each data instance from the noisy-labelled dataset based on the calculated importance. Algorithm 1 outlines the process of RENT.[3]

---

**Algorithm 1:** REsampling utilizing the Noise Transition matrix (RENT)

---

**Input:** Dataset $\tilde{D} = \{x_i, \tilde{y}_i\}_{i=1}^N$, classifier $f_\theta$, Transition matrix $T$, Resampling budget $M$

**Output:** Updated $f_\theta$

> **while** $f_\theta$ *not converge* **do**
>> Get $\tilde{\mu}_i = f_\theta(x_i)_{\tilde{y}_i} / (T f_\theta(x_i))_{\tilde{y}_i}$ for all $i$
>> Construct Categorical distribution $\pi_N = \text{Cat}(\frac{\tilde{\mu}_1}{\sum_{l=1}^N \tilde{\mu}_l}, \dots \frac{\tilde{\mu}_N}{\sum_{l=1}^N \tilde{\mu}_l})$
>> Independently sample $(x_1, \tilde{y}_1), ..., (x_M, \tilde{y}_M)$ from $\pi_N$
>> Update $f_\theta$ by $\theta \leftarrow \theta - \nabla_\theta \frac{1}{M} \sum_{j=1}^M l(f_\theta(x_j), \tilde{y}_j)$
>
> **end**

---

With the number of sampling as a hyperparameter, we fix $M = N$ for experiments unless specified otherwise. We provide the ablation study on $M$ in Section 4.6. Also, we conducted resampling based on mini-batch for implementation. We provide ablation for this sampling strategy in Appendix F.6. The empirical risk function of the resampled dataset is expressed as Eq. 8.

$$R_{l,\text{RENT}}^{emp} := \frac{1}{M} \sum_{i=1}^N n_i l(f_\theta(x_i), \tilde{y}_i), \text{ where } [n_1, ..., n_N] \sim \text{Multi}(M; \frac{\tilde{\mu}_1}{\sum_{l=1}^N \tilde{\mu}_l}, ..., \frac{\tilde{\mu}_N}{\sum_{l=1}^N \tilde{\mu}_l}) \quad (8)$$

$\tilde{\mu}_i = f_\theta(x_i)_{\tilde{y}_i} / (T f_\theta(x_i))_{\tilde{y}_i}$. $\sum_{j=1}^M w_i^j$ of Eq. 3 with $\alpha \to 0$ can be interpreted as $n_i$ in Eq. 8. In other words, this multinomial distribution can be interpreted as a distribution instance sampled from the Dirichlet distribution with a shape parameter, $\alpha$, according to Dirichlet-based Weight Sampling.

Next, we focus on the property of the dataset sampled with RENT. With proposition 3.1, we demonstrate that $R_{l,\text{RENT}}^{emp}$ satisfies statistical consistency to the true risk. It means that a dataset sampled from RENT can be regarded as i.i.d. sampled instances from the true clean label distribution, implying the possibility of RENT to build the noise-filtered dataset from the noisy-labelled dataset.

**Proposition 3.1.** *If $\boldsymbol{\mu}^*$ is accessible, $R_{l,RENT}^{emp}$ is statistically consistent to $R_l$ (Proof: Appendix D.3).*

## 4 EXPERIMENT

### 4.1 IMPLEMENTATION

**Datasets and Training Details**   We evaluate our method, RENT, on CIFAR10 and CIFAR100 (Krizhevsky & Hinton, 2009) with synthetic label noise and two real-world noisy dataset, CIFAR-10N (Wei et al., 2022) and Clothing1M (Xiao et al., 2015). The label noise in our experiments include 1) Symmetric flipping (Yao et al., 2020; Li et al., 2021; Bae et al., 2022) and 2) Asymmetric flipping (Li et al., 2020; Liu et al., 2020; Bae et al., 2022), marked as **SN** and **ASN**, respectively. CIFAR-10N is a real-world noisy-labelled dataset, with its label from Amazon M-turk. Clothing1M is another real-world noisy-labelled dataset with 1M images. For training, we report results with 5 times replications unless specified. See appendix F.1 for more implementation details.

$T$ **Estimation Baselines**   As RENT is a method for $T$ utilization, estimating $T$ is a prerequisite. To check the adaptability of RENT over the different estimation of $T$, we apply Forward (Patrini et al., 2017), DualT (Yao et al., 2020), TV (Zhang et al., 2021b), VolMinNet (Li et al., 2021) and Cycle (Cheng et al., 2022) as estimation methods on the experiments. For real-world label noise, we added PDN (Xia et al., 2020) and BLTM (Yang et al., 2022) as baselines for *instance dependent*

---

[3]We show the process of DWS on Appendix E.

Table 1: Test accuracies on CIFAR10 and CIFAR100 with various label noise settings. $-$ represents the training failure case. **Bold** is the best accuracy for each setting.

| Base | Risk | CIFAR10 | | | | CIFAR100 | | | |
|---|---|---|---|---|---|---|---|---|---|
| | | SN 20% | SN 50% | ASN 20% | ASN 40% | SN 20% | SN 50% | ASN 20% | ASN 40% |
| CE | ✗ | 73.4±0.4 | 46.6±0.7 | 78.4±0.2 | 69.7±1.3 | 33.7±1.2 | 18.5±0.7 | 36.9±1.1 | 27.3±0.4 |
| Forward | w/ FL | 73.8±0.3 | 58.8±0.3 | 79.2±0.6 | 74.2±0.5 | 30.7±2.8 | 15.5±0.4 | 34.2±1.2 | 25.8±1.4 |
| | w/ RW | 74.5±0.8 | 62.6±1.0 | 79.6±1.1 | 73.1±1.7 | 37.2±2.6 | 23.5±11.3 | 27.2±13.2 | 27.3±1.3 |
| | **w/ RENT** | **78.7**±0.3 | **69.0**±0.1 | **82.0**±0.5 | **77.8**±0.5 | **38.9**±1.2 | **28.9**±1.1 | **38.4**±0.7 | **30.4**±0.3 |
| DualT | w/ FL | 79.9±0.5 | 71.8±0.3 | 82.9±0.2 | 77.7±0.6 | 35.2±0.4 | 23.4±1.0 | 38.3±0.4 | 28.4±2.6 |
| | w/ RW | 80.6±0.6 | 74.1±0.7 | 82.5±0.2 | 77.9±0.4 | 38.5±1.0 | 12.0±13.5 | 38.5±1.6 | 24.0±11.6 |
| | **w/ RENT** | **82.0**±0.2 | **74.6**±0.4 | **83.3**±0.1 | **80.0**±0.9 | **39.8**±0.9 | **27.1**±1.9 | **39.8**±0.7 | **34.0**±0.4 |
| TV | w/ FL | 74.0±0.5 | 50.4±0.6 | 78.1±1.3 | 71.6±0.3 | **34.5**±1.4 | **21.0**±1.4 | 33.9±3.6 | **28.7**±0.8 |
| | w/ RW | 73.7±0.9 | 48.5±4.1 | 77.3±2.0 | 70.2±1.0 | 32.3±1.0 | 17.8±2.0 | 32.0±1.5 | 23.2±0.9 |
| | **w/ RENT** | **78.8**±0.8 | **62.5**±1.8 | **81.0**±0.4 | **74.0**±0.5 | 34.0±0.9 | 20.0±0.6 | **34.0**±0.2 | 25.5±0.4 |
| VolMinNet | w/ FL | 74.1±0.2 | 46.1±2.7 | 78.8±0.5 | 69.5±0.3 | 29.1±1.5 | 25.4±0.8 | 22.6±1.3 | 14.0±0.9 |
| | w/ RW | 74.2±0.5 | 50.6±6.4 | 78.6±0.5 | 70.4±0.8 | **36.9**±1.2 | 24.4±3.0 | 34.9±1.3 | 26.5±0.9 |
| | **w/ RENT** | **79.4**±0.3 | **62.6**±1.3 | **80.8**±0.5 | **74.0**±0.4 | 35.8±0.9 | **29.3**±0.5 | **36.1**±0.7 | **31.0**±0.8 |
| Cycle | w/ FL | 81.6±0.5 | — | **82.8**±0.4 | 54.3±0.3 | 39.9±2.8 | — | 39.4±0.2 | 31.3±1.2 |
| | w/ RW | 80.2±0.2 | 57.0±3.4 | 78.1±0.9 | **70.6**±1.1 | 37.8±2.7 | 30.2±0.6 | 38.1±1.6 | 29.3±0.6 |
| | **w/ RENT** | **82.5**±0.2 | **70.4**±0.3 | 81.5±0.1 | 70.2±0.7 | **40.7**±0.4 | **32.4**±0.4 | **40.7**±0.7 | **32.2**±0.6 |
| True $T$ | w/ FL | 76.7±0.2 | 57.4±1.3 | 75.0±11.9 | 70.7±8.6 | 34.3±0.5 | 22.0±1.5 | **35.8**±0.5 | **31.9**±1.0 |
| | w/ RW | 76.2±0.3 | 58.6±1.2 | — | — | 35.0±0.8 | 21.8±0.8 | 21.3±16.6 | 21.6±10.4 |
| | **w/ RENT** | **79.8**±0.2 | **66.8**±0.6 | **82.4**±0.4 | **78.4**±0.3 | **36.1**±1.1 | **24.0**±0.3 | 34.4±0.9 | 27.2±0.6 |

Table 2: Test accuracies on CIFAR-10N and Clothing1M. Due to the space issue, we report performances only from parts of the baselines. Please refer to Appendix F.2 for more results.

| Base | Risk | CIFAR-10N | | | | | Clothing1M |
|---|---|---|---|---|---|---|---|
| | | Aggre | Ran1 | Ran2 | Ran3 | Worse | |
| CE | ✗ | 80.8±0.4 | 75.6±0.3 | 75.3±0.4 | 75.6±0.6 | 60.4±0.4 | 66.9±0.8 |
| Forward | w/ FL | 79.6±1.8 | 76.1±0.8 | 76.4±0.4 | 76.0±0.2 | 64.5±1.0 | 67.1±0.1 |
| | w/ RW | 80.7±0.5 | 75.8±0.3 | 76.0±0.5 | 75.8±0.6 | 63.9±0.7 | 66.8±1.1 |
| | **w/ RENT** | **80.8**±0.8 | **77.7**±0.4 | **77.5**±0.4 | **77.2**±0.6 | **68.0**±0.9 | **68.2**±0.6 |
| PDN | w/ FL | 79.8±0.6 | 74.5±0.4 | 74.5±0.5 | 74.3±0.3 | 57.5±1.3 | 64.9±0.4 |
| | w/ RW | **80.6**±0.8 | 74.9±0.7 | 73.9±0.7 | 74.4±0.8 | 58.7±0.5 | — |
| | **w/ RENT** | 80.2±0.6 | **75.2**±0.7 | **75.0**±1.1 | **75.7**±0.4 | **61.6**±1.6 | **67.2**±0.2 |
| BLTM | w/ FL | **81.5**±0.7 | 78.1±0.3 | 77.5±0.6 | 77.8±0.5 | 65.8±1.0 | 67.2±0.8 |
| | w/ RW | 54.0±33.9 | 64.4±27.0 | 50.9±32.9 | 38.0±32.4 | 43.5±28.1 | 67.0±0.4 |
| | **w/ RENT** | 80.8±2.1 | **79.1**±0.9 | **78.9**±1.1 | **79.6**±0.6 | **69.7**±2.0 | **70.0**±0.4 |

transition matrix. To avoid the confusion, we denote **Forward** utilization explained in Section 2.3 as **FL** and the $T$ estimation method from Patrini et al. (2017) as Forward from now on. Also, we denote **Reweighting** utilization as **RW** for convenience. Please check Appendix F.1 for more explanations.

## 4.2 CLASSIFICATION ACCURACY

We compare **FL**, **RW** and **RENT** by applying each method to the various $T$ estimation methods. Table 1 shows the test accuracies of the classifiers trained with noisy-labelled CIFAR10 and CIFAR100.[4] Experiments are conducted with 1) estimated $T$ and 2) true $T$. First, RENT consistently outperforms FL in 42 out of 48 cases, as shown in the table. This demonstrates that RENT can improve performances of transition-based methods. It is noteworthy that the performance gaps between RENT and FL become larger in settings with higher noise ratios. Next, RENT outperforms RW in all cases except for two cases. Note that the performance gap between RW and RENT is marginal in the cases where RW is better, while the gap is significant when RENT exceeds RW.

Table 2 shows test accuracies for CIFAR-10N and Clothing1M. Again, RENT shows consistent improvement over the baselines for $T$ utilization for real-world noisy labelled dataset. Please check Appendix F.2 also for more experimental outputs including results over diverse $T$ estimations.

---

[4]Some reported performances on this paper are different from those of the original paper. We unified experimental settings, e.g. network structures, epochs, etc, that were varied in previous studies. This setting discrepancy resulted in the changes, and we provide more details in Appendix F.2.

### 4.3 IMPACT OF $\alpha$ TO DWS

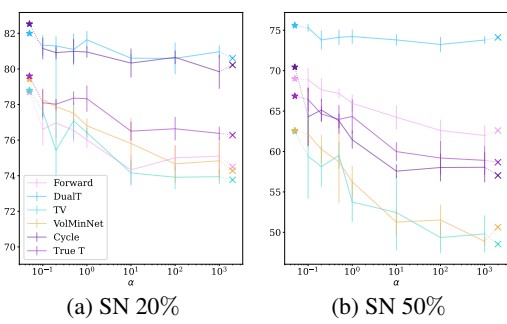

(a) SN 20%   (b) SN 50%

Figure 3: Test accuracy with regard to various $\alpha$ for CIFAR10. (**Star ($\star$)** is **RENT** and **Cross (x)** means **RW**, respectively.)

As we demonstrated in Section 3.1, $R_{l,\text{DWS}}^{emp}$ would be able to explain from $R_{l,\text{RW}}^{emp}$ to $R_{l,\text{RENT}}^{emp}$ with $\alpha$ adjustment. Also, we explained the impact of $\alpha$ to DWS in Section 3.2. Here, we report the model performances with various $\alpha$ empirically [5], along with RW and RENT. As we can see in figure 3, there is an increasing trend of test accuracy with smaller $\alpha$, and RENT shows the best performance for all cases consistently. It certainly aligns with the superior performances of RENT over RW, explaining the benefits of introducing the variance to per-sample weights term empirically. Please refer to Appendix F.3 for more results.

### 4.4 NOISE INJECTION IMPACT OF RENT

As in section 3.2, $R_{l,\text{DWS}}^{emp}$ and $R_{l,\text{SNL}}^{emp}$ shares similar form in their structures, and we compare the performance of DWS and SNL in this section. Specifically, we report the performance of RENT, $\alpha \to 0$ version of DWS. Since there is a difference in the static risk term between $R_{l,\text{SNL}}^{emp}$ and $R_{l,\text{DWS}}^{emp}$, we additionally implement the label perturbation technique suggested in the SNL paper (Chen et al., 2020) to **RW** for fair comparison. In figure 4, RENT consistently outperforms SNL, implying label perturbation alone may be insufficient for managing noisy label. Furthermore, RENT performs better or comparably with RW+$\epsilon$, indicating that RENT implicitly injects adequate noise during training process. Note that RW+$\epsilon$

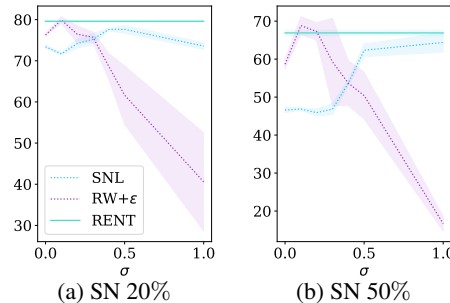

(a) SN 20%   (b) SN 50%

Figure 4: Test accuracies over various $\sigma$ for CIFAR10. RW+$\epsilon$ denotes the integration of RW and the label perturbation technique.

is highly sensitive to the value of $\sigma$, so a simple combination of RW and label perturbation technique may not be enough for wide adaptation. Please check Appendix F.4 also for more results.

### 4.5 OUTCOME ANALYSES OF RENT

$w_i$ **value**   Samples with noisy labels should have weight values close to zero ideally, indicating their exclusion. We analyze the statistics of $w_i$ after training in figure 5. It shows more than 80% of noisy labelled samples will be excluded.

We also provide statistics on the number of clean and noisy samples whose categorical distribution parameter is greater or smaller than $1/B$, respectively. For i.i.d. sampling within a mini-batch ($B$ represents the number of samples in the batch), all samples have the same weight value of $1/B$. If the normalized $\tilde{w}_i$ for $(x_i, \tilde{y}_i)$ is smaller than $1/B$, it indicates that the sample is less likely to be selected compared to the i.i.d. sampling. The presence of a large number of

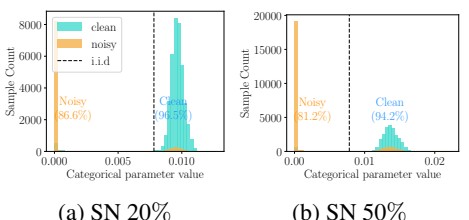

(a) SN 20%   (b) SN 50%

Figure 5: Histogram of $w_i$, of RENT on CIFAR10. Cycle for $T$ estimation. Blue and orange represents samples with clean and noisy labels, respectively. Vertical dotted line denotes $1/B$.

clean samples at the oversampling region and the concentration of noisy samples near zero support that $f_\theta$ trained with RENT effectively resamples clean samples when trained on a noisy label.

---

[5]We tested over $\alpha \in [0.1, 0.2, 0.5, 1.0, 10.0, 100.0, 1000.0]$.

**Confidence of wrong labelled samples** Next, we focus on the confidence value of samples with incorrect labels. We compare the number of samples with incorrect labels based on the confidence of the models trained using RW and RENT in figure 6. Two sets in the figure are distinguished by the threshold (0.5 in our experiment). It shows a larger proportion of noisy samples is in the *Uncertain* group when the model is trained with RENT. It implies that RW is more prone to fitting to noisy label, meaning the model memorizes incorrect labels during training.

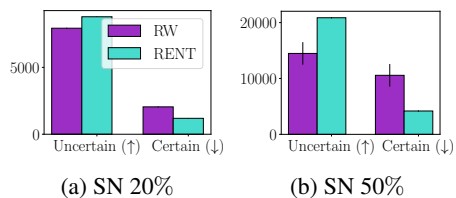

Figure 6: Training data with incorrect labels divided by the confidence (threshold=0.5). Cycle for $T$ estimation, on CIFAR10.

**Resampled dataset quality** We then analyze the quality of resampled instances by measuring their precision, recall and F1 score. Here, precision and recall can be recognized as the clarity and coverage of the resampled dataset, respectively. Figure 7 shows the efficacy of RENT over the baselines (FINE (Kim et al., 2021) and MCD (Lee et al., 2019)) considering the quality of resampled instances. RENT consistently surpasses the baselines in F1 score, meaning RENT resamples clean yet diverse samples well.

Please check Appendix F.5 also for more results of this section (Section 4.5).

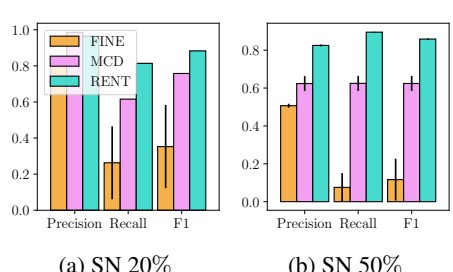

Figure 7: Resampled dataset evaluation. Cycle for $T$ estimation, on CIFAR 10.

### 4.6 ABLATION STUDY ON SAMPLING BUDGET

The size of the resampled dataset ($M$) can be modified as a hyper-parameter. In practice, the sampling size can be adjusted based on the user's need, e.g. computer memory is not enough to accommodate the huge entire noisy dataset. As part of an ablation study, we checked the classifier accuracy of $f_\theta$ when trained with different resampling budgets. Intuitively, the model performance could be either 1) higher, as it reduces the probability to resample noisy samples, or 2) lower, as it restricts the chance for model to learn various data samples. Figure 8 illustrates the classification accuracy with different resampling budget of $M$. The results tend to show higher model performance for RENT compared to FL, and indicate that RENT performs well even with smaller resampling budgets, suggesting for the further improvement of RENT.

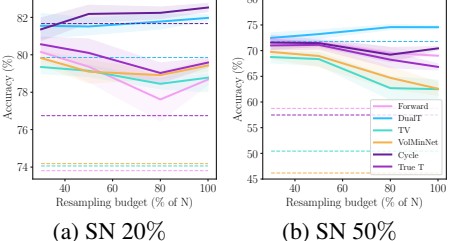

Figure 8: Ablation of $M$ on CIFAR10. Results from FL and RENT are denoted as dotted line and bold line, respectively.

For the ablation study regarding the sampling strategy, we report some results in Appendix F.6.

## 5 CONCLUSION

In this paper, we first decompose the training procedure for noisy label classification with the label transition matrix $T$ as *estimation* and *utilization*, underscoring the importance of adequate utilization. Next, we present an alternative utilization of the label transition matrix $T$ by resampling, RENT. RENT ensures the statistical consistency of risk function to the true risk for data resampling by utilizing $T$, yet it supports more robustness to learning with noisy label with the uncertainty on per-sample weights terms. By interpreting resampling and reweighting in one framework through Dirichlet distribution-based per-sample Weight Sampling (DWS), we integrated both techniques and analyzed the success of resampling over reweighting in learning with noisy label. Our benchmark experiments with synthetic and real-world label noises show consistent improvements over the existing $T$ utilization methods as well as the reweighting methods.

ACKNOWLEDGMENTS

This research was supported by Research and Development on Key Supply Network Identification, Threat Analyses, Alternative Recommendation from Natural Language Data (NRF) funded by the Ministry of Education (2021R1A2C200981613).

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

## A    ILLUSTRATION OF DWS IMPLEMENTATION EXAMPLE

Figure 9 shows an implementation example of our algorithm, DWS, as RW (previous) and RENT (ours).

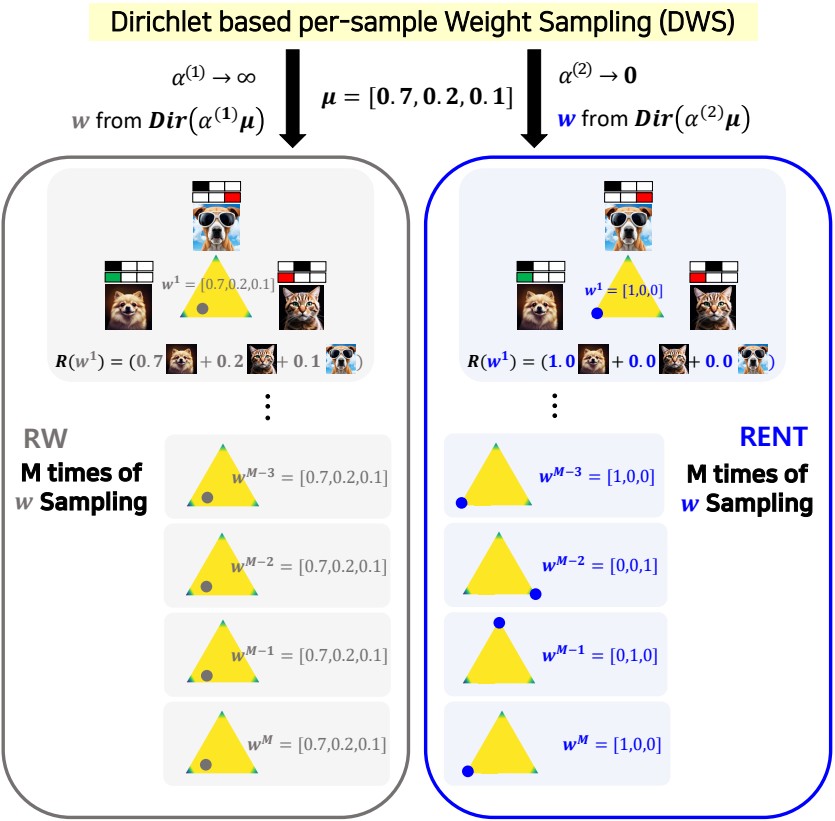

Figure 9: Dirichlet distribution-based per-sample Weight Sampling (DWS) with shape parameter $\alpha$ and the mean vector $\boldsymbol{\mu}$. Similar to Figure 1, image at the vertices of yellow triangles represents data instance. Blocks above the images represent **Class (upper)**, **Label (lower)**. Green means labels are same as true class (clean) and Red means labels are different from true class (noisy). $\boldsymbol{R(w^\dagger)}$ represents the risk function with $\boldsymbol{w^\dagger}$ as per-sample weight vector. Each colored box represents one sampled $\boldsymbol{w}$.

In DWS framework, we sample per-sample weights vectors, $\boldsymbol{w}$ for M times. Each sampled $\boldsymbol{w}^j$ becomes a per-sample weight vector. From the property of the Dirichlet distribution, if the shape parameter $\alpha \to \infty$, sampled $\boldsymbol{w}$ will be near to the mean vector. However, if $\alpha \to 0$, sampled $\boldsymbol{w}$ will be clustered to vertices. Due to the space issue, we showed only one sampled $\boldsymbol{w}$ in figure 1, while we illustrate more times of per-sample weights sampling here.

## B    PREVIOUS STUDIES

We first explain previous studies for learning with noisy labels. Next, focusing on the studies utilizing the transition matrix, we summarize previous studies considering the estimation of the transition matrix, which were not included in the main paper.

### B.1    LEARNING WITH NOISY LABEL

Considering learning with noisy label, several directions have been suggested, including sample selection (Han et al., 2018; Yu et al., 2019; Wei et al., 2020; Cheng et al., 2020), label correction

(Tanaka et al., 2018; Li et al., 2020; Zheng et al., 2020; Wang et al., 2021) and robust loss (Zhang & Sabuncu, 2018; Wang et al., 2019). Sample selection-based methods manage noisy-labelled instances by setting the objective as removing them during training. For example, Han et al. (2018) assumed that samples whose losses are small during training have clean labels. However, this will cause learning bias toward early learned easy samples because the deep neural network will easily overfit to those samples. As a solution, Han et al. (2018) proposes using two same-structured networks with different initialization point, and utilize the output of the other classifier as the metric to decide whether to select or not each sample for training a classifier. However, since two different classifiers have still finally converged to the same output, Yu et al. (2019) has proposed to select samples when the outputs of the sample from the two different networks are different. It may solve the problem of two different networks' alignment considering the output of a sample after enough time, it may not work better even than Han et al. (2018) since it selects too small portion of samples, especially when the noise ratio is high. Wei et al. (2020) focused on this problem and they relieve this limitation by updating two networks together with introducing the KL divergence between the output of the two networks as regularization, making the output of two networks become closer to true labels and that of their peer network's. Cheng et al. (2020) selects samples with dynamic threshold and regularize confidences of samples.

On the other hand, Label correction-based methods do not waste samples with recycling noisy-labelled instances by relabelling them. For example, Tanaka et al. (2018) optimizes both the classifier parameters and labels of data instances jointly, initialize the network parameters and train with the modified labels again. Li et al. (2020) considers samples with large loss as unlabeled samples and solve the noisy-labelled dataset problems utilizing semi-supervised learning techniques, giving the noisy-labelled samples pseudo-labels. Zheng et al. (2020) calculates the likelihood ratio between the classifier's confidence on noisy label and its confidence on its (assumed) true label prediction as its threshold to configure clean labeled dataset, and corrects the label into the prediction for samples with low likelihood ratio iteratively. Wang et al. (2021) analyzes several types of label modification approaches and suggests label correction regularization hyperparameter depending both on learning time stage and confidence of a sample.

Studies modeling the loss function which is more robust to noisy label include Zhang & Sabuncu (2018), which combine the advantages of the mean absolute loss (MAE) and the cross entropy loss (CE) and Wang et al. (2019), which adds the original cross entropy and reverse cross entropy. Apart from that, studies like Liu et al. (2020) relies on the fact that the deep neural networks memorize the noisy labels slowly, and they regularize the network not to memorize the noisy labels by maximizing the inner product between the model output and the weighted outputs of previous epochs.

Although these studies would show good performances empirically, these studies cannot ensure statistical consistency of (1) classifier or (2) the risk function to the true one Yao et al. (2020). Therefore, we now focus on the studies which utilize $T$.

## B.2 Previous Researches on $T$ Estimation

In this section, we explain more details of the previous studies considering $T$.

$T$ estimation under Class-Conditional Noise (CCN) setting stems from Patrini et al. (2017). It defines $T$ as the matrix of the transition probability from clean label to noisy label. It also suggest two loss structures, Forward loss and Backward loss, which ensures statistical consistency of classifier and statistical consistency of risk, respectively. Since these two loss structures ensure theoretical success of learning with noisy label utilizing $T$, studies have focused on the estimation process of $T$, since $T$ is actually unknown information.

Patrini et al. (2017) estimate $T$ with anchor points, yet finding the explicit anchor points in dataset may be unrealistic and difficult. Therefore, Yao et al. (2020) decomposes the objective of estimation as 2 easily learnable matrices: 1) a transition matrix from noisy label to bayes optimal label and 2) a transition matrix from bayes label to true label. Since bayes optimal label is one-hot label, 1) can be estimated by summing up the samples with specific noisy label and the specific bayes optimal label, and 2) estimation would be easier than the original $T$ estimation. Zhang et al. (2021b) points out the overconfidence issue for $T$ estimation, and find a way to estimate optimal $T$ by maximizing the total variation distance between clean label posterior probabilities. However, it has theoretical assumption of having anchor points and requires ensuring the multiplication of two matrices ($V$ and $U$) should

be equal to true $T$. Li et al. (2021) is a research studied at similar time as Zhang et al. (2021b), and it finds the optimal $T$ by minimizing the volume of simplex enclosed by the columns of $T$. It relieves the anchor point assumption, yet it is still vulnerable to overconfidence issue reported at Zhang et al. (2021b). Motivated by this error gap of $T$ estimation, Cheng et al. (2022) suggests a comprehensive loss of utilizing both Forward and Backward loss structure. It shows good performance empirically, but the optimal $T$ learned by Cheng et al. (2022) would be an identity matrix by its modeling, since both $T$ and $T'$, which is an approximation of $T^{-1}$, are modeled as diagonally dominant and all cells are nonnegative.

Apart from these directions, Zhu et al. (2021) calculates $T$ by solving the optimization problem with constraints. These constraints stems from *Clusterability*, which assumes samples with similar features would have same true label. As an effort to reflect the feature information to transition probability, $T$ estimation processes under IDN condition have been studied. For example, Xia et al. (2020) assumes weighted sum of transition matrices of parts can explain instance-dependent transition matrix. Berthon et al. (2021) assumes accessibility to $p(y = i|\tilde{y} = i, x)$ for every sample and calculates $p(\tilde{Y}|Y, X)$. Yang et al. (2022) parameterize $T(X)$ and trains a new deep neural network which gives instance-dependent transition matrix as its output using distilled dataset, which is the subset of the original training dataset with its maximum softmax output is large enough. Although modeling instance dependent transition matrix may be more realistic, estimating $T(X)$ is far more difficult because its domain space becomes much bigger than $T$ modeling. Therefore, its estimation is impossible without additional assumptions or information (Liu et al., 2023), since the true $C \times C$ matrix cell values per every single samples would have unbounded solutions per each sample, making it harder to analyze the properties of estimated $T(X)$.

Please note that to write as $\boldsymbol{p}(\tilde{Y}|x) = T(x)\boldsymbol{p}(Y|x)$, the definition of $T(x)$ notation should be as $T_{jk}(x) = p(\tilde{Y} = j|Y = k, x) \ \forall j.k = 1, ..., C$. Take a 2-dimension example. It should be as:

$$\begin{bmatrix} p(\tilde{Y} = 1|x) \\ p(\tilde{Y} = 2|x) \end{bmatrix} = \begin{bmatrix} p(\tilde{Y} = 1|Y = 1, x) & p(\tilde{Y} = 1|Y = 2, x) \\ p(\tilde{Y} = 2|Y = 1, x) & p(\tilde{Y} = 2|Y = 2, x) \end{bmatrix} \begin{bmatrix} p(Y = 1|x) \\ p(Y = 2|x) \end{bmatrix} \tag{9}$$

meaning that the $(j, k)-$th element of $T(x)$ should be $p(\tilde{Y} = j|Y = k, x)$.

## C  DISCUSSIONS FOR PREVIOUS TRANSITION MATRIX UTILIZATION

We assume that $T$ is invertible, which has been generally assumed in the previous researches.

First we discuss **Forward** utilization. As we defined in the main paper, let $\epsilon$ be $\tilde{\boldsymbol{p}}(\tilde{Y}|x) - \boldsymbol{p}(\tilde{Y}|x)$, where $\tilde{\boldsymbol{p}}(\tilde{Y}|x)$ is noisy label probability vector estimated from the classifier trained with noisy labels. Then, $\boldsymbol{p}(\tilde{Y}|x)$ becomes $\tilde{\boldsymbol{p}}(\tilde{Y}|x) - \epsilon$, which means, $T\boldsymbol{p}(Y|x) = \boldsymbol{p}(\tilde{Y}|x) = \tilde{\boldsymbol{p}}(\tilde{Y}|x) - \epsilon$. Therefore, $\boldsymbol{p}(Y|x)$ becomes $T^{-1}(\tilde{\boldsymbol{p}}(\tilde{Y}|x) - \epsilon) = T^{-1}\tilde{\boldsymbol{p}}(\tilde{Y}|x) - T^{-1}\epsilon$.

Following Eq. 2 on the main paper, $f_\theta(x)$ will be equal to $\boldsymbol{p}(Y|x)$ if and only if $Tf_\theta(x) = \boldsymbol{p}(\tilde{Y}|x)$ for all $(x_i, \tilde{y}_i)_{i=1}^n$. It means that $f_\theta(x) = \boldsymbol{p}(Y|x) \Leftrightarrow R_{l,F} = 0$ and $\boldsymbol{p}(\tilde{Y}|x)$ is required for training $f_\theta$. However, $\boldsymbol{p}(\tilde{Y}|x)$ should approximated by the noisy labels from the dataset since the exact value of $\boldsymbol{p}(\tilde{Y}|x)$ is unknown and it makes the gap. In other words, we get $f_\theta$ by minimizing $R_{l,F}^{emp} = \frac{1}{N}\sum_{i=1}^N l(Tf_\theta(x_i), \tilde{y}_i)$, and if we minimize $R_{l,F}^{emp}$, $f_\theta(x)$ will be $T^{-1}\tilde{\boldsymbol{p}}(\tilde{Y}|x)$ so that the gap between $f_\theta^*(x)$ and $f_\theta(x)$ will become $T^{-1}\epsilon$.

The difficulty of estimating $\boldsymbol{p}(\tilde{Y}|x)$ from learning a model $Tf_\theta(x)$ with $\tilde{D}$ has already been denoted before. Remark C.1 is one of those proposals.

**Remark C.1.** (Zhang et al., 2021b) The estimation error of the noisy label posterior distribution, $\boldsymbol{p}(\tilde{Y}|x)$, from neural networks trained with $\tilde{D}$ could be high. This confidence calibration might be more difficult than $\boldsymbol{p}(Y|X)$ estimation, because $\boldsymbol{p}(\tilde{Y}|x)$ should be within the convex hull of transition matrix $T$, $\text{Conv}(T)$, i.e., $\boldsymbol{p}(\tilde{Y}|x) \in \text{Conv}(T) \subset \Delta^{C-1}$, and $\text{Conv}(T) \neq \Delta^{C-1}$ if $T \neq I$.

Remark C.1 claims the difficulty of estimating $\boldsymbol{p}(\tilde{Y}|x)$ from $\tilde{D}$, because the value of $\boldsymbol{p}(\tilde{Y}|x)$ does not achieve the full range of $[0, 1]$. Intuitively, $\boldsymbol{p}(\tilde{Y}|x)$ would not be represented as one-hot because

the noisy label generation process would not be deterministic, which is also claimed by previous works Yao et al. (2020); Zhang et al. (2021b); Yao et al. (2021).

This failure of $\boldsymbol{p}(\tilde{Y}|x)$ estimation is a crucial issue considering the statistical consistency of $f_\theta$ to $f_\theta^*$ since $f_\theta(x) = \boldsymbol{p}(Y|x)$ if and only if $T f_\theta(x) = \boldsymbol{p}(\tilde{Y}|x)$. Therefore, the gap between $\boldsymbol{p}(\tilde{Y}|x)$ and $\tilde{\boldsymbol{p}}(\tilde{Y}|x)$ make the inevitable gap for estimating $\tilde{\boldsymbol{p}}(Y|x)$. It means that a classifier trained with empirical **Forward** risk may not be able to estimate $\boldsymbol{p}(Y|x)$ accurately even with true $T$.

We also propose $f_\theta$ trained with $R_{l,\mathrm{F}}^{emp}$ can memorize all noisy label under the following assumption.

**Assumption C.2.** Given a noisy training dataset $\tilde{D} = \{(x_i, \tilde{y}_i)\}_{i=1}^n$, let $x_i \neq x_j$ for all $i \neq j$, and $T_{jj} > T_{jk}$ for all $j \neq k$ of $T$, and $f_\theta$ has enough capacity to memorize all labels.

For the each term of the risk function $R_{l,\mathrm{F}}^{emp}$, we have $l\left(T f(x_i), \tilde{y}_i\right) = -\log \sum_{j=1}^C T_{\tilde{y}_i j} f_j(x_i) \geq -\log T_{\tilde{y}_i \tilde{y}_i} f_{\tilde{y}_i}(x_i)$. It is from $\sum_{j=1}^C T_{\tilde{y}_i j} f_j(x_i) \leq \sum_{j=1}^C \left(\max_k T_{\tilde{y}_i k}\right) f_j(x_i) = T_{\tilde{y}_i \tilde{y}_i}$. Note that $f_k(x) \geq 0$ for all $k$ and $\sum_{k=1}^C f_k(x) = 1$. Also $\left(\max_k T_{\tilde{y}_i k}\right) = T_{\tilde{y}_i \tilde{y}_i}$ by the assumption C.2.

$\sum_{j=1}^C T_{\tilde{y}_i j} f_j(x_i) = T_{\tilde{y}_i \tilde{y}_i}$ when $f_j(x_i) = 1$ for $j = \tilde{y}_i$ and $0$ otherwise. Therefore, $f_\theta$ that minimizes $R_{l,\mathrm{F}}^{emp}$ can result in $f_\theta^{\mathrm{F},emp}(x_i) = \tilde{y}_i$ for all $i = 1, ..., n,$.

Figure 10 supports the above proposal empirically. It depicts the train accuracies with noisy labels (left), train accuracies with the original true labels (center) and test accuracies (right) of various methods with **Forward**, where each method is trained on the noisy-labelled dataset. As training iterations progress, all methods utilizing **Forward** eventually memorize the noisy label, which leads to the degradation on the test accuracy. Note that the learning with true $T$ (denoted as True $T$) also memorizes the noisy label. In contrast, methods equipped with RENT alleviate noisy label memorization; they also show the correct inference capabilities for the clean labels of noisy training instances, which is an evidence of non-memorization.

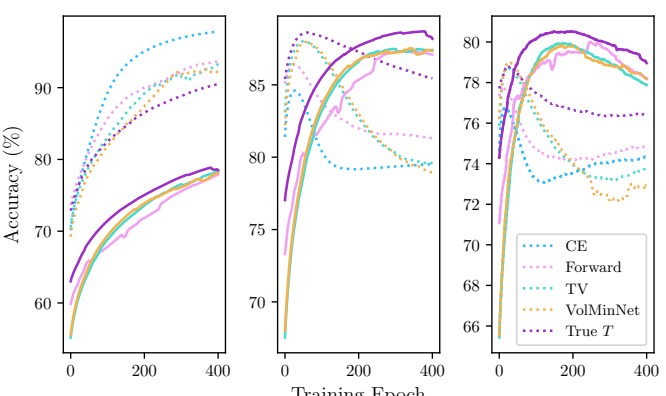

Figure 10: Training accuracies with regard to the noisy labels (**left**), the clean labels (**center**), and test accuracies (**right**) of various methods on CIFAR10 (symmetric 20% noise). We applied **Forward** and our method, RENT, to various $T$ estimation Patrini et al. (2017); Zhang et al. (2021b); Li et al. (2021). We express the result of **Forward** and the result of RENT as dotted line and bold line, for each estimation respectively. CE is the result of the classifier trained with Cross Entropy loss.

We also discuss **Backward** briefly. Please note that (with $l$ as Cross Entropy loss)

$$\left(\boldsymbol{l}(x) T^{-1}\right)_{\tilde{y}} = \sum_{k=1}^C l_k T_{k,\tilde{y}}^{-1} = -\sum_{k=1}^C T_{k,\tilde{y}}^{-1} \log(f_\theta(x)_k) \tag{10}$$

If $T = I$, then since $T^{-1} = I$, Eq. 10 becomes $-\log f_\theta(x)_{\tilde{y}}$.

If $T \neq I$, we first think of $C = 2$ case, when $T^{-1}$ has a simple form.

As $f_\theta(x)_{k \neq \tilde{y}} = 1 - f_\theta(x)_{\tilde{y}}$, the derivative of the empirical backward risk over $f_\theta(x)_{\tilde{y}}$ becomes $-\frac{T_{\tilde{y},\tilde{y}}^{-1}}{f_\theta(x)_{\tilde{y}}} + \frac{T_{k \neq \tilde{y},\tilde{y}}^{-1}}{1 - f_\theta(x)_{\tilde{y}}}$. Note that $T^{-1} = \frac{1}{T_{11} T_{22} - T_{12} T_{21}} \begin{bmatrix} T_{22} & -T_{12} \\ -T_{21} & T_{11} \end{bmatrix}$. Assuming $T_{11} > T_{21}$ and $T_{22} > T_{12}$, both $T_{12}^{-1}$ and $T_{21}^{-1}$ become negative. It means that $T_{k \neq \tilde{y},\tilde{y}}^{-1}$ is negative, making the derivative when $f_\theta(x)_{\tilde{y}}$ near $0$ or $1$ goes to $-\infty$.

For $C > 2$ case, the direct analysis is impossible because there is no explicit form for the inverse matrix. However, since $T \in [0, 1]^{C \times C}$, the inverse of the transition matrix can easily be negative if it is not an identity matrix so the same issue may happen.

## D   DERIVATION AND PROOF

### D.1   DERIVATION OF EQ. 6

Here, we show the derivation of Eq. 6 fully. Notations are same as the main paper, which means, $\boldsymbol{w^j} \in \mathbb{R}^N$ is $j$-th sample following $\text{Dir}(\alpha\boldsymbol{\mu})$. Let $\bar{w}_i := \sum_{j=1}^{M} \frac{w_i^j}{M}$. $l_i = l(f_\theta(x_i), \tilde{y}_i)$ for all $i = 1, ..., N$ for convenience.

$$R_{l,\text{DWS}}^{emp} = \frac{1}{M} \sum_{j=1}^{M} \sum_{i=1}^{N} w_i^j l_i = \sum_{i=1}^{N} \frac{1}{M} \sum_{j=1}^{M} w_i^j l_i = \sum_{i=1}^{N} \left( \sum_{j=1}^{M} \frac{w_i^j}{M} \right) l_i = \sum_{i=1}^{N} \bar{w}_i l_i \qquad (11)$$

Since $w_i^1, ..., w_i^j$ is i.i.d. for same $i$, we can apply Central Limit Theorem: $\bar{w}_i \sim \mathcal{N}\left(\mu_i, \frac{\mu_i(1-\mu_i)}{M(\alpha+1)}\right)$.

$$\begin{aligned} R_{l,\text{DWS}}^{emp} &= \sum_{i=1}^{N} \bar{w}_i l_i = \sum_{i=1}^{N} \left( \bar{w}_i - \mu_i + \mu_i \right) l_i \\ &= \sum_{i=1}^{N} \mu_i l_i + \sum_{i=1}^{N} \left( \bar{w}_i - \mu_i \right) l_i, \text{ with } \left( \bar{w}_i - \mu_i \right) \sim \mathcal{N}\left( 0, \frac{\mu_i(1-\mu_i)}{M(\alpha+1)} \right) \end{aligned} \qquad (12)$$

### D.2   EXPLANATION FOR THE CATEGORICAL DISTRIBUTION PARAMETER OF RENT

Here, we explain how the per-sample weights can be formulated as $\frac{p(Y=\tilde{y}|x)}{(T\boldsymbol{p}(Y|x))_{\tilde{y}}} = \frac{p(Y=\tilde{y}|x)}{p(\tilde{Y}=\tilde{y}|x)}$.

Following the derivation from Liu & Tao (2015), The likelihood ratio between $p(X, Y)$ and $p(X, \tilde{Y})$ will be same as the likelihood ratio between $p(Y|X)$ and $p(\tilde{Y}|X)$. It is because $p(X)$ does not change for noisy label classification task.

$$\begin{aligned} R_l(f_\theta) &= \mathbb{E}_{(x,y)\sim p(X,Y)} \left[ l(f_\theta(x), y) \right] = \mathbb{E}_{(x,\tilde{y})\sim p(X,\tilde{Y})} \left[ l(f_\theta(x), \tilde{y}) \frac{p(x, Y = \tilde{y})}{p(x, \tilde{Y} = \tilde{y})} \right] \\ &= \mathbb{E}_{(x,\tilde{y})\sim p(X,\tilde{Y})} \left[ l(f_\theta(x), \tilde{y}) \frac{p(Y = \tilde{y}|x)p(x)}{p(\tilde{Y} = \tilde{y}|x)p(x)} \right] = \mathbb{E}_{(x,\tilde{y})\sim p(X,\tilde{Y})} \left[ l(f_\theta(x), \tilde{y}) \frac{p(Y = \tilde{y}|x)}{p(\tilde{Y} = \tilde{y}|x)} \right] \\ &= \mathbb{E}_{(x,\tilde{y})\sim p(X,\tilde{Y})} \left[ \frac{p(Y = \tilde{y}|x)}{p(\tilde{Y} = \tilde{y}|x)} l(f_\theta(x), \tilde{y}) \right] \end{aligned} \qquad (13)$$

Following the concept of Sampling-Importance-Resampling (SIR) (Rubin, 1988; Smith & Gelfand, 1992), this importance sampling is transformed as algorithm 1.

### D.3   PROOF OF PROPOSITION 3.1

**Proposition D.1.** *If $\boldsymbol{\mu}^*$ is accessible, $R_{l,RENT}^{emp}$ is statistically consistent to $R_l$.*

Note that the true mean of weight vectors, $\boldsymbol{\mu}^*$, is assumed to be accessible for this part. In other words, we assume $\left[ \frac{\tilde{\mu}_1}{\sum_{l=1}^{N} \tilde{\mu}_l}, ..., \frac{\tilde{\mu}_N}{\sum_{l=1}^{N} \tilde{\mu}_l} \right]$ is equal to $\boldsymbol{\mu}^*$, with $\tilde{\mu}_i = \frac{f_\theta(x_i)_{\tilde{y}_i}}{(Tf_\theta(x_i))_{\tilde{y}_i}}$ for all $i$. $\tilde{\mu}_i^* = \frac{p(Y=\tilde{y}_i|x_i)}{p(\tilde{Y}=\tilde{y}_i|x_i)}$ and $\boldsymbol{\mu}^* = \left[ \frac{\tilde{\mu}_1^*}{\sum_{l=1}^{N} \tilde{\mu}_l^*}, ..., \frac{\tilde{\mu}_N^*}{\sum_{l=1}^{N} \tilde{\mu}_l^*} \right]$ as in the main paper.

*Proof.* We first start by rewriting the risk function of RENT (Same as the main paper). Let $\tilde{\mathcal{D}}_N = \{(x_i, \tilde{y}_i)\}_{i=1}^{N}$ is the training dataset, which comes from $p(X, \tilde{Y}) = Tp(X, Y)$, with size $N$.

$$R_{l,\text{RENT}}^{emp} := \frac{1}{M} \sum_{i=1}^{N} n_i l(f_\theta(x_i), \tilde{y}_i), \text{ where } [n_1, ..., n_N] \sim \text{Multi}(M; \frac{\tilde{\mu}_1}{\sum_{l=1}^{N} \tilde{\mu}_l}, ..., \frac{\tilde{\mu}_N}{\sum_{l=1}^{N} \tilde{\mu}_l}) \qquad (14)$$

Then,

$$
\mathbb{E}\left[R_{l,\text{RENT}}^{emp}\right] = \mathbb{E}\left[\mathbb{E}\left[R_{l,\text{RENT}}^{emp}|\tilde{\mathcal{D}}_N\right]\right] = \mathbb{E}\left[\mathbb{E}\left[\frac{1}{M}\sum_{i=1}^{N} n_i l(f_\theta(x_i),\tilde{y}_i)\Big|\tilde{\mathcal{D}}_N\right]\right]
$$

$$
= \mathbb{E}\left[\frac{1}{M}\sum_{i=1}^{N}\mathbb{E}\left[n_i l(f_\theta(x_i),\tilde{y}_i)|\tilde{\mathcal{D}}_N\right]\right] = \mathbb{E}\left[\sum_{i=1}^{N}\frac{\tilde{\mu}_i}{\sum_{k=1}^{N}\tilde{\mu}_k}l(f_\theta(x_i),\tilde{y}_i)\right] \quad (15)
$$

By the assumption above,

$$
\sum_{i=1}^{N}\frac{\tilde{\mu}_i}{\sum_{k=1}^{N}\tilde{\mu}_k}l(f_\theta(x_i),\tilde{y}_i) = \sum_{i=1}^{N}\frac{\tilde{\mu}_i^*}{\sum_{k=1}^{N}\tilde{\mu}_k^*}l(f_\theta(x_i),\tilde{y}_i)
$$

$$
= \frac{1}{\sum_{k=1}^{N}\tilde{\mu}_k^*}\sum_{i=1}^{N}\tilde{\mu}_i^* l(f_\theta(x_i),\tilde{y}_i) = \frac{1}{\sum_{k=1}^{N}\tilde{\mu}_k^*}\sum_{i=1}^{N}\frac{p(Y=\tilde{y}_i|x_i)}{p(\tilde{Y}=\tilde{y}_i|x_i)}l(f_\theta(x_i),\tilde{y}_i)
$$

$$
= \frac{1}{\sum_{k=1}^{N}\tilde{\mu}_k^*}\sum_{i=1}^{N}\frac{p(Y=\tilde{y}_i|x_i)p(x_i)}{p(\tilde{Y}=\tilde{y}_i|x_i)p(x_i)}l(f_\theta(x_i),\tilde{y}_i) = \frac{1}{\sum_{k=1}^{N}\tilde{\mu}_k^*}\sum_{i=1}^{N}\frac{p(Y=\tilde{y}_i,x_i)}{p(\tilde{Y}=\tilde{y}_i,x_i)}l(f_\theta(x_i),\tilde{y}_i)
$$

$$
= \frac{1}{\frac{1}{N}\sum_{k=1}^{N}\frac{p(Y=\tilde{y}_k,x_k)}{p(\tilde{Y}=\tilde{y}_k,x_k)}} \times \frac{1}{N}\sum_{i=1}^{N}\frac{p(Y=\tilde{y}_i,x_i)}{p(\tilde{Y}=\tilde{y}_i,x_i)}l(f_\theta(x_i),\tilde{y}_i) \quad (16)
$$

$\sum_{k=1}^{N}\tilde{\mu}_k^*$ works as a constant term with regard to $i$. Then, $R_{l,\text{RENT}}^{emp}$ converges to $\mathbb{E}_{p(X,Y)}[R_l]$ by the strong law of large number as $N$ goes to infinity. (Note that $\tilde{\mathcal{D}}_N$ comes from $p(X,\tilde{Y})$.) Therefore, $R_{l,\text{RENT}}^{emp}$ is statistically consistent to the true risk. $\qquad\square$

## E  ALGORITHM FOR DIRICHLET BASED SAMPLING

Here, we show the algorithm of dirichlet based per-sample weight sampling process.

---

**Algorithm 2:** Dirichlet distribution-based per-sample Weight Sampling (DWS)

---

**Input:** Noisy dataset $\tilde{D} = \{x_i, \tilde{y}_i\}_{i=1}^N$, classifier $f_\theta$, Transition matrix $T$, concentration
  parameter $\alpha$, the number of sampling $M$
**Output:** Updated $f_\theta$
  **while** $f_\theta$ *not converge* **do**
    Get $\mu_i = \tilde{\mu}_i / \sum_{j=1}^{N}\tilde{\mu}_j$, where $\tilde{\mu}_i = f_\theta(x_i)_{\tilde{y}_i}/(Tf_\theta(x_i))_{\tilde{y}_i}$ for all $i$
    **for** $j = 1, ..., M$ **do**
      Independently sample $\boldsymbol{w^j}$ from $\text{Dir}(\alpha, \boldsymbol{\mu})$
    **end**
    Update $f_\theta$ by $\theta \leftarrow \theta - \nabla_\theta \frac{1}{M}\sum_{j=1}^{M}\sum_{i=1}^{N}w_i^j l(f_\theta(x_i),\tilde{y}_i)$
  **end**

---

## F  EXPERIMENT

### F.1  IMPLEMENTATION DETAILS

**Network Architecture and Optimization**  We utilized ResNet34(He et al., 2016) for CIFAR10 and ResNet50(He et al., 2016) for CIFAR100. We used Adam Optimizer (Kingma & Ba, 2014) with learning rate 0.001 for training. No learning rate decay was applied. We trained total 200 epochs for all benchmark datasets and no validation dataset was utilized for early stopping. We utilized batch size of 128 and as augmentation, HorizontalFlip and RandomCrop were applied.

Considering Clothing1M, we used ResNet50 pretrained with ImageNet (Deng et al., 2009). As a same condition with benchmark dataset setting, we utilized Adam Optimizer with learning rate 0.001 with no learning rate decay. We trained 10 epochs and set batch size as 100. RandomCrop, RandomHorizontalflip and Normalization was applied during training, and only Centercrop and Normalization was applied at testing. For experiments over Clothing1M, we do not use a clean validation dataset in training.

Unless being specified, we keep this experimental settings for other experiments.

**Synthetic noisy label generation**    Considering CIFAR10 and CIFAR100, all samples from these benchmark dataset are assumed to have clean labels. Therefore, we arbitrarily inject noisy labels following the rules below. Let $\tau\%$ a noisy label ratio.

(1) Symmetric flipping (**SN**) Yao et al. (2020); Zhang et al. (2021b); Li et al. (2021); Bae et al. (2022) flips labels uniformly to all other classes. We set $(1-\tau)\%$ samples of each class unchanged.

(2) Asymmetric flipping (**ASN**) Li et al. (2020); Liu et al. (2020); Cheng et al. (2022); Bae et al. (2022) flips labels to pre-defined similar class. For CIFAR10, we flipped label class as Truck $\Rightarrow$ Automobile, Bird $\Rightarrow$ Airplane, Deer $\Rightarrow$ Horse, Cat $\Leftrightarrow$ Dog following the previous researches. For CIFAR100, we flipped between sub-classes within each super-class.

**Dataset description**    CIFAR-10N (Wei et al., 2022) contains real-world noisy-labels of CIFAR10 images from Amazon Mechanical Turk. The label noise includes 5 types; Aggregate (**Aggre**), Random1 (**Ran1**), Random2 (**Ran2**), Random3 (**Ran3**) and **Worse**. For detailed descriptions of how the noisy labels of each noise type are created, please refer to the original paper. According to the original paper, the noise ratio is $9.03\%$, $17.23\%$, $18.12\%$, $17.64\%$ and $40.21\%$, respectively.

Clothing 1M (Xiao et al., 2015) is real-world noisy-labelled dataset collected from online shopping websites. The dataset includes 1 million images with 14 classes. Noise ratio is estimated as $38\%$.

**Baseline description**    Here, we explain methods that we used in experiments to estimate $T$.

**Forward** Patrini et al. (2017) identifies anchor points based on the calculated noisy class posterior probabilities. Following the customs of the original paper, we chose the top $3\%$ confident sample for each class as anchor points.

**DualT** Yao et al. (2020) decomposes $T$ into 1) a transition from noisy label to intermediate class; and 2) a transition from intermediate to true class to reduce the estimation gap of $T$.

**TV** Zhang et al. (2021b) estimates $T$ by maximizing the total variation distance between clean label probabilities of samples. Although it requires an anchor point assumption theoretically, it does not need to find anchor points explicitly.

**VolMinNet** Li et al. (2021) estimates the optimal $T$ by minimizing the volume of the simplex formed by the column vectors of $T$. The volume is measured as log of the determinant of $T$ as original paper.

**Cycle** Cheng et al. (2022) develops an alternative method, which minimizes volume of $T$, without estimating the noisy class posterior probabilities as VolMinNet. It minimizes the comprehensive loss which takes the form of forward plus backward plus the regularization term which obligates the multiplication of $T$ and $T^{'}$ to be an identity.

**PDN** (Xia et al., 2020) assumes weighted sum of transition matrices of parts can explain instance-dependent transition matrix. Therefore it first calculates the transition matrix for anchor parts and get instance-wise weights to multiply to each part matrix.

**BLTM** (Yang et al., 2022) parameterize $T(X)$ and trains a new deep neural network which gives instance-dependent transition matrix as its output using distilled dataset, which is the subset of the original training dataset with its maximum softmax output is large enough.

We did not consider **CSIDN** (Berthon et al., 2021) as our baselines because they requires the information of $p(Y = \tilde{y}_i | \tilde{Y} = \tilde{y}_i)$, although we do not have those kind of information basically.

**Techniques used to hinder memorization in previous researches** We report regularization techniques in previous researches used to hinder memorization as table 3. Settings are reported based on CIFAR10 experiment conditions.

Table 3: Regularizations to hinder noisy label memorization used in previous researches (CIFAR10)

| Method | Dropout | Early Stopping | Lr decay | W decay | Augmentation |
|---|---|---|---|---|---|
| Forward Patrini et al. (2017) | X | O | O | $10^{-4}$ | HorizontalFlip, RandomCrop |
| DualT Yao et al. (2020) | X | O | O | $10^{-4}$ | HorizontalFlip, RandomCrop, Normalization |
| TV Zhang et al. (2021b) | X | X* | X | $10^{-4}$ | HorizontalFlip, RandomCrop, Normalization |
| VolMinNet Li et al. (2021) | X | O | O | $10^{-3}$ | HorizontalFlip, RandomCrop |
| Cycle Cheng et al. (2022) | X | O | O | $10^{-3}$ | Not reported |
| Ours | X | X | X | X | HorizontalFlip, RandomCrop |

We marked the asterisk on the early stopping of TV Zhang et al. (2021b) since the number of epoch is different (40) from our setting (200). As reported in table 3, we conducted experiments removing several techniques without augmentation. We show the difference between the reported performance in the original paper and the reproduced performance following their condition, the reproduced performance under our experiment settings in the following section.

## F.2 MORE RESULTS FOR CLASSIFICATION ACCURACY

**Performance with regard to the noise ratio** Figure 11 shows the performance comparison of FL, RW and RENT changing the noise ratio for several $T$ estimation methods. As we can see in the figure, the gaps between red lines and others become larger with higher noise ratio.

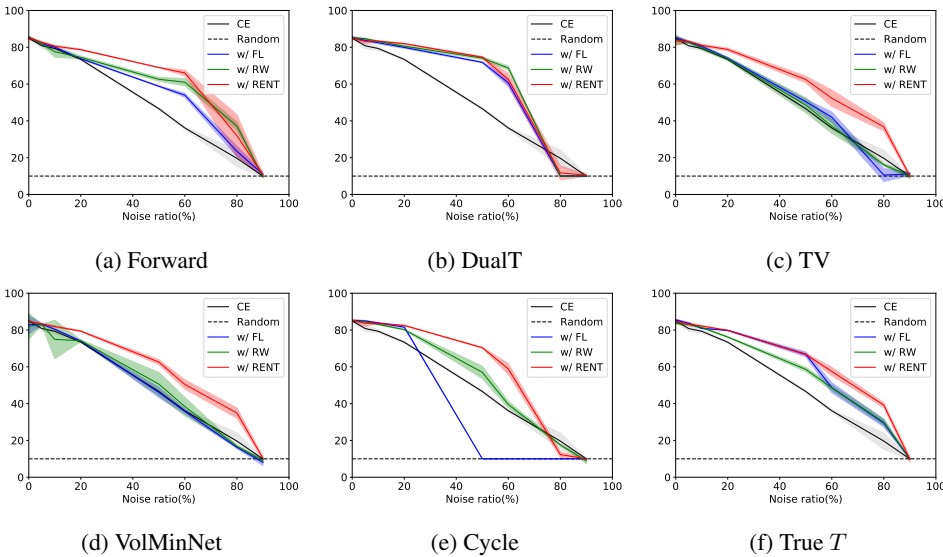

(a) Forward      (b) DualT      (c) TV

(d) VolMinNet      (e) Cycle      (f) True $T$

Figure 11: Performance comparison when using different $T$ utilization methods over several $T$ estimation methods. Subscripts of each figure represents $T$ estimation baselines and colored regions mean standard deviation. X-axis and Y-axis of each figure represents noisy label ratio and the test accuracy, respectively. CE means training with Cross Entropy loss.

Experimental results are based on CIFAR10 datasets with symmetric noise, and replicated over 5 times. Please note that when the noise ratio becomes 90%, it means nothing but random label status, so it is natural the test accuracy is equal to 10%.

**True $T$ is NOT the best?** One of the interesting findings we can see in Table 1 is that the model performance is not the best when it is trained with the true transition matrix (denoted as True $T$).

To check what happens, we first report the performances of RENT with several T estimation methods over the differences between the resulting estimated T and true T. Furthermore, to check any performance pattern over T estimation gap is unique for RENT or not, we also report results for the performances of Forward loss as T utilization (the conventional).

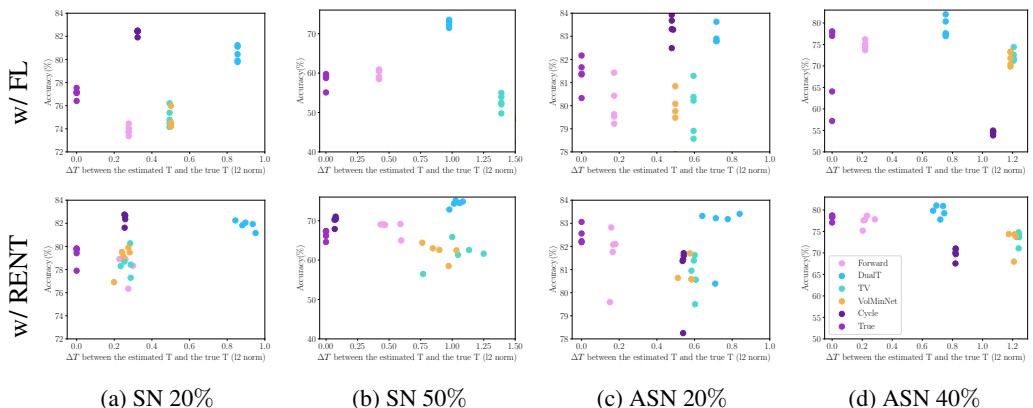

(a) SN 20%          (b) SN 50%          (c) ASN 20%          (d) ASN 40%

Figure 12: Performances with several transition matrix estimation methods over $T$ estimation gap ($\Delta T$, calculated as l2 norm). Sub captions mean noise settings and each color represents different T estimation methods. Figures on upper lines are performances of Forward (the conventional utilization), and that on lower lines are performances of RENT as T utilization. X axes mean T estimation error (calculated as l2 norm) and y axes mean the respective performances. Dots mean different seeds. We do not report Cycle SN 50 (%) cases in the figure because its test accuracy is 10 (%).

Table 4: Pearson Correlation coefficient with regard to $\Delta T$ and model performances.

| | w/ FL | | | | w/ RENT | | | |
|---|---|---|---|---|---|---|---|---|
| | **SN** 20% | **SN** 50% | **ASN** 20% | **ASN** 40% | **SN** 20% | **SN** 50% | **ASN** 20% | **ASN** 40% |
| Pearson Correlation | 0.2234 | 0.1448 | 0.2355 | -0.2601 | 0.4648 | -0.1590 | -0.0644 | -0.5597 |

Figure 12 shows the results. Interestingly, not only can we find out that the performance is not the best when $\Delta T$ is equal to 0, but we can also check that the relation between $\Delta T$ and the model performance is not linear. Check Pearson Correlation also in Table 4.

At first, we conjecture that this failure of finding the correlation between $\Delta T$ and the model performance is due to two reasons: (1) considering TV, VolMinNet and Cycle includes regularization terms for updating a classifier, those terms may have affected the classifier training process and (2) For TV, VolMinNet and Cycle, the transition matrix changes during training procedure, so the transition matrix estimation gap would have reduced while training (according to their original paper, the estimation error seems to decrease with training process). Therefore, the transition matrix gap of those algorithms would have been larger than the reported transition matrix estimation gap.

To analyze the impact of T estimation error to the model performance, comparing performances under same risk function (including no regularization terms) and other learning procedures is required. Therefore, we subtracted $\epsilon_T$ to diagonal terms and added $\epsilon_T/$(the number of classes-1) to others. Figure 13 shows the model performance over $\Delta T$.

Figure 13 shows the performance over $\Delta T$. For SN 20 (%), we set $\epsilon_T$ =[0.0, 0.05, 0.1, 0.2, 0.3, 0.4, 0.5, 0.6] and for SN 50 (%), we set $\epsilon_T$ =[0.0, 0.05, 0.1, 0.2, 0.3], since 0.7 and 0.4 would mean same as total random label flipping respectively. Lines represent performances with arbitrary corrupted transition matrix and small dots is mean performance of each $\epsilon_T$. It again shows that the performance is not the best when the transition matrix estimation error is 0.

Considering Forward, DualT and True, their performances are within the colored region, possibly supporting the explainability of this arbitrary transition matrix corruption experiment.

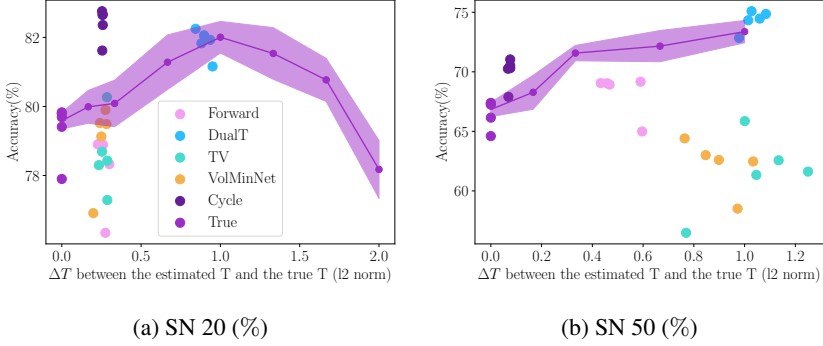

(a) SN 20 (%)                                    (b) SN 50 (%)

Figure 13: Performance over $\Delta T$. Dots are same as 12. Colored regions are standard deviation over 5 seeds. For SN 20 (%), we set $\epsilon_T = [0, 0.05, 0.1, 0.2, 0.3, 0.4, 0.5, 0.6]$ and for SN 50 (%), we set $\epsilon_T = [0, 0.05, 0.1, 0.2, 0.3]$, since 0.7 and 0.4 would mean same as total random label flipping respectively.

**Performance with Lin et al. (2022)**   We compare the performance of RENT and Lin et al. (2022), whose method name is VRNL. We also report the performances with Forward utilization (w/ FL) again as baselines, following VRNL.

Table 5: Test accuracies on CIFAR10 and CIFAR100 with various label noise settings including VRNL. $-$ represents the training failure case. **Bold** is the best accuracy for each setting.

| Base | Risk | CIFAR10 | | | | CIFAR100 | | | |
|---|---|---|---|---|---|---|---|---|---|
| | | SN 20% | SN 50% | ASN 20% | ASN 40% | SN 20% | SN 50% | ASN 20% | ASN 40% |
| Forward | w/ FL | 73.8±0.3 | 58.8±0.3 | 79.2±0.6 | 74.2±0.5 | 30.7±2.8 | 15.5±0.4 | 34.2±1.2 | 25.8±1.4 |
| | w/ VRNL | 76.9±0.4 | 64.8±2.3 | 54.2±9.2 | 59.3±12.7 | 34.1±3.4 | 18.4±2.8 | 35.5±2.4 | 27.2±1.6 |
| | **w/ RENT** | **78.7**±0.3 | **69.0**±0.1 | **82.0**±0.5 | **77.8**±0.5 | **38.9**±1.2 | **28.9**±1.1 | **38.4**±0.7 | **30.4**±0.3 |
| DualT | w/ FL | 79.9±0.5 | 71.8±0.3 | 82.9±0.2 | 77.7±0.6 | 35.2±0.4 | 23.4±1.0 | 38.3±0.4 | 28.4±2.6 |
| | w/ VRNL | 81.2±0.3 | 73.7±0.8 | **83.5**±0.1 | 78.1±2.2 | 38.2±1.4 | 25.2±3.7 | **40.0**±1.2 | **34.4**±1.3 |
| | **w/ RENT** | **82.0**±0.2 | **74.6**±0.4 | 83.3±0.1 | **80.0**±0.9 | **39.8**±0.9 | **27.1**±1.9 | 39.8±0.7 | 34.0±0.4 |
| TV | w/ FL | 74.0±0.5 | 50.4±0.6 | 78.1±1.3 | 71.6±0.3 | **34.5**±1.4 | 21.0±1.4 | 33.9±3.6 | **28.7**±0.8 |
| | w/ VRNL | 76.0±0.6 | 51.3±0.7 | 78.8±0.3 | 58.5±9.2 | 28.5±1.2 | **22.9**±2.8 | 29.9±1.0 | 26.4±2.3 |
| | **w/ RENT** | **78.8**±0.8 | **62.5**±1.8 | **81.0**±0.4 | **74.0**±0.5 | 34.0±0.9 | 20.0±0.6 | **34.0**±0.2 | 25.5±0.4 |
| VolMinNet | w/ FL | 74.1±0.2 | 46.1±2.7 | 78.8±0.5 | 69.5±0.3 | 29.1±1.5 | 25.4±0.8 | 22.6±1.3 | 14.0±0.9 |
| | w/ VRNL | 76.3±0.9 | 50.3±1.4 | 72.3±9.0 | 67.4±5.3 | 28.1±0.6 | 26.6±2.2 | 19.5±3.7 | 14.7±1.4 |
| | **w/ RENT** | **79.4**±0.3 | **62.6**±1.3 | **80.8**±0.5 | **74.0**±0.4 | **35.8**±0.9 | **29.3**±0.5 | **36.1**±0.7 | **31.0**±0.8 |
| Cycle | w/ FL | 81.6±0.5 | $-$ | 82.8±0.4 | 54.3±0.3 | 39.9±2.8 | $-$ | 39.4±0.2 | 31.3±1.2 |
| | w/ VRNL | 82.4±0.6 | $-$ | **83.0**±0.4 | 54.3±0.4 | **41.9**±2.1 | $-$ | **42.1**±1.6 | **32.5**±1.2 |
| | **w/ RENT** | **82.5**±0.2 | **70.4**±0.3 | 81.5±0.1 | **70.2**±0.7 | 40.7±0.4 | **32.4**±0.4 | 40.7±0.7 | 32.2±0.6 |
| True $T$ | w/ FL | 76.7±0.2 | 57.4±1.3 | 75.0±11.9 | 70.7±8.6 | 34.3±0.5 | 22.0±1.5 | **35.8**±0.5 | **31.9**±1.0 |
| | w/ VRNL | 79.3±0.6 | 63.8±1.3 | 49.2±4.4 | 47.7±6.2 | **36.6**±1.1 | **25.5**±0.7 | 26.2±3.3 | 22.5±5.7 |
| | **w/ RENT** | **79.8**±0.2 | **66.8**±0.6 | **82.4**±0.4 | **78.4**±0.3 | 36.1±1.1 | 24.0±0.3 | 34.4±0.9 | 27.2±0.6 |

One step further from the original paper, we also report experimental results with DualT (Yao et al., 2020), TV (Zhang et al., 2021b), Cycle (Cheng et al., 2022) and True $T$, since it can be applied to various transition matrix estimation methods orthogonally. We reported these performances following **Work with VolMinNet** in **Practical Implementation** part of Lin et al. (2022). In other words, we added the variance of the risk function (not including another regularization terms) for implementing VRNL with DualT, TV, Cycle and True $T$. Performances in table 5 consistently shows VRNL improves the original $T$ estimation methods, and RENT tends to show better performances than VRNL. We demonstrate that RENT can increase the variance of the risk function enough without the need to choose the hyper-parameter for variance regularization term (denoted as $\alpha$ in Lin et al. (2022)).

For the hyper-parameter $\alpha$, we followed settings from their paper.

Table 6: Performance comparison with regard to variance of the risk function over various $T$ estimation. **Bold** means the better accuracy between VRNL and RENT for each data setting and $T$ estimation. If the performance of the integrated method is better than the original both methods, we underline the performances.

| Base | Regularizer | CIFAR10 SN 20% | CIFAR10 SN 50% | CIFAR10 ASN 20% | CIFAR10 ASN 40% | CIFAR100 SN 20% | CIFAR100 SN 50% | CIFAR100 ASN 20% | CIFAR100 ASN 40% |
|---|---|---|---|---|---|---|---|---|---|
| Forward | VRNL | $76.9_{\pm0.4}$ | $64.8_{\pm2.3}$ | $54.2_{\pm9.2}$ | $59.3_{\pm12.7}$ | $34.1_{\pm3.4}$ | $18.4_{\pm2.8}$ | $35.5_{\pm2.4}$ | $27.2_{\pm1.6}$ |
| | RENT | $\mathbf{78.7}_{\pm0.3}$ | $\mathbf{69.0}_{\pm0.1}$ | $\mathbf{82.0}_{\pm0.5}$ | $\mathbf{77.8}_{\pm0.5}$ | $\mathbf{38.9}_{\pm1.2}$ | $\mathbf{28.9}_{\pm1.1}$ | $\mathbf{38.4}_{\pm0.7}$ | $\mathbf{30.4}_{\pm0.3}$ |
| | 0.0001 | $78.0_{\pm0.8}$ | $68.3_{\pm1.5}$ | $81.0_{\pm1.0}$ | $76.8_{\pm1.2}$ | $36.9_{\pm1.6}$ | $25.4_{\pm2.1}$ | $38.5_{\pm0.7}$ | $29.9_{\pm1.5}$ |
| | 0.001 | $77.9_{\pm1.3}$ | $68.5_{\pm0.3}$ | $81.0_{\pm0.5}$ | $72.5_{\pm8.9}$ | $38.0_{\pm1.5}$ | $25.8_{\pm1.4}$ | $\underline{38.9}_{\pm0.1}$ | $28.2_{\pm4.0}$ |
| | 0.01 | $77.5_{\pm0.9}$ | $68.9_{\pm0.7}$ | $80.9_{\pm1.1}$ | $76.8_{\pm1.0}$ | $37.6_{\pm0.5}$ | $27.1_{\pm2.0}$ | $37.9_{\pm1.4}$ | $30.3_{\pm2.0}$ |
| | 0.05 | $\underline{78.9}_{\pm0.9}$ | $\underline{71.1}_{\pm0.4}$ | $52.7_{\pm34.9}$ | $52.0_{\pm32.9}$ | $37.7_{\pm1.7}$ | $26.9_{\pm2.9}$ | $38.5_{\pm0.6}$ | $\underline{30.8}_{\pm1.5}$ |
| DualT | VRNL | $81.2_{\pm0.3}$ | $73.7_{\pm0.8}$ | $\mathbf{83.5}_{\pm0.1}$ | $78.1_{\pm2.2}$ | $38.2_{\pm1.4}$ | $25.2_{\pm3.7}$ | $\mathbf{40.0}_{\pm1.2}$ | $\mathbf{34.4}_{\pm1.3}$ |
| | RENT | $\mathbf{82.0}_{\pm0.2}$ | $\mathbf{74.6}_{\pm0.4}$ | $83.3_{\pm0.1}$ | $\mathbf{80.0}_{\pm0.9}$ | $\mathbf{39.8}_{\pm0.9}$ | $\mathbf{27.1}_{\pm1.9}$ | $39.8_{\pm0.7}$ | $34.0_{\pm0.4}$ |
| | 0.0001 | $81.6_{\pm0.4}$ | $74.2_{\pm0.7}$ | $82.9_{\pm0.5}$ | $79.4_{\pm0.3}$ | $39.0_{\pm0.4}$ | $26.7_{\pm4.8}$ | $38.5_{\pm0.6}$ | $33.5_{\pm0.7}$ |
| | 0.001 | $81.4_{\pm0.5}$ | $74.0_{\pm1.2}$ | $\underline{83.5}_{\pm0.3}$ | $79.3_{\pm1.3}$ | $38.5_{\pm2.1}$ | $26.7_{\pm5.3}$ | $\underline{40.1}_{\pm0.4}$ | $32.7_{\pm0.9}$ |
| | 0.01 | $81.1_{\pm0.5}$ | $73.4_{\pm0.7}$ | $82.8_{\pm0.3}$ | $78.7_{\pm0.8}$ | $39.3_{\pm1.2}$ | $26.2_{\pm6.7}$ | $38.7_{\pm0.8}$ | $33.0_{\pm1.4}$ |
| | 0.05 | $81.8_{\pm0.5}$ | $71.8_{\pm1.5}$ | $82.6_{\pm0.7}$ | $\underline{80.3}_{\pm0.4}$ | $\underline{40.0}_{\pm1.2}$ | $27.0_{\pm2.8}$ | $37.1_{\pm1.1}$ | $33.4_{\pm1.4}$ |
| TV | VRNL | $76.0_{\pm0.6}$ | $51.3_{\pm0.7}$ | $78.8_{\pm0.3}$ | $58.5_{\pm9.2}$ | $28.5_{\pm1.2}$ | $\mathbf{22.9}_{\pm2.8}$ | $29.9_{\pm1.0}$ | $\mathbf{26.4}_{\pm2.3}$ |
| | RENT | $\mathbf{78.8}_{\pm0.8}$ | $\mathbf{62.5}_{\pm1.8}$ | $\mathbf{81.0}_{\pm0.4}$ | $\mathbf{74.0}_{\pm0.5}$ | $\mathbf{34.0}_{\pm0.9}$ | $20.0_{\pm0.6}$ | $\mathbf{34.0}_{\pm0.2}$ | $25.5_{\pm0.4}$ |
| | 0.0001 | $77.7_{\pm0.3}$ | $61.5_{\pm4.2}$ | $78.2_{\pm4.3}$ | $\underline{74.3}_{\pm0.8}$ | $33.2_{\pm2.2}$ | $19.7_{\pm0.6}$ | $\underline{34.8}_{\pm0.5}$ | $25.4_{\pm1.4}$ |
| | 0.001 | $78.2_{\pm0.7}$ | $60.4_{\pm1.5}$ | $80.0_{\pm0.8}$ | $73.1_{\pm1.1}$ | $31.0_{\pm4.0}$ | $20.7_{\pm0.5}$ | $33.5_{\pm1.6}$ | $\underline{26.5}_{\pm0.5}$ |
| | 0.01 | $78.2_{\pm1.1}$ | $62.6_{\pm2.8}$ | $81.1_{\pm0.9}$ | $74.1_{\pm1.1}$ | $32.8_{\pm2.9}$ | $20.0_{\pm0.7}$ | $33.4_{\pm1.0}$ | $25.6_{\pm0.9}$ |
| | 0.05 | $\underline{80.3}_{\pm1.0}$ | $\underline{68.8}_{\pm1.4}$ | $80.7_{\pm1.2}$ | $73.7_{\pm0.9}$ | $\underline{35.2}_{\pm1.8}$ | $21.6_{\pm0.4}$ | $34.0_{\pm2.3}$ | $23.2_{\pm4.4}$ |
| VolMinNet | VRNL | $76.3_{\pm0.9}$ | $50.3_{\pm1.4}$ | $72.3_{\pm9.0}$ | $67.4_{\pm5.3}$ | $28.1_{\pm0.6}$ | $26.6_{\pm2.2}$ | $19.5_{\pm3.7}$ | $14.7_{\pm1.4}$ |
| | RENT | $\mathbf{79.4}_{\pm0.3}$ | $\mathbf{62.6}_{\pm1.3}$ | $\mathbf{80.8}_{\pm0.5}$ | $\mathbf{74.0}_{\pm0.4}$ | $\mathbf{35.8}_{\pm0.9}$ | $\mathbf{29.3}_{\pm0.5}$ | $\mathbf{36.1}_{\pm0.7}$ | $\mathbf{31.0}_{\pm0.8}$ |
| | 0.0001 | $78.1_{\pm1.0}$ | $60.4_{\pm2.3}$ | $80.7_{\pm0.5}$ | $72.0_{\pm1.0}$ | $36.3_{\pm0.8}$ | $27.8_{\pm0.7}$ | $32.1_{\pm1.3}$ | $29.1_{\pm1.2}$ |
| | 0.001 | $77.8_{\pm0.7}$ | $62.6_{\pm3.2}$ | $78.4_{\pm4.5}$ | $72.2_{\pm1.3}$ | $31.8_{\pm4.0}$ | $28.2_{\pm1.7}$ | $35.0_{\pm2.5}$ | $29.5_{\pm0.7}$ |
| | 0.01 | $78.3_{\pm0.5}$ | $64.2_{\pm2.6}$ | $80.1_{\pm0.5}$ | $73.0_{\pm2.2}$ | $35.9_{\pm0.7}$ | $27.4_{\pm1.9}$ | $34.7_{\pm1.1}$ | $29.4_{\pm1.9}$ |
| | 0.05 | $\underline{80.1}_{\pm0.7}$ | $\underline{69.7}_{\pm1.0}$ | $80.5_{\pm0.2}$ | $72.3_{\pm1.1}$ | $\underline{37.0}_{\pm1.7}$ | $\underline{30.8}_{\pm0.8}$ | $33.0_{\pm6.9}$ | $\underline{31.3}_{\pm1.9}$ |
| Cycle | VRNL | $82.4_{\pm0.6}$ | – | $\mathbf{83.0}_{\pm0.4}$ | $54.3_{\pm0.4}$ | $\mathbf{41.9}_{\pm2.1}$ | – | $\mathbf{42.1}_{\pm1.6}$ | $\mathbf{32.5}_{\pm1.2}$ |
| | RENT | $\mathbf{82.5}_{\pm0.2}$ | $\mathbf{70.4}_{\pm0.3}$ | $81.5_{\pm0.1}$ | $\mathbf{70.2}_{\pm0.7}$ | $40.7_{\pm0.4}$ | $\mathbf{32.4}_{\pm0.4}$ | $40.7_{\pm0.7}$ | $32.2_{\pm0.6}$ |
| | 0.0001 | $82.1_{\pm0.5}$ | $70.4_{\pm1.0}$ | $80.8_{\pm0.8}$ | $69.1_{\pm1.1}$ | $40.9_{\pm1.2}$ | $31.3_{\pm0.9}$ | $40.5_{\pm1.4}$ | $31.4_{\pm1.2}$ |
| | 0.001 | $81.5_{\pm0.4}$ | $69.8_{\pm1.3}$ | $80.5_{\pm0.9}$ | $68.4_{\pm0.9}$ | $37.9_{\pm5.3}$ | $30.9_{\pm0.7}$ | $35.9_{\pm4.7}$ | $31.9_{\pm0.5}$ |
| | 0.01 | $81.5_{\pm0.7}$ | $70.5_{\pm1.0}$ | $80.6_{\pm0.8}$ | $69.1_{\pm0.4}$ | $42.0_{\pm0.1}$ | $32.1_{\pm0.9}$ | $35.9_{\pm3.6}$ | $30.4_{\pm0.9}$ |
| | 0.05 | $82.1_{\pm0.6}$ | $\underline{71.6}_{\pm1.0}$ | $80.9_{\pm1.0}$ | $69.9_{\pm1.0}$ | $\underline{42.2}_{\pm0.8}$ | $31.4_{\pm0.3}$ | $40.8_{\pm0.4}$ | $29.1_{\pm0.1}$ |
| True $T$ | VRNL | $79.3_{\pm0.6}$ | $63.8_{\pm1.3}$ | $49.2_{\pm4.4}$ | $47.7_{\pm6.2}$ | $\mathbf{36.6}_{\pm1.1}$ | $\mathbf{25.5}_{\pm0.7}$ | $26.2_{\pm3.3}$ | $22.5_{\pm5.7}$ |
| | RENT | $\mathbf{79.8}_{\pm0.2}$ | $\mathbf{66.8}_{\pm0.6}$ | $\mathbf{82.4}_{\pm0.4}$ | $\mathbf{78.4}_{\pm0.3}$ | $36.1_{\pm1.1}$ | $24.0_{\pm0.3}$ | $\mathbf{34.4}_{\pm0.9}$ | $\mathbf{27.2}_{\pm0.6}$ |
| | 0.0001 | $79.1_{\pm0.5}$ | $66.9_{\pm0.7}$ | $82.0_{\pm0.4}$ | $77.2_{\pm0.5}$ | $33.9_{\pm2.2}$ | $23.1_{\pm1.2}$ | $33.8_{\pm0.7}$ | $25.4_{\pm1.2}$ |
| | 0.001 | $79.5_{\pm0.6}$ | $66.6_{\pm1.5}$ | $81.0_{\pm0.6}$ | $77.2_{\pm0.9}$ | $35.4_{\pm1.6}$ | $23.3_{\pm0.4}$ | $32.5_{\pm1.6}$ | $26.3_{\pm0.2}$ |
| | 0.01 | $79.1_{\pm0.9}$ | $67.4_{\pm0.9}$ | $81.5_{\pm0.7}$ | $77.7_{\pm1.2}$ | $32.2_{\pm2.7}$ | $23.8_{\pm0.2}$ | $33.1_{\pm2.4}$ | $25.8_{\pm1.9}$ |
| | 0.05 | $\underline{80.5}_{\pm0.5}$ | $\underline{70.2}_{\pm0.6}$ | – | $9.8_{\pm0.4}$ | $35.0_{\pm1.2}$ | $24.9_{\pm0.9}$ | $2.0_{\pm0.0}$ | $6.6_{\pm3.1}$ |

Then, the next question arises: would integrating RENT and VRNL enhance model performance?, since both methods can be orthogonally applied to the original transition matrix estimation methods. It could show improved performance by increasing the variance of the risk function efficiently, hindering overfitting to noisy labels. It might also lead to worse performance if it prevents training process too much, e.g. when the variance increasing is too much.

Table 6 shows the performance comparison with VRNL, RENT and RENT+variance increasing regularizer. Numbers in front of each row represents the hyperparameter representing the variance ($\alpha$ in Lin et al. (2022)). We report all results with various hyperparameter values to show its sensitivity.

There are cases when the resulting performance is even better than the original both methods, showing its possibility of future development. However, it should be noted that there is no "good-for-all" hyperparameter, indicating its possible limitation of applying it robustly to diverse settings.

Table 7: Test accuracies on CIFAR10 and CIFAR100 with various label noise settings for other baselines. **Bold** is the best accuracy for each setting.

| Terminology | Base | CIFAR10 | | | | CIFAR100 | | | |
|---|---|---|---|---|---|---|---|---|---|
| | | SN 20% | SN 50% | ASN 20% | ASN 40% | SN 20% | SN 50% | ASN 20% | ASN 40% |
| Regularization | ELR | 75.5±0.9 | 47.7±0.5 | 79.5±0.7 | 70.8±1.1 | 34.7±0.8 | 18.4±1.3 | 37.0±0.9 | 27.7±1.2 |
| | SNL ($\sigma = 0.1$) | 71.7±0.3 | 46.9±0.3 | 80.5±1.1 | 72.7±1.2 | 30.2±1.5 | 17.3±0.3 | 33.4±1.5 | 26.5±0.8 |
| Robust loss | SCE | 79.5±0.6 | 54.8±0.5 | 79.5±0.8 | 69.5±0.9 | 34.3±1.2 | 18.3±0.6 | 36.2±1.1 | 27.6±0.5 |
| | APL | 79.3±1.2 | 61.5±3.0 | 76.9±1.7 | 64.1±1.0 | 33.5±2.0 | 18.3±5.5 | 35.9±2.0 | 24.0±4.1 |
| Data cleaning | LRT | 74.9±0.5 | 46.5±1.2 | 77.7±1.9 | 69.2±0.6 | 33.8±1.8 | 20.1±0.4 | 35.9±0.8 | 25.4±3.8 |
| | Coteaching | 78.7±1.4 | **76.4**±3.1 | 81.7±0.5 | 73.9±0.5 | 37.8±4.0 | 12.5±1.3 | 39.7±2.0 | 26.9±3.2 |
| | Jocor | **83.4**±1.6 | 62.9±5.6 | 80.1±1.1 | 65.9±3.4 | 27.5±4.9 | 7.9±1.4 | 34.7±2.5 | 26.9±2.3 |
| | DKNN | 55.5±1.0 | 30.8±0.8 | 62.3±1.1 | 54.1±0.7 | 5.1±0.5 | 2.9±0.4 | 5.3±0.1 | 4.3±0.4 |
| | CORES[2] | 74.7±5.0 | 26.3±4.1 | 71.3±2.3 | 60.7±5.5 | 37.8±2.3 | 6.5±2.1 | 37.8±1.7 | 27.2±1.3 |
| **RENT** (DualT) | | 82.0±0.2 | 74.6±0.4 | **83.3**±0.1 | **80.0**±0.9 | **39.8**±0.9 | **27.1**±1.9 | **39.8**±0.7 | **34.0**±0.4 |

**Performance with other baselines**  In the main paper, we compared our method with Transition matrix based methods to demonstrate the improvement effect of RENT with regard to the transition matrix utilization. In this section, we compare other baselines for the learning with noisy label task itself for a wider range of comparison. Baselines included in this section are:

**ELR Liu et al. (2020)** suggests early learning regularization. Based on the finding that simple patterns are learned fast, they use the output of a learning classifier in early time iterations to regularize overfitting to noisy labels.

**SCE Wang et al. (2019)** suggests a reverse cross entropy as robust loss.

**APL Ma et al. (2020)** theoretically shows any loss with normalization can be made robust to noisy labels and suggests to use two types of robust loss, active loss and passive loss.

**LRT Zheng et al. (2020)** calculate the likelihood ratio between noisy label and the possible pseudo-label, which can be defined as the dimension whose output is the maximum. Based on this criterion, it arbitrary sets a threshold and corrects the label into pseudo label or not.

Please refer to Section 4.4 or Appendix F.4 also for SNL Chen et al. (2020), and Section F.7 for sample selection based methods.

Again, RENT shows best or second best performance.

**More results on real dataset**  Due to the space constraints, we reported the accuracies from only some of the baselines in the main paper. In this part, we present results for more baselines. As shown in table 8, RENT consistently outperforms other $T$ utilization (FL and RW). Similar to the results for CIFAR10 and CIFAR100 presented in Table 1 in the main paper, the performance gap between RENT and previous $T$ utilization widens as the noise ratio increases (please refer to the dataset description section for details on the noise ratio). When estimating $T$ as Cycle, RENT does not always exhibit the best performance. We believe this may be due to the substantial gap between the estimated per-sample weights during training and the true per-sample weights, a hypothesis supported by the consistently poor performance of the RW case. Please also note that although there is the case of PDN CIFAR-10N Aggre when utilizing $T$ as RW yields the best results, the gap between RW and RENT is marginal.

Table 8: Whole test accuracies on CIFAR-10N and Clothing1M. **Bold** is the best.

| Base | Risk | CIFAR-10N | | | | | Clothing1M |
| | | Aggre | Ran1 | Ran2 | Ran3 | Worse | - |
|---|---|---|---|---|---|---|---|
| CE | ✗ | $80.8_{\pm0.4}$ | $75.6_{\pm0.3}$ | $75.3_{\pm0.4}$ | $75.6_{\pm0.6}$ | $60.4_{\pm0.4}$ | $66.9_{\pm0.8}$ |
| Forward | w/ FL | $79.6_{\pm1.8}$ | $76.1_{\pm0.8}$ | $76.4_{\pm0.4}$ | $76.0_{\pm0.2}$ | $64.5_{\pm1.0}$ | $67.1_{\pm0.1}$ |
| | w/ RW | $80.7_{\pm0.5}$ | $75.8_{\pm0.3}$ | $76.0_{\pm0.5}$ | $75.8_{\pm0.6}$ | $63.9_{\pm0.7}$ | $66.8_{\pm1.1}$ |
| | **w/ RENT** | $\mathbf{80.8}_{\pm0.8}$ | $\mathbf{77.7}_{\pm0.4}$ | $\mathbf{77.5}_{\pm0.4}$ | $\mathbf{77.2}_{\pm0.6}$ | $\mathbf{68.0}_{\pm0.9}$ | $\mathbf{68.2}_{\pm0.6}$ |
| DualT | w/ FL | $81.9_{\pm0.2}$ | $79.4_{\pm0.4}$ | $79.3_{\pm1.0}$ | $79.4_{\pm0.4}$ | $72.1_{\pm0.9}$ | $68.2_{\pm1.0}$ |
| | w/ RW | $81.8_{\pm0.4}$ | $79.8_{\pm0.2}$ | $79.4_{\pm0.6}$ | $79.6_{\pm0.4}$ | $71.4_{\pm1.0}$ | $68.5_{\pm0.4}$ |
| | **w/ RENT** | $\mathbf{82.0}_{\pm1.2}$ | $\mathbf{80.5}_{\pm0.5}$ | $\mathbf{80.4}_{\pm0.7}$ | $\mathbf{80.5}_{\pm0.6}$ | $\mathbf{73.5}_{\pm0.7}$ | $\mathbf{69.9}_{\pm0.7}$ |
| TV | w/ FL | $80.5_{\pm0.7}$ | $76.4_{\pm0.4}$ | $76.2_{\pm0.5}$ | $76.1_{\pm0.1}$ | $60.2_{\pm5.2}$ | $66.7_{\pm0.3}$ |
| | w/ RW | $80.7_{\pm0.4}$ | $75.8_{\pm0.6}$ | $75.2_{\pm1.1}$ | $75.4_{\pm1.5}$ | $62.3_{\pm2.9}$ | $67.4_{\pm0.5}$ |
| | **w/ RENT** | $\mathbf{81.0}_{\pm0.4}$ | $\mathbf{77.4}_{\pm0.6}$ | $\mathbf{77.8}_{\pm1.0}$ | $\mathbf{76.7}_{\pm0.4}$ | $\mathbf{66.9}_{\pm3.1}$ | $\mathbf{68.1}_{\pm0.4}$ |
| VolMinNet | w/ FL | $80.9_{\pm0.3}$ | $76.3_{\pm0.5}$ | $75.9_{\pm0.7}$ | $75.9_{\pm0.6}$ | $61.8_{\pm1.3}$ | $65.0_{\pm0.1}$ |
| | w/ RW | $80.7_{\pm0.6}$ | $76.2_{\pm0.5}$ | $75.5_{\pm0.8}$ | $75.5_{\pm0.2}$ | $63.0_{\pm3.2}$ | $66.6_{\pm0.1}$ |
| | **w/ RENT** | $\mathbf{81.3}_{\pm0.4}$ | $\mathbf{77.6}_{\pm1.0}$ | $\mathbf{77.7}_{\pm0.3}$ | $\mathbf{77.2}_{\pm0.7}$ | $\mathbf{66.9}_{\pm0.5}$ | $\mathbf{67.7}_{\pm0.3}$ |
| Cycle | w/ FL | $\mathbf{83.3}_{\pm0.2}$ | $\mathbf{81.0}_{\pm0.4}$ | $\mathbf{81.6}_{\pm0.7}$ | $\mathbf{81.2}_{\pm0.4}$ | $51.6_{\pm1.0}$ | $67.1_{\pm0.2}$ |
| | w/ RW | $81.7_{\pm0.8}$ | $79.1_{\pm0.4}$ | $78.4_{\pm0.4}$ | $78.2_{\pm1.7}$ | $66.0_{\pm0.9}$ | $67.3_{\pm1.2}$ |
| | **w/ RENT** | $82.0_{\pm0.8}$ | $80.0_{\pm0.3}$ | $81.0_{\pm0.8}$ | $80.4_{\pm0.4}$ | $\mathbf{70.5}_{\pm0.4}$ | $\mathbf{68.0}_{\pm0.4}$ |
| PDN | w/ FL | $79.8_{\pm0.6}$ | $74.5_{\pm0.4}$ | $74.5_{\pm0.5}$ | $74.3_{\pm0.3}$ | $57.5_{\pm1.3}$ | $64.9_{\pm0.4}$ |
| | w/ RW | $\mathbf{80.6}_{\pm0.8}$ | $74.9_{\pm0.7}$ | $73.9_{\pm0.7}$ | $74.4_{\pm0.8}$ | $58.7_{\pm0.5}$ | — |
| | **w/ RENT** | $80.2_{\pm0.6}$ | $\mathbf{75.2}_{\pm0.7}$ | $\mathbf{75.0}_{\pm1.1}$ | $\mathbf{75.7}_{\pm0.4}$ | $\mathbf{61.6}_{\pm1.6}$ | $\mathbf{67.2}_{\pm0.2}$ |
| BLTM | w/ FL | $\mathbf{81.5}_{\pm0.7}$ | $78.1_{\pm0.3}$ | $77.5_{\pm0.6}$ | $77.8_{\pm0.5}$ | $65.8_{\pm1.0}$ | $67.2_{\pm0.8}$ |
| | w/ RW | $54.0_{\pm33.9}$ | $64.4_{\pm27.0}$ | $50.9_{\pm32.9}$ | $38.0_{\pm32.4}$ | $43.5_{\pm28.1}$ | $67.0_{\pm0.4}$ |
| | **w/ RENT** | $80.8_{\pm2.1}$ | $\mathbf{79.1}_{\pm0.9}$ | $\mathbf{78.9}_{\pm1.1}$ | $\mathbf{79.6}_{\pm0.6}$ | $\mathbf{69.7}_{\pm2.0}$ | $\mathbf{70.0}_{\pm0.4}$ |

**Performance reproduce & comparison with our experiment setting** Table 9 shows [1] the reported performance at the original paper, [2] the reproduced performance using our code, and [3] the model performance under our experiment settings for each $T$ estimation methods. Comparing three rows for each $T$ estimation method, note that (1) we reproduced the original performance enough and (2) the model performance drops significantly under our experiment setting. We propose again that Forward may not be enough to regularize noisy label memorization.

Please note that there was high variance to the test accuracy with regard to the seed for TV (Zhang et al., 2021b), so better performances could be reproduced if we have explored more times (Currently, we reported best 5 results over 10 times for [2] considering TV). Also since there is no official code for Cycle (Cheng et al., 2022), we reproduced it only with the paper and if some settings are not reported in the paper (e.g. augmentation), we followed settings

Table 9: Test accuracies for CIFAR10. [1] means the reported performance at the original paper. [2] means the reproduced performance using our code. [3] is the model performance under our experiment settings (same as the performance reported in the main paper as Forward (FL)). − is not-reported or training failure.

| | | CIFAR10 | | | |
| | | SN | | ASN | |
| $T$ estimation | [1] ∼ [3] | 20% | 50% | 20% | 40% |
|---|---|---|---|---|---|
| Forward | [1] | $83.4_{\pm-}$ | − | $87.0_{\pm-}$ | − |
| | [2] | $82.6_{\pm0.3}$ | $67.4_{\pm1.0}$ | $87.9_{\pm0.2}$ | $82.9_{\pm0.4}$ |
| | [3] | $73.8_{\pm0.3}$ | $58.8_{\pm0.3}$ | $79.2_{\pm0.6}$ | $74.2_{\pm0.5}$ |
| DualT | [1] | $78.4_{\pm0.3}$ | $70.0_{\pm0.7}$ | − | − |
| | [2] | $85.6_{\pm0.2}$ | $74.2_{\pm0.1}$ | $87.8_{\pm0.7}$ | $81.9_{\pm0.7}$ |
| | [3] | $79.9_{\pm0.5}$ | $71.8_{\pm0.3}$ | $82.9_{\pm0.2}$ | $77.7_{\pm0.6}$ |
| TV | [1] | − | $82.6_{\pm0.4}$ | − | − |
| | [2] | $87.5_{\pm0.2}$ | $76.6_{\pm0.2}$ | $80.6_{\pm7.4}$ | $75.0_{\pm13.3}$ |
| | [3] | $74.0_{\pm0.5}$ | $50.4_{\pm0.6}$ | $78.1_{\pm1.3}$ | $71.6_{\pm0.3}$ |
| VolMinNet | [1] | $89.6_{\pm0.3}$ | $83.4_{\pm0.3}$ | − | − |
| | [2] | $91.4_{\pm0.1}$ | $79.2_{\pm0.1}$ | $94.9_{\pm0.1}$ | $88.0_{\pm4.5}$ |
| | [3] | $74.1_{\pm0.2}$ | $46.1_{\pm2.7}$ | $78.8_{\pm0.5}$ | $69.5_{\pm0.3}$ |
| Cycle | [1] | $90.4_{\pm0.2}$ | − | $90.6_{\pm0.0}$ | $87.3_{\pm0.0}$ |
| | [2] | $90.4_{\pm0.4}$ | − | $86.5_{\pm0.1}$ | $66.4_{\pm0.4}$ |
| | [3] | $81.6_{\pm0.5}$ | − | $82.8_{\pm0.4}$ | $54.3_{\pm0.3}$ |

from Li et al. (2021). Also note that the resnet structure used in TV (Zhang et al., 2021b) and VolMinNet (Li et al., 2021) did not include maxpooling layer unlike the standard structure (He et al., 2016), and it made the difference in model performance. Although there are cases when excluding the maxpooling layer increased the test accuracy up to 5 %, we report the test accuracy with the maxpooling layer following Yao et al. (2020) and Yao et al. (2021) to show the performance utilizing the original resnet structure.

**RENT utilizes $T$ better robustly over Experimental Settings** We conducted a total of 180 experiments under various settings and this number is from the multiplication of below settings.

- $T$ estimation (6): Forward, DualT, TV, VolMinNet, Cycle, True T
- Optimizer (2): SGD, Adam
- Network Architecture (3): ResNet 18, 34, 50
- Seed (5)

Table 10: Beating number (Total 180 times) and average gap (%) of RENT over Forward loss (FL). Perf. gap is the abbreviation of the performance gap.

| | CIFAR10 | | | | CIFAR100 | | | |
| | SN | | ASN | | SN | | ASN | |
| Metric | 20% | 50% | 20% | 40% | 20% | 50% | 20% | 40% |
|---|---|---|---|---|---|---|---|---|
| Number | 162 (90%) | 172 (96%) | 117 (65%) | 148 (82%) | 143 (80%) | 146 (81%) | 110 (61%) | 101 (56%) |
| Perf. gap | 3.5% | 16.0% | 1.0% | 3.7% | 2.2% | 5.4% | 1.0% | 1.2% |

Table 10 demonstrates the general improvement of RENT over Forward loss for $T$ utilization. Note that the average performance gap between RENT and Forward loss is larger when the noise ratio is higher, indicating the increased difficulty of noisy label distribution matching with higher levels of noise. Another interesting point is the superiority of RENT for CIFAR100 since RENT resamples dataset in a mini-batch, meaning it will be harder to get samples from all classes when the number of class becomes more. Therefore, the gradient from the resampled samples could be biased to the classes of the selected samples. Nevertheless, it still utilizes $T$ better than Forward (FL).

## F.3 MORE RESULTS FOR THE IMPACT OF $\alpha$ TO DWS

We show the additional results including other noisy label setting of CIFAR10 and CIFAR100, with colored-lines in each figure reporting all $T$ estimation methods we experimented. Note that the interval of $\alpha$ value in x-axis are drawn in log-scale, meaning that we assigned more space for smaller $\alpha$ region (Same in the figure 3). For experimental details, we used the same experimental settings that we explained earlier.

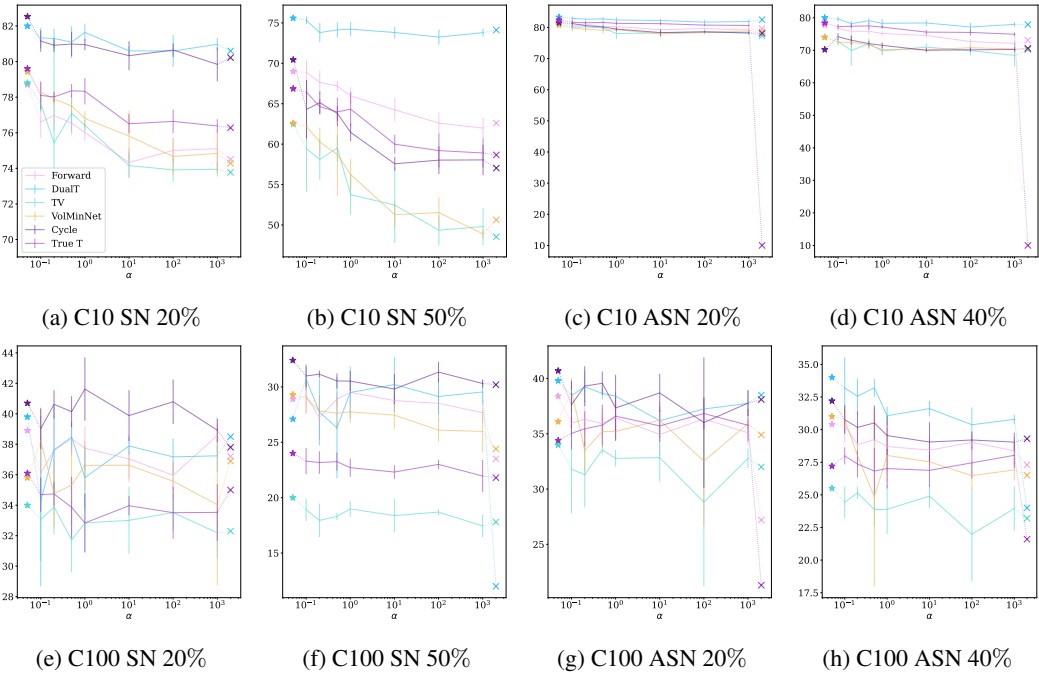

Figure 14: Test accuracy with regard to various $\alpha$ values for CIFAR10 and CIFAR100. Similar to the main paper, **Star** (⋆) and **cross (x)** represents the performances of **RENT** and **RW** respectively.

As shown in the figure 14, performances tend to be higher with smaller $\alpha$ for most of the cases of CIFAR10. However, there are a few cases when the smaller $\alpha$ may not be the answer considering CIFAR100, showing higher lines than the star point (⋆, meaning RENT). Intuitively, it implies that larger variance may not always be the answer and there could exist optimal $\alpha$ for each dataset representing the difficulty and noisy property of the dataset, which could be the direction of further research. Yet, please note that still the Stars are higher than the Crosses (x, meaning RW) except only one case (please refer to table 1 for their exact values).

## F.4 MORE RESULTS FOR NOISE INJECTION IMPACT OF RENT

Table 11: Noise injection impact comparison over various $T$ estimation. **Bold** means best accuracy for each data setting and $T$ estimation. CE, RW and RENT is the same as the one in the main paper. **w.o./**$T$ and **w/**$T$ means without $T$ and with $T$, respectively.

| | | | CIFAR10 | | | | CIFAR100 | | | |
|---|---|---|---|---|---|---|---|---|---|---|
| | | | SN | | ASN | | SN | | ASN | |
| Loss | Base | | 20% | 50% | 20% | 40% | 20% | 50% | 20% | 40% |
| **w.o./** $T$ | SNL | CE (0.0) | 73.4±0.4 | 46.6±0.7 | 78.4±0.2 | 69.7±1.3 | 33.7±1.2 | 18.5±0.7 | **36.9**±1.1 | 27.3±0.4 |
| | | $\sigma=0.1$ | 71.7±0.3 | **46.9**±0.3 | **80.5**±1.1 | 72.7±1.2 | 30.2±1.5 | 17.3±0.3 | 33.4±1.5 | 26.5±0.8 |
| | | $\sigma=0.2$ | 74.2±1.1 | 45.9±1.0 | 79.6±1.3 | **74.8**±0.6 | **35.8**±2.4 | **25.5**±1.0 | 36.5±1.0 | **30.5**±4.3 |
| | | $\sigma=0.3$ | **75.2**±1.4 | 46.8±1.5 | 78.6±1.3 | 72.0±1.5 | 31.7±3.1 | 22.6±2.2 | 31.7±1.1 | 27.1±0.9 |
| **w/** $T$ | Forward | RW (0.0) | 74.5±0.8 | 62.6±1.0 | 79.6±1.1 | 73.1±1.7 | 37.2±2.6 | 23.5±11.3 | 27.2±13.2 | 27.3±1.3 |
| | | $\sigma=0.1$ | 77.8±1.5 | 64.8±6.1 | 74.0±11.5 | 77.5±1.0 | 26.1±3.9 | 1.2±0.1 | 29.9±2.4 | 11.0±8.6 |
| | | $\sigma=0.2$ | 75.9±4.1 | 65.9±3.3 | 76.7±3.5 | 76.3±3.6 | 2.0±1.0 | 1.1±0.2 | 9.1±7.3 | 3.1±2.0 |
| | | $\sigma=0.3$ | 77.5±1.6 | 45.5±12.3 | 76.4±2.9 | 74.5±2.8 | 1.6±0.4 | 1.0±0.1 | 2.3±1.5 | 1.3±0.4 |
| | | **RENT** | **78.7**±0.3 | **69.0**±0.1 | **82.0**±0.5 | **77.8**±0.5 | **38.9**±1.2 | **28.9**±1.1 | **38.4**±0.7 | **30.4**±0.3 |
| | DualT | RW (0.0) | 80.6±0.6 | 74.1±0.7 | 82.5±0.2 | 77.9±0.4 | 38.5±1.0 | 12.0±13.5 | 38.5±1.6 | 24.0±11.6 |
| | | $\sigma=0.1$ | 74.7±3.6 | 48.2±9.7 | 78.4±1.1 | 72.2±4.7 | 18.5±7.2 | 1.9±0.6 | 31.7±4.1 | 11.8±9.1 |
| | | $\sigma=0.2$ | 66.4±9.2 | 31.6±5.0 | 76.6±0.7 | 69.4±5.1 | 2.2±1.3 | 1.4±0.7 | 7.9±4.2 | 3.5±2.2 |
| | | $\sigma=0.3$ | 59.7±11.7 | 26.2±10.8 | 69.6±2.8 | 64.2±2.7 | 1.5±0.7 | 1.2±0.1 | 4.9±3.0 | 1.4±0.6 |
| | | **RENT** | **82.0**±0.2 | **74.6**±0.4 | **83.3**±0.1 | **80.0**±0.9 | **39.8**±0.9 | **27.1**±1.9 | **39.8**±0.7 | **34.0**±0.4 |
| | TV | RW (0.0) | 73.7±0.9 | 48.5±4.1 | 77.3±2.0 | 70.2±1.0 | 32.3±1.0 | 17.8±2.0 | 32.0±1.5 | 23.2±0.9 |
| | | $\sigma=0.1$ | 70.7±10.1 | 49.0±10.6 | 76.2±4.7 | 69.8±3.1 | 1.4±0.3 | 1.8±1.0 | 15.0±11.9 | 5.6±7.9 |
| | | $\sigma=0.2$ | 57.6±16.5 | 15.6±7.8 | 38.6±29.4 | 43.9±19.5 | 1.4±0.2 | 1.1±0.1 | 1.4±0.4 | 1.7±0.9 |
| | | $\sigma=0.3$ | 30.3±23.3 | 15.6±4.9 | 30.5±22.6 | 19.5±11.4 | 1.0±0.1 | 1.1±0.2 | 1.2±0.3 | 1.2±0.4 |
| | | **RENT** | **78.8**±0.8 | **62.5**±1.8 | **81.0**±0.4 | **74.0**±0.5 | **34.0**±0.9 | **20.0**±0.6 | **34.0**±0.2 | **25.5**±0.4 |
| | VolMinNet | RW (0.0) | 74.2±0.5 | 50.6±6.4 | 78.6±0.5 | 70.4±0.8 | **36.9**±1.2 | 24.4±3.0 | 34.9±1.3 | 26.5±0.9 |
| | | $\sigma=0.1$ | 77.1±2.0 | 56.6±2.0 | 79.0±2.3 | 71.2±2.3 | 1.9±0.7 | 1.2±0.1 | 2.5±0.9 | 2.3±0.8 |
| | | $\sigma=0.2$ | 67.7±7.0 | 25.8±12.1 | 75.3±2.4 | 59.9±3.9 | 1.2±0.1 | 1.1±0.1 | 1.2±0.2 | 1.2±0.3 |
| | | $\sigma=0.3$ | 21.8±10.3 | 15.6±4.2 | 39.9±22.2 | 32.8±18.4 | 1.1±0.3 | 1.2±0.2 | 1.2±0.2 | 1.2±0.2 |
| | | **RENT** | **79.4**±0.3 | **62.6**±1.3 | **80.8**±0.5 | **74.0**±0.4 | 35.8±0.9 | **29.3**±0.5 | **36.1**±0.7 | **31.0**±0.8 |
| | Cycle | RW (0.0) | 80.2±0.2 | 57.0±3.4 | 78.1±0.9 | 70.6±1.1 | 37.8±2.7 | 30.2±0.6 | 38.1±1.6 | 29.3±0.6 |
| | | $\sigma=0.1$ | 77.9±1.3 | **71.1**±2.5 | 75.9±2.7 | **74.4**±1.4 | 7.9±11.1 | 2.5±1.8 | 2.0±1.3 | 5.2±3.3 |
| | | $\sigma=0.2$ | 61.0±23.8 | 64.1±5.6 | 69.7±2.9 | 66.8±1.5 | 1.7±0.3 | 1.3±0.4 | 1.9±0.5 | 2.7±1.4 |
| | | $\sigma=0.3$ | 63.0±2.6 | 61.2±5.4 | 58.4±4.7 | 42.7±11.1 | 1.2±0.2 | 1.4±0.2 | 1.2±0.1 | 1.4±0.2 |
| | | **RENT** | **82.5**±0.2 | 70.4±0.3 | **81.5**±0.1 | 70.2±0.7 | **40.7**±0.4 | **32.4**±0.4 | **40.7**±0.7 | **32.2**±0.6 |
| | True T | RW (0.0) | 76.2±0.3 | 58.6±1.2 | − | − | 35.0±0.8 | 21.8±0.8 | 21.3±16.6 | 21.6±10.4 |
| | | $\sigma=0.1$ | **79.9**±0.8 | **68.8**±0.6 | 81.3±0.6 | 78.0±0.7 | 30.7±3.3 | **24.1**±0.5 | **34.5**±1.9 | 27.1±2.4 |
| | | $\sigma=0.2$ | 76.5±2.1 | 67.3±2.3 | 81.7±0.4 | 77.8±1.1 | 25.7±3.4 | 12.4±8.4 | 32.5±2.7 | **27.2**±2.4 |
| | | $\sigma=0.3$ | 75.7±1.3 | 59.3±11.5 | 78.8±1.5 | 75.5±2.7 | 14.3±4.7 | 1.5±0.4 | 25.9±2.6 | 23.6±2.9 |
| | | **RENT** | 79.8±0.2 | 66.8±0.6 | **82.4**±0.4 | **78.4**±0.3 | **36.1**±1.1 | 24.0±0.3 | 34.4±0.9 | 27.2±0.6 |

Table 11 shows model performance comparison with regard to CIFAR10 and CIFAR100 under various label noise settings. In the table, we report the results under the whole $T$ estimation baselines that we experimented in the table 1.

Interestingly, label perturbation to either (1) the naive cross entropy loss (SNL row in the table) or (2) the reweighting loss with true $T$ (True $T$ row in the table) generally improves the model performance compared to the baselines, which refer to the model performances achieved when trained solely with the basic loss itself without any random label noise. On the other hand, there are cases where the model performance become worse when the random label noise injection technique is utilized with the reweighting loss from the estimated $T$ under the same $\sigma$. Also note that integrating the label perturbation technique directly to various $T$ estimation methods can easily lead to training failure, especially for CIFAR100. These failure could be attributed to both the inaccurate objective function resulting from the inaccurate per-sample weights, which are calculated based on the estimated $T$, and the injected label noise. These could also imply that application of the label noise injection technique to the reweighting loss might be more sensitive to the parameter $\sigma$ compared to its original application, which is the application to the naive cross entropy loss. This sensitivity can be problematic in term of its applicability. In contrast, RENT tends to exhibit better performance than RW, further highlighting its robust adaptability to diverse situations.

Figure 15 shows additional results for CIFAR10 over figure 4. Similar to figure 4, RENT consistently outperforms SNL and RW+$\epsilon$ also under ASN settings. In the figure, we removed $\sigma = 0.0$ case, which is RW, for ASN plots because the training failed considering both cases. Also note that RW+$\epsilon$ is highly sensitive to the value of $\sigma$ and tuning the parameter is required for adequate model performances.

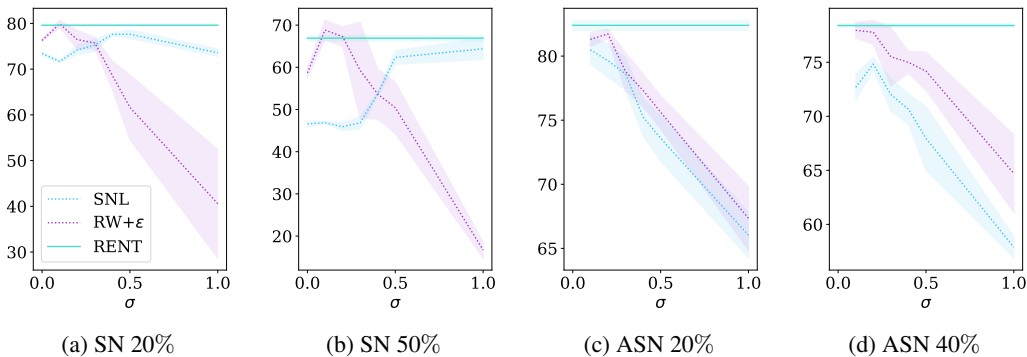

(a) SN 20%    (b) SN 50%    (c) ASN 20%    (d) ASN 40%

Figure 15: Test accuracies over various $\sigma$ for CIFAR10. Same with figure 4, RW+$\epsilon$ denotes the integration of RW and the label noise perturbation technique.

We present the model performances using different $\sigma \in [0.0, 0.1, 0.2, 0.3, 0.4, 0.5, 1.0]$ for both figure 4 and figure 15. We utilized the true transition matrix $T$ for getting the model performances in the figures, because we wanted to ensure that $T$ estimation gap does not have impact on the performance gap between SNL and others. All other experimental details remain consistent with the settings described in the earlier section.

## F.5 MORE RESULTS FOR OUTCOME ANALYSES

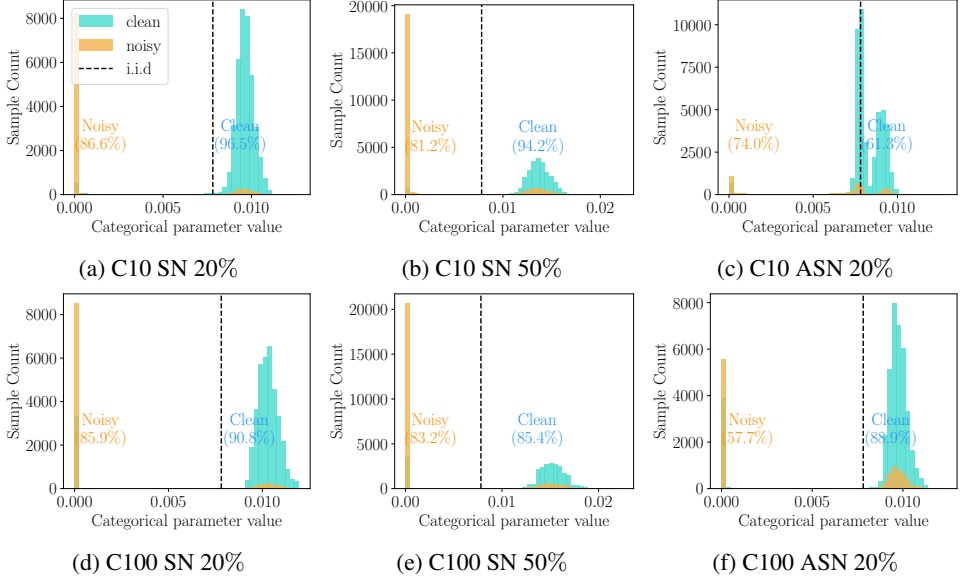

(a) C10 SN 20%    (b) C10 SN 50%    (c) C10 ASN 20%

(d) C100 SN 20%    (e) C100 SN 50%    (f) C100 ASN 20%

Figure 16: Histogram of $w_i$ of RENT on CIFAR10 and CIFAR100. Same as figure 5, we utilized Cycle (Cheng et al., 2022) as $T$ estimation method for this plot. Blue and orange represents the number of samples with clean and noisy labels, respectively. Vertical dotted line denotes $1/B$. Samples with $w_i \le 1/B$ would likely be less sampled than i.i.d..

$w_i$ **value** Figure 16 shows the additional results of the figure 5 over CIFAR10 and CIFAR100. Percentages reported in both figures are calculated as the ratio between the number of samples

whose $w_i$ is within each range and the total number of samples whose labels as data instances ($\tilde{y}_i$) are same with (clean) or different from (noisy) the original class labels ($\times 100\%$). Similar to figure 5, it consistently shows high ratio of clean samples in oversampling region and high ratio of noisy samples near zero.

With the results reported in Figure 5 and figure 16, $w_i$ value can be divided easily with threshold, $1/B$. However, Table 12 shows it is not true.

Table 12: Test accuracies on CIFAR10 and CIFAR100 with various label noise settings comparing **RENT** vs. thresholding to $w_i$+random sampling. **Bold** is the best accuracy for each setting.

| Base | Criterion | CIFAR10 | | | | CIFAR100 | | | |
|---|---|---|---|---|---|---|---|---|---|
| | | **SN** 20% | **SN** 50% | **ASN** 20% | **ASN** 40% | **SN** 20% | **SN** 50% | **ASN** 20% | **ASN** 40% |
| Forward | $w$+random | 74.0±0.8 | 45.7±1.6 | 80.0±0.8 | 71.3±2.0 | 30.8±1.8 | 16.1±1.1 | 30.6±2.2 | 24.6±0.5 |
| | **w/ RENT** | **78.7**±0.3 | **69.0**±0.1 | **82.0**±0.5 | **77.8**±0.5 | **38.9**±1.2 | **28.9**±1.1 | **38.4**±0.7 | **30.4**±0.3 |
| DualT | $w$+random | 74.8±0.6 | 48.2±1.2 | 80.1±0.6 | 72.4±0.9 | 31.9±0.9 | 16.5±1.1 | 34.7±1.8 | 25.1±0.7 |
| | **w/ RENT** | **82.0**±0.2 | **74.6**±0.4 | **83.3**±0.1 | **80.0**±0.9 | **39.8**±0.9 | **27.1**±1.9 | **39.8**±0.7 | **34.0**±0.4 |
| TV | $w$+random | 70.7±1.8 | 45.5±1.6 | 78.7±0.9 | 71.9±1.7 | 28.2±2.8 | 16.1±0.9 | 28.8±2.0 | 21.6±1.9 |
| | **w/ RENT** | **78.8**±0.8 | **62.5**±1.8 | **81.0**±0.4 | **74.0**±0.5 | **34.0**±0.9 | **20.0**±0.6 | **34.0**±0.2 | **25.5**±0.4 |
| VolMinNet | $w$+random | 71.4±0.4 | 45.3±2.3 | 76.8±2.9 | 71.7±1.7 | 28.7±1.5 | 17.2±1.5 | 30.4±2.0 | 23.9±2.2 |
| | **w/ RENT** | **79.4**±0.3 | **62.6**±1.3 | **80.8**±0.5 | **74.0**±0.4 | **35.8**±0.9 | **29.3**±0.5 | **36.1**±0.7 | **31.0**±0.8 |
| Cycle | $w$+random | 70.8±12.2 | 44.7±0.8 | 78.1±0.7 | 69.2±1.2 | 31.5±0.8 | 16.3±1.4 | 33.2±1.6 | 24.5±2.3 |
| | **w/ RENT** | **82.5**±0.2 | **70.4**±0.3 | **81.5**±0.1 | **70.2**±0.7 | **40.7**±0.4 | **32.4**±0.4 | **40.7**±0.7 | **32.2**±0.6 |

Here, $w$+random means when if we just set a threshold for $w$ and randomly sampling again from the selected samples. This gap happens because during training, especially in the earlier learning iterations, $w$ of clean samples and noisy samples may be more mixed, since the model parameter is yet more similar to the random assignment. Therefore, sorting with an arbitrary threshold may be riskier. However, Dirichlet-based resampling and RENT may be safer to this problem since the sampling probability of the samples below threshold is not 0.This would result in the overall performance differences.

Also, choosing a good threshold may be difficult and some training iteration and noise condition adaptive strategy could be needed. Figure 17 shows this need.

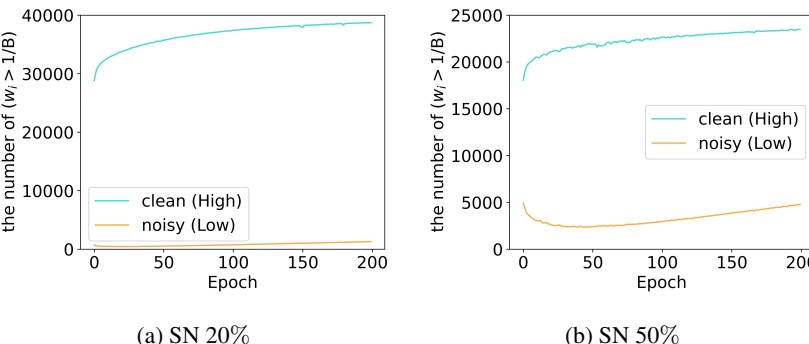

(a) SN 20%            (b) SN 50%

Figure 17: The number of samples whose $w_i$ value is larger than $1/B$ over time iterations. We visualize clean labels and noisy labels as blue and orange. Dataset is CIFAR10. Note that 40,000 and 25,000 samples are total number of clean labels for 20% and 50%, respectively.

It shows the number of samples whose $w_i$ value is larger than $1/B$ over time iterations, with clean labels and noisy labels, respectively, while training the classifier with RENT. Result shows that clean label samples whose $w_i$ is larger than $1/B$ is not as many as the final iteration (which is natural), which underlines the training process adaptive strategy for threshold value.

**Confidence of wrong labelled samples** We show additional results over figure 6 in figure 18 for CIFAR10 and CIFAR100. The model trained with RENT consistently reports lower confidence

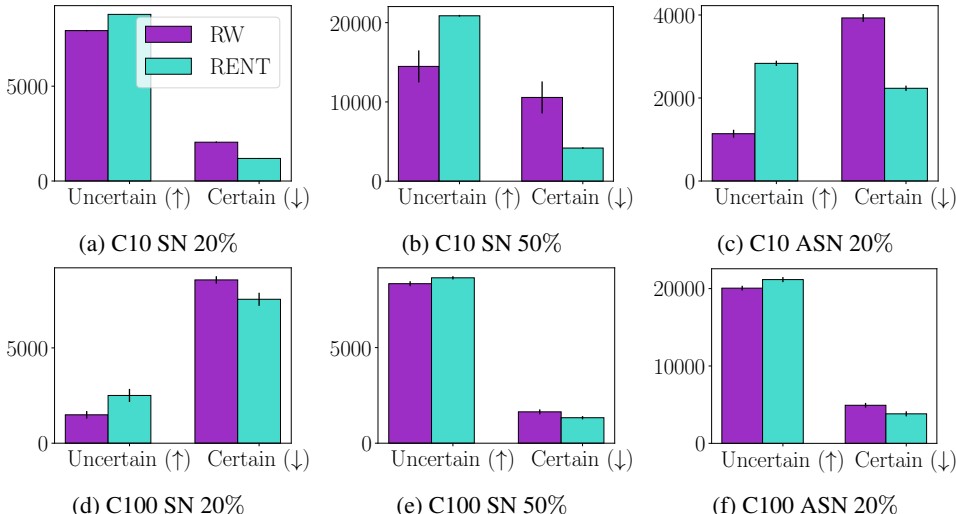

Figure 18: Training data with incorrect labels divided by the confidence (threshold=0.5). Cycle (Cheng et al., 2022) for $T$ estimation, on CIFAR10 and CIFAR100.

values for noisy-labelled samples across different datasets compared to RW. This observation again supports less memorization of RENT over RW.

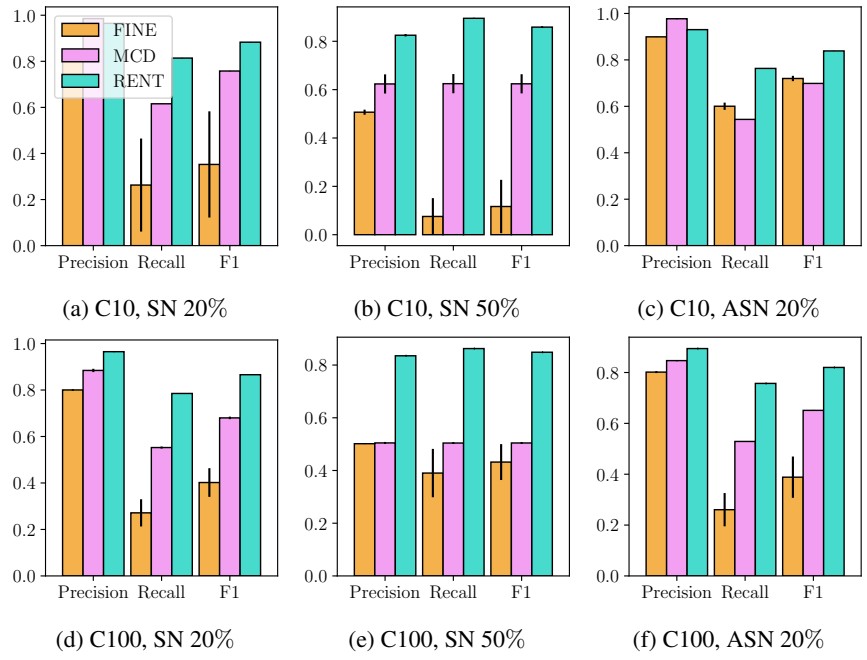

Figure 19: Evaluation based on the selected metrics for CIFAR10 and CIFAR100. We adopted Cycle (Cheng et al., 2022) for $T$ estimation.

**Resampled dataset quality** Here, we show additional results over figure 7 in figure 19 for CIFAR10 and CIFAR100. We show that RENT consistently surpasses the baselines in F1 score, implying the good quality of the resampled dataset. We provide details about the baselines that were used in our experiments as follows.

**MCD** Lee et al. (2019) assumes features of samples with noisy label will be far away from that of samples with clean label. In this sense, they assume feature distribution as normal distribution and

filter out noisy-labelled data using the Mahalanobis distance as the metric to discriminate whether a sample is clean or noisy. Following the original paper, we select to use $50\%$ samples of the data with smallest distances.

**FINE** Kim et al. (2021) filters out noisy-labelled samples by eigenvector. Following the original paper, we use the hyperparameter of clean probability of GMM as 0.5.

Considering FINE, there are cases when it shows unimaginably bad performance. We conjecture these bad performances can be from two factors. First, please note that there are differences in experimental settings. In the original paper, FINE utilizes ResNet 34 network, without max pooling layer, with SGD optimizer. We utilize ResNet 34 for CIFAR10 and ResNet 50 for CIFAR100 with Adam optimizer. Considering the dataset condition, we did not normalize input data following Zhang et al. (2021b); Li et al. (2021) but FINE did and it would make differences in performance. Second, since there is no reported result on the original paper when applying the FINE algorithm to the model trained with transition matrix based loss functions, comparing this performance to the performance from the original paper is impossible. With lower recall values, we assume eigen decomposition of feature vectors would not discriminate clean samples efficiently.

## F.6  ABLATION OVER THE SAMPLING STRATEGIES

Table 13: Ablation over the sampling strategies.

| | | CIFAR10 | | | CIFAR100 | | |
|---|---|---|---|---|---|---|---|
| **Base** | **Sampling** | **SN 20%** | **SN 50%** | **ASN 20%** | **SN 20%** | **SN 50%** | **ASN 20%** |
| Forward | Batch | $78.7_{\pm0.3}$ | $69.0_{\pm0.1}$ | $82.0_{\pm0.5}$ | $38.9_{\pm1.2}$ | $28.9_{\pm1.1}$ | $38.4_{\pm0.7}$ |
| | Global | $81.8_{\pm0.5}$ | $72.6_{\pm0.9}$ | $83.2_{\pm0.6}$ | $32.3_{\pm2.2}$ | $4.4_{\pm1.9}$ | $33.3_{\pm2.1}$ |
| | Global.C | $81.8_{\pm0.5}$ | $75.9_{\pm0.4}$ | $83.5_{\pm0.3}$ | $37.3_{\pm0.8}$ | $23.3_{\pm2.5}$ | $36.7_{\pm2.7}$ |
| DualT | Batch | $82.0_{\pm0.2}$ | $74.6_{\pm0.4}$ | $83.3_{\pm0.1}$ | $39.8_{\pm0.9}$ | $27.1_{\pm1.9}$ | $39.8_{\pm0.7}$ |
| | Global | $81.7_{\pm0.6}$ | $74.6_{\pm0.7}$ | $83.0_{\pm0.4}$ | $39.1_{\pm1.4}$ | $26.9_{\pm2.3}$ | $36.4_{\pm1.9}$ |
| | Global.C | $81.3_{\pm0.7}$ | $74.1_{\pm0.7}$ | $82.7_{\pm0.6}$ | $38.9_{\pm1.3}$ | $27.3_{\pm3.0}$ | $38.7_{\pm1.6}$ |

RENT utilizes the resampling from the mini-batch as a default setting. However, sampling from the mini-batch could also be changed to the dataset-level. As another ablation study on the sampling strategies of RENT, we conduct sampling from the whole dataset level, which could be divided into two; 1) global sampling from whole dataset, which we denote as Global, and 2) class-wise sampling from whole dataset, which we denote as Global.C. Table 13 shows that each strategy shows robust performances over the different experimental settings.

## F.7  RENT VS. SAMPLE-SELECTION

Apart from transition matrix based methods, sample selection methods dynamically select a subset of training dataset during the classifier training, by filtering out the incorrectly-labelled instances from the noisy dataset Han et al. (2018); Yu et al. (2019); Wei et al. (2020); Bahri et al. (2020); Cheng et al. (2020). Our method, RENT, shares the same property with sample selection methods by recognizing the resampling of RENT as noise-filtering procedure. Therefore, we compare the performances of RENT with Coteaching Han et al. (2018), Jocor Wei et al. (2020), DKNN Bahri et al. (2020) and CORES[2] Cheng et al. (2020). We report the details of the baselines as follows.

**Coteaching** Han et al. (2018) assumes samples with large loss are assumed to be noisy-labelled. However, selecting samples based on the output of the trained classifier may cause the problem of sampling only already-trained samples again. Therefore, it utilizes two networks with same structures from different initialization point. It selects data with small loss from other network and train model parameter with the selected data.

**Jocor** Wei et al. (2020) also considers the loss value as a metric to discriminate noisy-labelled data and they utilize two networks with same structures, but they optimize two network at one time.

**DKNN** Bahri et al. (2020) considers data whose labels are different from the K-Nearest Neighbor algorithm result as noisy-labelled data. We utilize $k = 500$ as reported in the original paper.

**CORES** Cheng et al. (2020) select samples based on its output confidence with regularization term making the model be more confident, since a classifier can be uncertain for clean labels when training with noisy-labelled dataset. We set $\beta = 2$ following the original paper.

For training Coteaching and Jocor, noisy label ratio is required as input information to decide the sample selection portion. However, since we do not know this information, we followed the same way as DualT Yao et al. (2020) for noisy ratio estimation, i.e. get the average of the diagonal term of $T$ as 1-noisy ratio. For $T$ estimation, we utilize the algorithm of Yao et al. (2020).

Table 14: Comparison with Sample selection methods - Clothing1M. We did not report results on DKNN because of too much computation.

| Methods | Coteaching | Jocor | CORES[2] | DKNN | RENT |
|---------|-----------|-------|----------|------|------|
| Accuracy (%) | $67.2_{\pm 0.4}$ | $68.4_{\pm 0.1}$ | $66.4_{\pm 0.7}$ | $-$ | $\mathbf{69.9}_{\pm 0.7}$ |

Table 7 and table 14 compares test accuracies over the baselines and RENT. Our method shows competitive performances over the baselines. It should be noted that Coteaching and Jocor requires the training of two different classifiers, which requires more memory space and computation than other methods.

## F.8    WHEN NOISY DATASET IS CLASS IMBALANCED

Solving class imbalanced dataset with noisy label can be important (Koziarski et al., 2020; Chen et al., 2021; Huang et al., 2022). Therefore, we experimented to show which T utilization would work well under class imbalance. Following studies that solves class imbalanced learning Cao et al. (2019); Zhang et al. (2023), we first make the imbalanced dataset with (the maximum number of samples in class)/(the minimum number of samples in class) is defined as imbalance ratio and between two classes the number of samples in each class should increase in exponential order. Then, we flipped the label to be noisy as we did previously.

Figure 20 shows the performance comparison of FL, FW and RENT with several T estimation baselines. As we can see in the figure, again, RENT shows consistently good result over FL and RW. However, since there is no treatment to solve class imbalance issue in all of FL, RW and RENT, the model performances decrease with higher class imbalance ration. This direction could be an interesting future study.

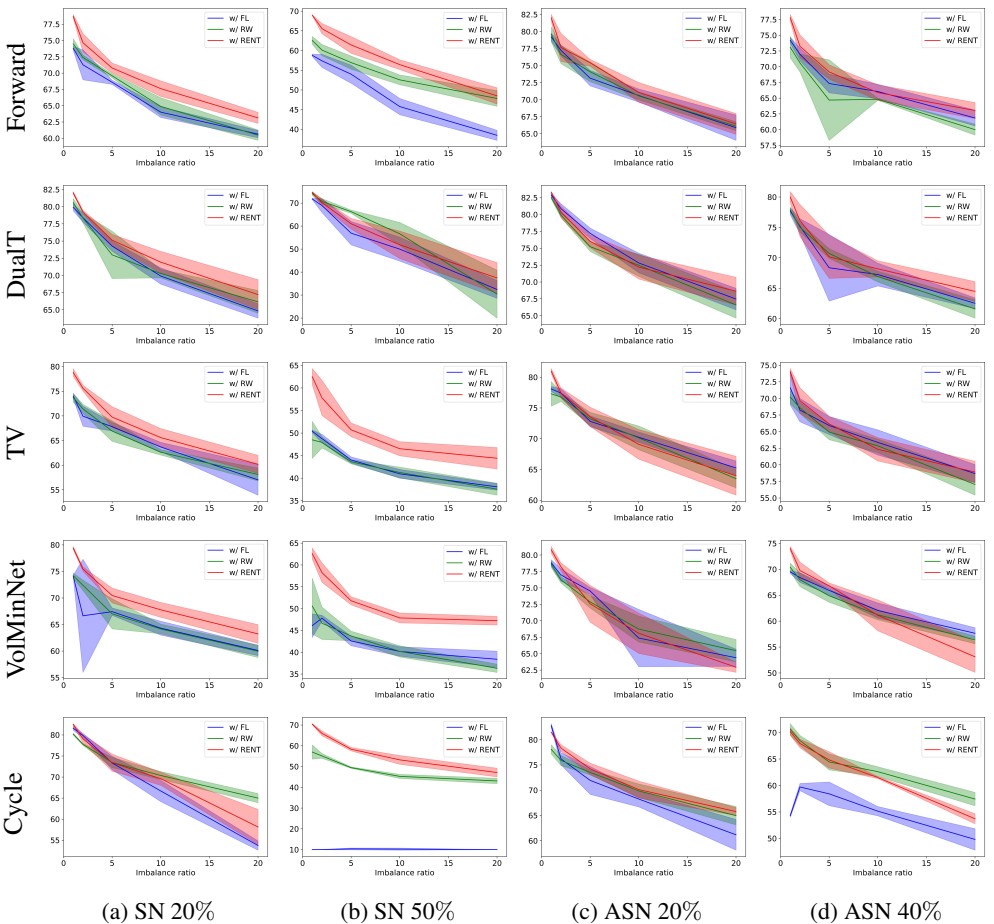

(a) SN 20%   (b) SN 50%   (c) ASN 20%   (d) ASN 40%

Figure 20: Performance comparison of FL, RW and RENT over class imbalance ratio. Subcaptions represent noisy label setting and vertical captions mean the transition matrix estimation methods. RED, GREEN and BLUE represents RENT, RW and FL, respectively.

## F.9   TIME COMPLEXITY

Table 15: Iterations denotes the number of iterations per epoch.

| Dataset | CIFAR10 | CIFAR100 | Clothing1M |
|---|---|---|---|
| Dataset size / Iterations | 50,000 / 391 | 50,000 / 391 | 1,000,000 / 10,000 |
| $t_{\text{total}}$ | 15.62s | 20.44s | 2315.10s |
| $t_{\text{sample}}$ | 0.13s (**0.83**%) | 0.12s (**0.59**%) | 4.02s (**0.17**%) |

To check whether there is increment in the time complexity by RENT, Table 15 presents the wall-clock time in resampling procedure of RENT. $t_{\text{total}}$ is the total wall-clock for a single epoch, and $t_{\text{sample}}$ is the time only for the resampling. Given a large-scale dataset, i.e. Clothing1M, the resampling time becomes ignorable.

