# OpenReview forum: "Dirichlet-based Per-Sample Weighting by Transition Matrix for Noisy Label Learning"
_ICLR.cc/2024/Conference — ICLR 2024 poster_

### Official Review · Reviewer_4M7K · 2023-10-25

**Soundness:** 3 good
**Presentation:** 2 fair
**Contribution:** 3 good
**Rating:** 6
**Confidence:** 4

**Summary:**

Different from previous researches that focus more on how to estimate the transition matrix, which is significant for the risk estimator to achieve the statistical consistency, this paper proposes a new utilization of the transition matrix to deal with noisy labels.

**Strengths:**

1.This paper proposes a new approach, named RENT, a resampling method with Noise Transition Matrix.

2.The proposed approach origins from the analysis of comparing reweighting and resampling under Dirichlet distribution based per-sample Weight Sampling (DWS) framework, which is reasonable.

3.Experimental results show the effectiveness of the approach.

**Weaknesses:**

1.Since the final algorithm chooses resampling, my main concern is the advantage of resampling comparing to reweighting, which should be given more details about and hightlighted in some sections like Introduction.

2.Figure 1 should be polished up to clearly illustrate how to implement the proposed framework. Otherwise, it is difficult to understand the 4-th and 5-th paragraphs of Section Introduction.

3.It is suggested to summarize the contributions at the end of Section Introduction, such as extension of DWS, the analysis of reweighting and resampling, the new resampling method.

4.Some details about the implementation of the resampling should be given in the main body of the paper. For example, the reason why $\tilde{\mu}$ and the parameters of Categorical Distribution are calculated in the way illustrated in Algorithm 1.

**Questions:**

1.As it is mentioned above, why $\tilde{\mu}$ and the parameters of Categorical Distribution are calculated in the way illustrated in Algorithm 1?

---

> ### Author Response · Authors · 2023-11-18
>
> **W1: Advantage of resampling over reweighting**
>
> -	**[The advantage of resampling comparing to reweighting]** Sorry for our possible misunderstanding to the reviewer’s question, and sorry for our not enough explanations that may lead the reviewer to be confused.
>
> -	Actually, the final advantage of resampling over reweighting is simple: good performance when learning with noisy label. We also showed the performance empirically and reported that resampling may memorize noisy labelled samples less than reweighting by visualizing the confidence of wrong labelled samples in Section 4.5. We thought we explained why it works through section 3, but we explain why here and modified the manuscript.
>
> 1. In the previous study[1], the variance increase of the risk function has improved the model performance empirically. Since the variance of risk with resampling is larger than that with reweighting, we can explain the good performance of RENT following the previous study.
>
> 2. Empirically, per-sample weights should be different from the true weights. Nevertheless, per-sample weights from resampled dataset (in this case, it would mean the sampling ratio for each sample) is closer to the true weight than reweighting in mahalanobis distance manner. It means that the gap between the true risk and the empirical risk from per-sample weight estimation gap is smaller when resampling than reweighting.
>
> 3. Resampling can be interpreted as label perturbation. The superiority of resampling over reweighting is supported with several previous studies [2,3] showing that perturbations during training results in better performance for learning with noisy label
>
> -	If it is not the answer that the reviewer wanted, can the reviewer explain in more details what the reviewer wants?
>
> -	**[highlighting in Introduction]** Following the reviewer’s suggestion, we added the summary of what we explained in section 3 in the introduction part by changing 5-th paragraph and adding the contribution summary.
>
> **W2: Figure1**
>
> -	Sorry for your confusion. we will represent the reviewer’s concern and change our figure in the final version. For helping the reviewer’s understanding possibly, we added a figure to clearly illustrate the implementation of RENT in Appendix A currently.
>
> -	We wonder whether we represented the reviewer’s concern adequately. We hope the reviewer responds to us in more details so that we can relieve the reviewer’s concern.
>
>
> **W3: Summarization of contribution at Introduction**
>
> -	We represented the reviewer’s suggestion in the revised manuscript.
>
> **W4&Q1: $\mu$ of algorithm 1**
>
> -	Sorry for your confusion.
> -	Due to the space issue, we updated why $\mu$ is calculated as algorithm 1 in appendix D.2. It may help you how the parameters of the categorical distribution are calculated.
>
> [1] Lin, Y., Yao, Y., Du, Y., Yu, J., Han, B., Gong, M., & Liu, T. (2022). Do We Need to Penalize Variance of Losses for Learning with Label Noise?. arXiv preprint arXiv:2201.12739.
>
> [2] Chen, P., Chen, G., Ye, J., & Heng, P. A. (2020, October). Noise against noise: stochastic label noise helps combat inherent label noise. In International Conference on Learning Representations.
>
> [3] Wei, H., Tao, L., Xie, R., & An, B. (2021). Open-set label noise can improve robustness against inherent label noise. Advances in Neural Information Processing Systems, 34, 7978-7992.

---

> ### Author Response · Authors · 2023-11-22
> **reminder**
>
> We humbly remind the reviewer that we have uploaded responses to the reviewer's comments.
>
> Could the reviewer please provide feedback, as the discussion session ends in approximately 27 hours?
>
> We are eager to engage in further discussion with you!

---

> > ### Comment · Reviewer_4M7K · 2023-11-23
> >
> > I greatly appreciate your significant efforts in addressing my concerns.
> >
> > Upon reviewing the comments from fellow reviewers and the corresponding responses, I have decided to elevate my score to 6.

---

> ### Author Response · Authors · 2023-11-23
>
> We sincerely appreciate the reviewer's dedication in reviewing our responses and the comments from fellow reviewers. We are happy that our responses have addressed the concerns. Thank you!

---

### Official Review · Reviewer_KTbE · 2023-10-29

**Soundness:** 3 good
**Presentation:** 3 good
**Contribution:** 2 fair
**Rating:** 6
**Confidence:** 4

**Summary:**

This paper primarily explores the use of transition matrices in the context of noisy label learning. The authors conduct a meticulous analysis to expound upon the challenges that arise during the practical application of extant methods employing transition matrices. In response to these challenges, the authors introduce the Dirichlet-based per-sample Weight Sampling (DWS) framework. This framework effectively unifies two distinct methodologies, sample reweighting and resampling, facilitating their comparison within a comprehensive framework. Furthermore, the authors undertake an exhaustive analysis, establishing the resampling technique as supremely effective in addressing issues pertinent to noisy label learning.

**Strengths:**

1. The background and the motivation of the setting is well-introduced. The motivation of the work is reasonable.

2. The authors integrates sample reweighting and resampling methods into a single framework for comparative analysis, highlighting the superiority of the resampling approach.

3. The method is simple and the results on several datasets seem good.

**Weaknesses:**

1. I acknowledge the authors' theoretical contributions, but it must be said that the method proposed by the author lacks innovation. The approach in this paper bears similarity in concept to a significant category of noisy label learning methods based on sample selection, and their success strongly underscores the superiority of resampling methods.

2. On CIFAR-10N dataset, RENT performs well on rand1-3 and worse scenarios but exhibits poor performance on aggre. In the context of crowdsourcing, aggre should be a more common noise setting. Therefore, it is essential to analyze the reasons for the subpar performance in this particular scenario.

**Questions:**

Why use the Dirichlet distribution to sample the weight?

---

> ### Author Response · Authors · 2023-11-18
>
> **W1: method lacks innovation**
>
> -	We admit RENT of our study shares some similarities with sample-selection based methods as the reviewer pointed out. Specifically, both RENT and sample selection methods will filter out noisy labelled samples. Therefore, if we interpret sample selection methods as giving $p(Y=\tilde{y}|x)$ to noisy labelled sample (or, giving 0 to the numerator of $\frac{p(Y=\tilde{y}|x)}{p(\tilde{Y}=\tilde{y}|x)}$), RENT may also be interpreted as sample selection methods. In this way, we propose that RENT suggests a new direction of sample selection utilizing the transition matrix ensuring statistical consistency of the risk function. Techniques to improve selection quality in sample selection based methods would also be applied to RENT, e.g. using two same structure networks with different initialization points [1], adding confidence regularizer[2], etc.
>
> -	However, sample selection methods deterministically waste samples whose criteria value is below the threshold, while RENT would have possibility of sampling clean yet whose criteria is below the threshold. We admit this can also increase the possibility of sampling noisy data again, but we empirically showed good performance of RENT in the experiment that the reviewer HW3A suggested at Q2 as one case.
>
> -	More analysis including how many clean samples are below the threshold of each sample selection method and how much RENT can resamples those samples may be an interesting study, and we will consider it as prominent future work.
>
> -	Nevertheless, we want to underline our implemented method, RENT, is suggested based on the theoretical superiority of resampling over reweighting. If the reviewer knows any similar papers that theoretically proves the superiority of sample selection methods over other methods which shares the same form of objective function. please let us know so that we can refer to and upgrade our study.
>
> **W2: performance on Aggre noise**
>
> -	We basically think experimental results with Agree noise setting does not show statistically significant difference, and we conjecture that it is because the noise ratio of Aggregate is as small as 9.03% (from the original paper[1]).
>
> -	We experimented to see the performance difference of each transition matrix utilization method changing the noise ratio. Figure 11 in Appendix F.2 of the updated manuscript shows this relation. As we can see in the figure, the performance gap between red lines and others are subtle when the noise ratio is small. Having said that, the performance gap between FL and RENT on Aggre setting is not statistically significant.
>
> -	Still, RENT shows larger gap over two other methods under high noise ratio settings
>
> **Q1:Why Dirichlet distribution?**
> -	The motivation of introducing the concept of per-sample weight sampling is to integrate both reweighting and resampling. To explain both methods, we needed to express one-hot vector form (sampling) or generate the similar vectors to the mean vector  always (weighting). We thought Dirichlet distribution is a good choice since we can express both situations easily only by changing the shape parameter($\alpha$). With utilizing the Dirichlet distribution, we can construct a unique framework which connects the reweighting and resampling. Under this unified framework, we could directly compare reweighting and resampling strategies, and provide the rationale on why resampling generally performs better than reweighting for learning with noisy label.
>
> [1] Han, B., Yao, Q., Yu, X., Niu, G., Xu, M., Hu, W., ... & Sugiyama, M. (2018). Co-teaching: Robust training of deep neural networks with extremely noisy labels. Advances in neural information processing systems, 31.
>
> [2] Cheng, H., Zhu, Z., Li, X., Gong, Y., Sun, X., & Liu, Y. (2020, October). Learning with Instance-Dependent Label Noise: A Sample Sieve Approach. In International Conference on Learning Representations.

---

> > ### Comment · Reviewer_KTbE · 2023-11-22
> > **Response to Rebuttal**
> >
> > Thank you very much for your great efforts on addressing my concerns.
> >
> > After reading the comments from other reviewers and the responses, I have a better understanding of the contributions and the motivation behind the proposed methods. I decide to increase my score to 6.

---

> > > ### Author Response · Authors · 2023-11-22
> > >
> > > We thank the reviewer's efforts to check our responses and appreciate the opportunity to relieve the reviewer's concerns. Thank you!

---

### Official Review · Reviewer_HW3A · 2023-10-31

**Soundness:** 2 fair
**Presentation:** 3 good
**Contribution:** 2 fair
**Rating:** 6
**Confidence:** 3

**Summary:**

The paper proposes a resampling based training method for noisy label learning. The paper builds upon the resampling framework which has been proven to be superior to the reweighting method.  The proposed method utilizes Dirichlet distribution-based per-sample weight sampling to re-formalize reweighting and resampling, aiming to improve the utilization of the transition matrix. The experiments confirm the advantage of the resampling method.

**Strengths:**

The paper is well organized and clearly written.
The paper has a good angle that focuses on utilizing the transition matrix instead of estimating the transition matrix.
The paper formalizes the reweighting and resampling methods and provides a good analysis of both methods.
The paper provides comprehensive evaluation results regarding the reweighting and resampling methods.

**Weaknesses:**

1. The paper does not propose a "new" method. Reweighting and resampling are both existing methods, and may not be considered novel contributions.
2. Although the Dirichlet based analysis provides some good angles, the theoretical results do not have a clear insight. The distance from the true weight part simply says that the distance becomes smaller as $\alpha\rightarrow0$, but this does not tell us much because $\alpha\rightarrow0$ just gives a weight assignment closer to one-hot as demonstrated.
3. The paper lacks comparison with other means to deal with noisy label learning.
With these concerns, I have some doubt about the overall contribution of the paper.

**Questions:**

1. What can we get from the theoretical analysis other than resampling being better than reweighting when $\mu^*$ and $\mu$ are different, which is a conclusion from previous work (An et al. 2020)?
2. Following the last point, how different is the Dirichlet-based resampling from just setting a threshold for $\mu$ and randomly sampling?
3. Will the method work better if combined with data cleaning or abstention kind of ideas? There is one reference that I found ([1] Koziarski, Michał, Michał Woźniak, and Bartosz Krawczyk. "Combined cleaning and resampling algorithm for multi-class imbalanced data with label noise." Knowledge-Based Systems 204 (2020): 106223.). Although I'm not familiar with this work, I think it might be interesting to the authors and other reviewers.
4. Can the proposed method be compared with other noisy label learning methods (regularization, loss correction, or data cleaning)?

---

> ### Author Response · Authors · 2023-11-18
>
> **W1: novelty**
>
> -	If we misunderstood the reviewer’s intention, please let us know so that we can answer to the reviewer’s concerns and discuss those issues more deeply.
>
> -	We want to know whether there is any resampling-based study to solve noisy label classification task. We think our study is the first solution, but if the reviewer knows another, please let us know so that we can check similarities and differences. Under this situation, our study should also be the first to combine reweighing and resampling in one framework. We think it is meaningful because based on this unified framework, we can directly compare reweighting and resampling, and explain the reason why resampling can be better than reweighting for learning with noisy label.
>
> -	**[Comparison with importance sampling]** First, we want to underline considering importance sampling for noisy label task should be managed differently from the studies for different tasks. Considering what the traditional importance sampling method is, it assumes the source distribution (the distribution from which we draw samples, $p_{s}$) and the target distribution (the distribution for which you want to obtain information, $p_{t}$) is already given. For our problem setting, $p_{s}$ and $p_{t}$ would be the noisy label distribution and the clean label distribution, respectively. Since we know neither $p_{s}$ nor $p_{t}$ but have only $\tilde{D}$, the noisy labelled dataset, for learning with noisy label task, it makes the problem differently.
>
> -	We admit that there is already a study [1] that solves noisy label classification task with importance reweighting. [1] tried to solve the problem that we pointed out above by using the transition matrix. Since $p(\tilde{Y}|X)=Tp(Y|X)$ according to the definition of the transition matrix, the problem will be solved if we can estimate the transition matrix well and know only one of $p(\tilde{Y}|X)$ or $p(Y|X)$. From several studies [1,2], $p(Y|X)$ has been approximated from the on-training classifier’s output, as we also reported in Section 2.3 of the manuscript. Therefore, it is natural to assume that there will be approximation error for true weights from $\tilde{\mu}=\frac{f(x)}{Tf(x)}$. In section 3, we analyzed resampling can be more robust than reweighting when these weights are different from the true weights, which explains why RENT outperforms RW in most cases experimentally. We think this point is the main difference of our study from [1].
>
> -	**[Comparison with resampling/reweighting study]** Next, we admit that there are studies that compare and utilize resampling and reweighting, and [3] is one of those. [3] has studied resampling outperforms reweighting for correcting sampling bias with stochastic gradients and we are also motivated by them for hypothesizing that resampling may outperform reweighting even for learning with noisy label. However, their study and ours are different for the objective. The objective of resampling or reweighting in [3] is to find the optimal parameter which minimizes the risk over the whole population. If we translate this objective directly for learning with noisy label, the objective should be minimizing the risk over both clean label distribution and noisy label distribution. It is not the objective for noisy label learning, so [3] cannot be applied to learning with noisy label task. Under this situation, analyses in [3] should be modified to apply to noisy label learning.
>
> -	**[Comparison with sample selection methods]** Finally, we admit our study have some similarities with sample selection-based methods. Sample selection methods for learning with noisy labels selects subset of samples with some criteria and thresholds. RENT and sample selection methods share similar concept in that both use subset of samples for training. However, RENT has the possibility of sampling difficult clean samples, while sample selection based methods cannot since they deterministically divide between the selected samples and the wasted samples. Also the fact that we can assure the resampled dataset with RENT can be considered as the sampled dataset from clean label distribution theoretically is also another strength of RENT.

---

> ### Author Response · Authors · 2023-11-18
>
> **W2: theoretical insight**
>
> -	When $\alpha \rightarrow 0$, it means that the shape of the weight vector $\boldsymbol{w}$ becomes closer to one-hot as demonstrated in the main paper, and as the reviewer pointed out. It is straightforward this weight vector resembles dataset resampling. Therefore, the mahalanobis distance analysis in section 3.2 shows that the mean vector $\boldsymbol{\bar{w}}$ calculated from M times of $\boldsymbol{w}$ vector sampling is closer to $\mu^*$ (the true per-sample weight vector), even with same $\mu$. This means the empirical risk of RENT is more similar to the true risk than that of Reweighting.
>
> **W3: comparison with other baselines**
>
> -	Thanks for the reviewer’s suggestion. Below are the results. **Bold** means best, and *Italic* means second-best.
>
> **CIFAR10**
> | Terminology | Base | SN 20% | SN 50% | ASN 20% | ASN 40% |
> | --- | --- | --- | --- | --- | --- |
> | Regularization | ELR | 75.5$\pm$0.9 | 47.7$\pm$0.5 | 79.5$\pm$0.7 | 70.8$\pm$1.1 |
> | Regularization | SNL $(\sigma=0.1)$ | 71.7$\pm$0.3 | 46.9$\pm$0.3 | 80.5$\pm$1.1 |72.7$\pm$1.2 |
> | Robust loss | SCE | 79.5$\pm$0.6 | 54.8$\pm$0.5 | 79.5$\pm$0.8 | 69.5$\pm$0.9 |
> | Robust loss | APL | 79.3$\pm$1.2 | 61.5$\pm$3.0 | 76.9$\pm$1.7 | 64.1$\pm$1.0 |
> | Data cleaning | LRT | 74.9$\pm$0.5 | 46.5$\pm$1.2 | 77.7$\pm$1.9 | 69.2$\pm$0.6 |
> | Data cleaning | Coteaching | 78.7$\pm$1.4 | **76.4**$\pm$3.1 | *81.7*$\pm$0.5 |*73.9*$\pm$0.5 |
> | Data cleaning | Jocor | **83.4**$\pm$1.6 | 62.9$\pm$5.6 | 80.1$\pm$1.1 | 65.9$\pm$3.4 |
> | Data cleaning | DKNN | 55.5$\pm$1.0 | 30.8$\pm$0.8 | 62.3$\pm$1.1 | 54.1$\pm$0.7 |
> | Data cleaning | CORES$^2$ | 74.7$\pm$5.0 | 26.3$\pm$4.1 | 71.3$\pm$2.3 | 60.7$\pm$5.5 |
> | |**RENT** | *82.0*$\pm$0.2 | *74.6*$\pm$0.4 | **83.3**$\pm$0.1 | **80.0**$\pm$0.9 |
>
> **CIFAR100**
>
> | Terminology | Base | SN 20% | SN 50% | ASN 20% | ASN 40% |
> | --- | --- | --- | --- | --- | --- |
> | Regularization | ELR |  34.7$\pm$0.8 | 18.4$\pm$1.3 | 37.0$\pm$0.9 | *27.7*$\pm$1.2 |
> | Regularization | SNL $(\sigma=0.1)$  | 30.2$\pm$1.5 | 17.3$\pm$0.3 | 33.4$\pm$1.5 | 26.5$\pm$0.8 |
> | Robust loss | SCE | 34.3$\pm$1.2 | 18.3$\pm$0.6 | 36.2$\pm$1.1 | 27.6$\pm$0.5 |
> | Robust loss | APL | 33.5$\pm$2.0 | 18.3$\pm$5.5 | 35.9$\pm$2.0 | 24.0$\pm$4.1 |
> | Data cleaning | LRT |  33.8$\pm$1.8 | *20.1*$\pm$0.4 | 35.9$\pm$0.8 | 25.4$\pm$3.8 |
> | Data cleaning | Coteaching | *37.8*$\pm$4.0 | 12.5$\pm$1.3 | *39.7*$\pm$2.0 | 26.9$\pm$3.2 |
> | Data cleaning | Jocor | 27.5$\pm$4.9 | 7.9$\pm$1.4 | 34.7$\pm$2.5 | 26.9$\pm$2.3 |
> | Data cleaning | DKNN | 5.1$\pm$0.5 | 2.9$\pm$0.4 | 5.3$\pm$0.1 | 4.3$\pm$0.4 |
> | Data cleaning | CORES$^2$ | 37.8$\pm$2.3 | 6.5$\pm$2.1 | 37.8$\pm$1.7 | 27.2$\pm$1.3 |
> | |**RENT** | **39.8**$\pm$0.9 | **27.1**$\pm$1.9 | **39.8**$\pm$0.7 | **34.0**$\pm$0.4 |
>
> -	First, we have reported experimental results of other means for learning with noisy labels in the previous manuscript. We reported sample-selection based methods which include coteaching[4], jocor[5], DKNN[6] and CORES2[7]. Also, please note that we have compared RENT with SNL, which manages noisy labels as regularization method in section 4.4 and Appendix F.4 .
>
> -	We added more baselines to deal with noisy label learning following the reviewer’s suggestion. Baselines include ELR[8] as regularization, SCE[9] and APL[10] as robust loss, and LRT[11] as label correction. We did not consider SELF[12] or Dividemix[13] following [14].
>
> -	We mainly compared our method with Transition matrix based methods to demonstrate the improvement with regard to the transition matrix utilization, but we admit other baselines for the learning with noisy label task itself may be needed for a wider range of comparison.

---

> ### Author Response · Authors · 2023-11-18
>
> **Q1:** Please refer to the response of **W1** as **[Comparison with resampling/reweighting study]**
>
> **Q2:Dirichlet resampling v.s $w$ thresholding+random sampling**
>
> -	Thanks review for their interesting question. We think the reviewer already knows the difference between Dirichlet-based resampling from setting \mu threshold and random sampling for equation, so we directly report the experiment results.
> -	We think the motivation of this question is possibly the experiment result that we show at Section 4.5 ($w$ value), since $w$ values are quite divided according to the figure. Therefore, we set the threshold of $w$ as 1/(batch size) and randomly resampled dataset from whose $w$ value is larger than the threshold. We denote it as $w$+random in the table below.
>
> **CIFAR10**
> | **Base** | **Criterion** | **SN 20%** | **SN 50%** | **ASN 20%** | **ASN 40%** |
> | --- | --- | --- | --- | --- | --- |
> | Forward | $w+$random | 74.0$\pm$0.8 | 45.7$\pm$1.6 | 80.0$\pm$0.8 | 71.3$\pm$2.0 |
> | Forward | **w/ RENT** | **78.7**$\pm$0.3 | **69.0**$\pm$0.1 | **82.0**$\pm$0.5 | **77.8**$\pm$0.5 |
> | DualT | $w+$random | 74.8$\pm$0.6 | 48.2$\pm$1.2 | 80.1$\pm$0.6 | 72.4$\pm$0.9 |
> | DualT | **w/ RENT** | **82.0**$\pm$0.2 | **74.6**$\pm$0.4 | **83.3**$\pm$0.1 | **80.0**$\pm$0.9 |
> | TV | $w+$random | 70.7$\pm$1.8 | 45.5$\pm$1.6 | 78.7$\pm$0.9 | 71.9$\pm$1.7 |
> | TV | **w/ RENT** | **78.8**$\pm$0.8 | **62.5**$\pm$1.8 | **81.0**$\pm$0.4 | **74.0**$\pm$0.5 |
> | VolMinNet | $w+$random | 71.4$\pm$0.4 | 45.3$\pm$2.3 | 76.8$\pm$2.9 | 71.7$\pm$1.7 |
> | VolMinNet | **w/ RENT** | **79.4**$\pm$0.3 | **62.6**$\pm$1.3 | **80.8**$\pm$0.5 | **74.0**$\pm$0.4 |
> | Cycle | $w+$random | 70.8$\pm$12.2 | 44.7$\pm$0.8 | 78.1$\pm$0.7 | 69.2$\pm$1.2 |
> | Cycle | **w/ RENT** | **82.5**$\pm$0.2 | **70.4**$\pm$0.3 | **81.5**$\pm$0.1 | **70.2**$\pm$0.7 |
>
> **CIFAR100**
> | **Base** | **Criterion** | **SN 20%** | **SN 50%** | **ASN 20%** | **ASN 40%** |
> | --- | --- | --- | --- | --- | --- |
> | Forward | $w+$random | 30.8$\pm$1.8 | 16.1$\pm$1.1 | 30.6$\pm$2.2 | 24.6$\pm$0.5 |
> | Forward | **w/ RENT** | **38.9**$\pm$1.2 | **28.9**$\pm$1.1 | **38.4**$\pm$0.7 | **30.4**$\pm$0.3 |
> | DualT | $w+$random | 31.9$\pm$0.9 | 16.5$\pm$1.1 | 34.7$\pm$1.8 | 25.1$\pm$0.7 |
> | DualT | **w/ RENT** | **39.8**$\pm$0.9 | **27.1**$\pm$1.9 | **39.8**$\pm$0.7 | **34.0**$\pm$0.4 |
> | TV | $w+$random | 28.2$\pm$2.8 | 16.1$\pm$0.9 | 28.8$\pm$2.0 | 21.6$\pm$1.9 |
> | TV | **w/ RENT** |  **34.0**$\pm$0.9 | **20.0**$\pm$0.6 | **34.0**$\pm$0.2 | **25.5**$\pm$0.4 |
> | VolMinNet | $w+$random | 28.7$\pm$1.5 | 17.2$\pm$1.5 | 30.4$\pm$2.0 | 23.9$\pm$2.2 |
> | VolMinNet | **w/ RENT** | **35.8**$\pm$0.9 | **29.3**$\pm$0.5 | **36.1**$\pm$0.7 | **31.0**$\pm$0.8 |
> | Cycle | $w+$random | 31.5$\pm$0.8 | 16.3$\pm$1.4 | 33.2$\pm$1.6 | 24.5$\pm$2.3 |
> | Cycle | **w/ RENT** | **40.7**$\pm$0.4 | **32.4**$\pm$0.4 | **40.7**$\pm$0.7 | **32.2**$\pm$0.6 |
>
> -	RENT and $w$+random shows significant gap. We think this gap happens because during training, especially in the earlier learning iterations, w of clean samples and noisy samples may be more mixed (since the model parameter is yet more similar to the random assignment). Therefore, sorting with an arbitrary threshold may be riskier. However, Dirichlet-based resampling and RENT may be safer to this problem since the sampling probability of the samples below threshold is not 0.This would result in the overall performance differences.
>
> -	Also, choosing a good threshold may be difficult, and we admit that $w$+random performance may be improved with threshold tuning. However, some training iteration  and noise condition adaptive strategy could be needed. Figure 17 in the appendix F.5 in the revised manuscript shows this need. It shows the number of samples whose $w_i$ is larger than 1/B, with clean samples and noisy samples, respectively, while training the classifier with RENT. Result shows that clean samples whose w is larger than 1/B is not as many as the final iteration (which is natural), which underlines the training process adaptive strategy for threshold value.

---

> ### Author Response · Authors · 2023-11-18
>
> **Q3: Learning with noisy label with class imbalace**
>
> -	Thank the reviewer for suggesting the possible extension direction of our research.
>
> -	First, we need to clarify the terminology difference between their study and our study. In their study, resampling is used to mitigate the sampling bias for imbalanced data. Otherwise, we use resampling to change the distribution from noisy label distribution to true label distribution. In other words, resampling in class imbalanced learning study considers $p(y)$, while we consider $p(y|x)$. We previously discussed this issue more in **W1**.
>
> -	Considering the difference in terminology, what we are doing as resampling is more similar to data cleaning of their study.
>
> -	However, we admit solving class imbalanced dataset with noisy label can be important [16, 17]. Therefore, we experimented with synthetic label noise with class imbalance. Figure 20 in Appendix F.8 the revised manuscript shows the performance comparison of FL, FW and RENT with several T estimation baselines. As we can see in the figure, Again, RENT shows consistently good result over FL and RW. However, we admit there is no treatment to solve class imbalance issue in RENT, and this direction could be an interesting future study. We will report more experimental details in the appendix.
>
> **Q4:** We reported the model performance as above (**W3**)
>
> -	Can the reviewer explain what exactly the loss correction is? According to [15], it is the other expression of the loss modification with the transition matrix, and we have already reported several loss correction baselines on the paper.

---

> ### Author Response · Authors · 2023-11-18
>
> [1] Liu, T., & Tao, D. (2015). Classification with noisy labels by importance reweighting. IEEE Transactions on pattern analysis and machine intelligence, 38(3), 447-461.
>
> [2] Berthon, A., Han, B., Niu, G., Liu, T., & Sugiyama, M. (2021, July). Confidence scores make instance-dependent label-noise learning possible. In International conference on machine learning (pp. 825-836). PMLR.
>
> [3] An, J., Ying, L., & Zhu, Y. (2020, October). Why resampling outperforms reweighting for correcting sampling bias with stochastic gradients. In International Conference on Learning Representations.
>
> [4] Han, B., Yao, Q., Yu, X., Niu, G., Xu, M., Hu, W., ... & Sugiyama, M. (2018). Co-teaching: Robust training of deep neural networks with extremely noisy labels. Advances in neural information processing systems, 31.
>
> [5] Wei, H., Feng, L., Chen, X., & An, B. (2020). Combating noisy labels by agreement: A joint training method with co-regularization. In Proceedings of the IEEE/CVF conference on computer vision and pattern recognition (pp. 13726-13735).
>
> [6] Bahri, D., Jiang, H., & Gupta, M. (2020, November). Deep k-nn for noisy labels. In International Conference on Machine Learning (pp. 540-550). PMLR.
>
> [7] Cheng, H., Zhu, Z., Li, X., Gong, Y., Sun, X., & Liu, Y. (2020, October). Learning with Instance-Dependent Label Noise: A Sample Sieve Approach. In International Conference on Learning Representations.
>
> [8] Liu, S., Niles-Weed, J., Razavian, N., & Fernandez-Granda, C. (2020). Early-learning regularization prevents memorization of noisy labels. Advances in neural information processing systems, 33, 20331-20342.
>
> [9] Wang, Y., Ma, X., Chen, Z., Luo, Y., Yi, J., & Bailey, J. (2019). Symmetric cross entropy for robust learning with noisy labels. In Proceedings of the IEEE/CVF international conference on computer vision (pp. 322-330).
>
> [10] Ma, X., Huang, H., Wang, Y., Romano, S., Erfani, S., & Bailey, J. (2020, November). Normalized loss functions for deep learning with noisy labels. In International conference on machine learning (pp. 6543-6553). PMLR.
>
> [11] Zheng, S., Wu, P., Goswami, A., Goswami, M., Metaxas, D., & Chen, C. (2020, November). Error-bounded correction of noisy labels. In International Conference on Machine Learning (pp. 11447-11457). PMLR.
>
> [12] Nguyen, D. T., Mummadi, C. K., Ngo, T. P. N., Nguyen, T. H. P., Beggel, L., & Brox, T. (2019, September). SELF: Learning to Filter Noisy Labels with Self-Ensembling. In International Conference on Learning Representations.
>
> [13] Li, J., Socher, R., & Hoi, S. C. (2020). DivideMix: Learning with Noisy Labels as Semi-supervised Learning.
>
> [14] Yong, L. I. N., et al. "A Holistic View of Label Noise Transition Matrix in Deep Learning and Beyond." The Eleventh International Conference on Learning Representations. 2022.
>
> [15] Patrini, G., Rozza, A., Krishna Menon, A., Nock, R., & Qu, L. (2017). Making deep neural networks robust to label noise: A loss correction approach. In Proceedings of the IEEE conference on computer vision and pattern recognition (pp. 1944-1952).
>
> [16] Chen, B., Xia, S., Chen, Z., Wang, B., & Wang, G. (2021). RSMOTE: A self-adaptive robust SMOTE for imbalanced problems with label noise. Information Sciences, 553, 397-428.
>
> [17] Huang, Y., Bai, B., Zhao, S., Bai, K., & Wang, F. (2022, June). Uncertainty-aware learning against label noise on imbalanced datasets. In Proceedings of the AAAI Conference on Artificial Intelligence (Vol. 36, No. 6, pp. 6960-6969).

---

> ### Comment · Reviewer_HW3A · 2023-11-21
> **Response to rebuttal**
>
> Thank you for the detailed answers and clarifications. I do not mean to undermine the contribution by raising the question about the novelty, but wonder about the similarity with data selection and thresholding/random sampling. However, I think that after the additional results are added, the paper shows a thorough analysis of not only the proposed method but also the comparison between existing methods. Sorry about the confusing questions but I think you already answered them well. Therefore, I have changed my rating to 6.

---

> > ### Author Response · Authors · 2023-11-22
> >
> > We appreciate the opportunity to relieve the reviewer's concerns and provide further information, and we are happy our comments answered the reviewer's questions.
> > Thank you!

---

### Official Review · Reviewer_z7WX · 2023-11-01

**Soundness:** 3 good
**Presentation:** 2 fair
**Contribution:** 2 fair
**Rating:** 5
**Confidence:** 3

**Summary:**

This paper studies the problem of improving the uilization of transition matrix in label noise learning, authors propose that due to the poor estimation of class posterior, existing approaches such as loss correction or re-weighting might easily fail. To counter this issue, the authors proposed REsampling method to utilize the Noise Transition matrix (RENT), which utilizes Dirichlet distribution based resampling to assign instance-dependent weights.

**Strengths:**

1. The proposition of improving the utilization of transition matrix is interesting, and shows good insight - due to factors such as complexity, estimation error in noise class posterior and so on, even with a perfectly estimated transition matrix, it might still exhibts subpar performances.

2. The idea of re-sampling instead of re-weighting for loss correction is straight forward and intuitive.

**Weaknesses:**

**Major issues:**

1. Some experimental results are inconsistent with prior works, why are the performances of learning with true $T$ worse than Cycle consistency and Dual-$T$?

2. The discussions in section B seems problematic, in $\sum_{j=1}^{C}\(max_{j} T_{\hat{y}_{i}j} \)$, are we trying to sum over all $j$, or finding the maximum $j$? It seems that you're trying to do both, can you instead give an intuitive explaination and refine your mathmatical statements?

3. The motivation of DWS is not strong enough, if we are going with the assumption that the classifier trained with noisy labels can not accurately estimate noise class posterior (poorly calibrated, high estimation error, etc.), then we can hardly assume that the transition matrix is accurate, as the transition matrix is usually estimated from the noise class posterior.

**Minor issues:**

1. More recent and SOTA $T$ estimation method is not included [1].

[1] Yong, L. I. N., et al. "A Holistic View of Label Noise Transition Matrix in Deep Learning and Beyond." The Eleventh International Conference on Learning Representations. 2022.

**Questions:**

1. It is well-known that instance-dependent transition matrix might exhibit high complexity when the class number increases, for instance, for cifar-100, we need to compute a tensor of size $50000 \times 100 \times 100$, which is prohibited in real-world applications, can DWS mitigate this issue? More discussions in this aspect might bring more contribution.

---

> ### Author Response · Authors · 2023-11-18
>
> -	Considering W1, We think the reviewer asks two things in this question, and we respond to those one by one respectively.
>
> **W1: Experimental results inconsistency**
> - We have reported the reason of experimental results inconsistency with regard to the prior works in Appendix F.2 (Performance reproduce & comparison with our experiment setting). In Table 8 in the revised manuscript, [1] means the reported performance from their original paper, [2] means reproduced performance following each paper’s experimental condition and [3] is the model performance we reported in the main paper following our experimental settings. Below we report table 8 for the reviewer’s convenience.
>
> | T estimation | [1] ~ [3] | SN 20% | SN 50% | ASN 20% | ASN 40% |
> |--------------|-----------|--------|--------|---------|---------|
> | Forward| [1]  | 83.4$\pm$- | -      | 87.0$\pm$- | -      |
> | Forward| [2]  | 82.6$\pm$0.3 | 67.4$\pm$1.0 | 87.9$\pm$0.2 | 82.9$\pm$0.4 |
> | Forward |[3]  | 73.8$\pm$0.3 | 58.8$\pm$0.3 | 79.2$\pm$0.6 | 74.2$\pm$0.5 |
> | DualT| [1]    | 78.4$\pm$0.3 | 70.0$\pm$0.7 | -      | -      |
> | DualT |[2]    | 85.6$\pm$0.2 | 74.2$\pm$0.1 | 87.8$\pm$0.7 | 81.9$\pm$0.7 |
> | DualT| [3]    | 79.9$\pm$0.5 | 71.8$\pm$0.3 | 82.9$\pm$0.2 | 77.7$\pm$0.6 |
> | TV |[1]       | -            | 82.6$\pm$0.4 | -      | -      |
> | TV |[2]       | 87.5$\pm$0.2 | 76.6$\pm$0.2 | 80.6$\pm$7.4 | 75.0$\pm$13.3 |
> | TV |[3]       | 74.0$\pm$0.5 | 50.4$\pm$0.6 | 78.1$\pm$1.3 | 71.6$\pm$0.3 |
> | VolMinNet| [1]| 89.6$\pm$0.3 | 83.4$\pm$0.3 | -      | -      |
> | VolMinNet |[2]| 91.4$\pm$0.1 | 79.2$\pm$0.1 | 94.9$\pm$0.1 | 88.0$\pm$4.5 |
> | VolMinNet |[3]| 74.1$\pm$0.2 | 46.1$\pm$2.7 | 78.8$\pm$0.5 | 69.5$\pm$0.3 |
> | Cycle |[1]    | 90.4$\pm$0.2 | -            | 90.6$\pm$0.0 | 87.3$\pm$0.0 |
> | Cycle| [2]    | 90.4$\pm$0.4 | -            | 86.5$\pm$0.1 | 66.4$\pm$0.4 |
> | Cycle |[3]    | 81.6$\pm$0.5 | -            | 82.8$\pm$0.4 | 54.3$\pm$0.3 |
>
> -	One of the main differences is evaluation time. We chose to report the test accuracy after convergence following the below reasons.
>
> 1. Utilizing validation dataset costs more, and when collecting the dataset is expensive, it will be an important issue.
>
> 2. The quality of the validation dataset may not be trustful enough, and considering noisy label learning setting, the problem may be more important than the clean dataset setting.
>
> 3. Evaluation period term would impact choosing the best model, e.g. if model evaluation happens every 1,000 epochs, it will be no useful; if the model is evaluated on every step, it will cost too much time.
>
> 4. Evaluation during training cost more time.
>
> - Therefore, we suppose the model performance after finishing training is more reliable and realistic, so our reported performances are basically the performance after convergence.
>
> **W1: Performance of True T vs. Cycle, DualT**
>
> - Please check Appendix F.2 (True T is not the best? paragraph) on the revised manuscript for the related figures.
>
> - First, we experimented to analyze the correlation between the performances of RENT and the estimation gap of the resulting transition matrix with regard to several estimation algorithms. Figure 12 shows the results.
>
> - The model performance does not seem to have much correlation with T estimation error. We conjecture that this is due to two reasons: (1) since TV, VolMinNet and Cycle includes regularization terms for updating a classifier, those terms may have affected the classifier training process and (2) For TV, VolMinNet and Cycle, the transition matrix changes during training procedure, so the estimation gap would have reduced while training (according to their original paper, the estimation error seems to decrease with training process). Therefore, the transition matrix gap of those algorithms would have been larger.
>
> - To solely compare the impact of T estimation error to the model performance under same risk function and learning procedures, we generated error-injected transition matrix arbitrarily and reported the model performances as Figure 13 of the revised manuscript. Specifically, we subtracted $\epsilon$ to diagonal terms and added $\epsilon/$(the number of classes-1) to others.
>
> - Interestingly, we found out that the performance is not the best when the transition matrix estimation error is 0, showing the possible direction of improving RENT.
> - Please check Appendix F.2. for more details of the experiment.

---

> ### Author Response · Authors · 2023-11-18
>
> **W2: Notation error**
> - Sorry for our mistakes. The j inside the parenthesis should not be j (the same notation as the outside). We modified the notation in the revised manuscript.
>
> **W3: Noisy class posterior estimation**
>
> - First, we want to underline that the concept of “the estimation error of the noisy label posterior distribution from neural networks trained with noisy dataset could be high” is not new, and has already been studied in several previous studies [1,2,3]. Those studies have already suggested methods to estimate the transition matrix well under this situation, and we leveraged their methods for transition matrix estimation.
>
> -	We also reported one of these studies([2]) as Remark B.1 in Appendix C. Again, the difference between their study and ours, is that they focused on how to estimate good transition matrix when the noisy posterior distribution is inaccurate (since the transition matrix is usually estimated from the noise class posterior as the reviewer pointed out), while we studied how to utilize the transition matrix well.
>
> -	We admit we first assumed that the true T is accessible in section 2.3 to exclude the impact from the transition matrix estimation error and solely compare different T utilization. We demonstrated that current utilization can cause problem even with true transition matrix in Section 2.3, and this is the main motivation to find a new transition matrix utilization method. We are sorry if this causes any confusion for your idea.
>
> -	However, we do not consider that the transition matrix is accurate after Section 2.3. Rather, we implicitly include the case when the transition matrix is inaccurate. In Section 3, we focus on the situation when per-sample weights vector, $\boldsymbol{\mu}$ is different from $\boldsymbol{\mu}^*$. We think both the estimation gap of noisy class posterior and the estimation gap of transition matrix are encompassed in $\boldsymbol{\mu}$. While we admit that we did not directly analyze the impact of the transition matrix estimation error and the calibration error of the classifier yet, Eq. (5) implicitly addresses the potential impact of these errors. Since mu is composed of the transition matrix and the classifier output, the possible errors in transition matrix estimation and classifier calibration are inherent in our analysis. If the reviewer has some ideas on how to directly analyze the impact of these errors to the classifier, we are happy to further discuss this issue so that we can relieve any of your concerns.
>
> -	To represent the possible transition matrix estimation error from the previous transition matrix estimation methods, we reported experimental results with several transition matrix estimation baselines.
>
> **W4: SOTA baseline**
>
> -	Thank the reviewer for the suggestion. We report the experiment result as below, showing the experimental result over CIFAR10 and CIFAR100. We first checked whether the performance is reproducible, and we changed the Forward loss part as RENT and got the result. We used RCE as robust loss.
>
> | **Dataset**  | | **CIFAR10**   |  | | | **CIFAR100**   |  | | |
> |--------------  |--|------------|--|--|-|--------------|--|--|-|
> | **Noise**   |   | **UNI 20%**       | **UNI 50%**       | **FLIP 20%**      | **FLIP 45%**      |  **UNI 20%**       | **UNI 50%**       | **FLIP 20%**      | **FLIP 45%**      |
> | (ROBOT+RCE)    | w/FL (Reported)   | 92.1$\pm$0.0      | **88.5**$\pm$0.1  | 93.6$\pm$0.0      | 92.4$\pm$0.2      | 73.0$\pm$0.1      | 65.1$\pm$0.5      | 75.8$\pm$0.5      | 70.2$\pm$0.4      |
> |                | w/RW              | 88.9              | 84.5              | 72.7              | 86.9              | 6.3               | $-$               | 4.5               | 4.7               |
> |                | w/RENT            | **92.3**$\pm$0.3  | 87.5$\pm$0.4      | **93.9**$\pm$0.2  | **93.0**$\pm$0.2  | **74.1**$\pm$0.5  | **66.3**$\pm$0.4  | **76.9**$\pm$0.1  | **71.7**$\pm$1.2  |
>
> -	RENT again improves the model performances with ROBOT.

---

> ### Author Response · Authors · 2023-11-18
>
> **Q1: Intance dependent transition matrix estimation complexity**
>
> -	Sorry for our possible misunderstanding to the reviewer’s question, but let us respond to the reviewer’s question based on what we understood
>
> - Computation complexity issue: We want to clarify a tensor of size 50000 $\times$ 100 $\times$ 100 computation usually does not happen at one time for instance dependent transition matrix; we think the transition matrices would be calculated in batch-wise manner. For example, a baseline like [4] trains a new network that returns an instance dependent transition matrix as output. Therefore, it does not need to calculate the whole dataset at one time. If the reviewer knows some studies which compute the whole transition matrix in one time, please let us know so that we can understand how they could did that.
>
> - Estimation gap issue: Then, the problem of instance-dependent transition matrix is the difficulty of estimating the transition matrix, since a tensor of size 50000 $\times$ 100 $\times$ 100 should be estimated by using 50000 samples, for example, for cifar-100, as the reviewer pointed out. Please not that we focused on how to utilize the transition matrix well, and it is orthogonal to various T estimation methods. Therefore, we think the difficulty of estimating the transition matrix is beyond of our research scope. Nevertheless, it can be an interesting direction to study further as a future work.
>
> - Still, we want to underline again that RENT shows better performance than Forward loss (FL) even for instance dependent transition matrix based methods, as we reported in Table 2. (PDN[5] and BLTM[4]).
>
> [1] Yao, Y., Liu, T., Han, B., Gong, M., Deng, J., Niu, G., & Sugiyama, M. (2020). Dual t: Reducing estimation error for transition matrix in label-noise learning. Advances in neural information processing systems, 33, 7260-7271.
>
> [2] Zhang, Y., Niu, G., & Sugiyama, M. (2021, July). Learning noise transition matrix from only noisy labels via total variation regularization. In International Conference on Machine Learning (pp. 12501-12512). PMLR.
>
> [3] Li, S., Xia, X., Zhang, H., Zhan, Y., Ge, S., & Liu, T. (2022). Estimating noise transition matrix with label correlations for noisy multi-label learning. Advances in Neural Information Processing Systems, 35, 24184-24198.
>
> [4] Yang, S., Yang, E., Han, B., Liu, Y., Xu, M., Niu, G., & Liu, T. (2022, June). Estimating instance-dependent bayes-label transition matrix using a deep neural network. In International Conference on Machine Learning (pp. 25302-25312). PMLR.
>
> [5] Xia, X., Liu, T., Han, B., Wang, N., Gong, M., Liu, H., ... & Sugiyama, M. (2020). Part-dependent label noise: Towards instance-dependent label noise. Advances in Neural Information Processing Systems, 33, 7597-7610.

---

> ### Author Response · Authors · 2023-11-22
> **reminder**
>
> We humbly remind the reviewer that we have uploaded responses to the reviewer's comments.
>
> Could the reviewer please provide feedback, as the discussion session ends in approximately 27 hours?
>
> We are eager to engage in further discussion with you!

---

### Official Review · Reviewer_Epey · 2023-11-01

**Soundness:** 3 good
**Presentation:** 3 good
**Contribution:** 3 good
**Rating:** 6
**Confidence:** 3

**Summary:**

The paper first extends the reweighting strategy used in noisy-label learning, with a Dirichlet-based framework. This framework encompasses both reweighting and resampling as the two extremes of the Dirichlet distribution. The paper provides analysis of the impact of the shape parameter to the empirical risk and discuss resampling is better than reweighting. The paper finally proposes a method called RENT (resampling utilizing the noise transition matrix). Various experiments show the superiority and characteristics of the proposed method.

**Strengths:**

- The proposed algorithm is simple.
- Discussions about the relationship between related work is explained in detail.
- Experimental results are encouraging: we can see the benefit of introducing RENT.

**Weaknesses:**

- While reviewing the paper, I encountered some challenges in comprehending the content, primarily due to the clarity and organization of the presentation and missing notations/definitions. Some examples:
	- An explanation of $M$ seemed to be missing in page 4 but appears later on in page 6 (explained as a resampling budget.)
	- Would it be better to have Equation 3 in page 4 right after the definition of $R_{\ell, RW}$ in page 3?
	- I wasn't sure how $x_1, \ldots, x_M$ are determined in Algorithm 1 page 6. Since $\pi_N$ is a categorical distribution (and not a joint distribution of instances and labels), it seems to me that we can only sample $\tilde{y}_1, \ldots, \tilde{y}_M$ (without $x_1, \ldots x_M$)?
- For Proposition 3.1, it might be more clear to present the assumptions directly within the proposition rather than explaining them in the appendix.
- Is Eq. 7 correct? Should we introduce a different notation for the quantity in Eq.7 instead of writing that this is equivalent with $R_{l, DWS}^{emp}$, since we are applying the CLT in the proof in Appendix C.1 and $N$ is finite?

**Questions:**

I already wrote some of my questions in the previous section. Some other minor comments/questions:

- The 'et al.' in the references could be written out in full.

---

> ### Author Response · Authors · 2023-11-18
>
> **W1: Notations**
>
> - Sorry for our mistakes or the reviewer's confusion.
>
> - In page 4, M is the sampling number of $\boldsymbol{w}$. It can be translated directly to a resampling budget of RENT as in page 6. We revised the manuscript.
>
> - We modified our script with regard to the emprical risk of RW as the reviewer suggested (page 3)
>
> - $\pi_N$ is a categorical distribution to resample data instances. It is not for resampling label for a specific sample, and also not for resampling a specific input $(x_1,…,x_M)$. We resample data instances, which is a joint of $(x_i,\tilde{y}_i)$. We also reported some explanations with regard to the parameters of the categorical distribution in Appendix D.2. It may help you understanding the sampling process of RENT.
>
> **W2: Assumtion of proposition 3.1**
>
> -	Sorry for not directly presenting the assumption in Proposition 3.1 We changed it as in the revised manuscript (page 6, blue color)
>
> **W3: Correctness of Eq. 7**
>
> -	First, Eq 7 is not ours; it is the empirical risk function of SNL. We think what the reviewer asks is related to Eq.6, so we answer to this question based on Eq. 6.
> -	As the reviewer pointed out, we admit $R_{l,DWS}^{emp}$ can only be decomposed as the static risk and the stochastic normally distributed noise related term as Eq 6 when N goes to infinity. Therefore, we added the notation to show that N is infinite. We modified the manuscript in page 5, (Eq 6 part).
>
> **Q1: References**
>
> -	Sorry. We copied and pasted the bibtex-form codes by finding each reference, so we did not recognize there were et al. sign in the references. The et al. mark was included in total of four references, so we checked whether the authors of those references were included in each reference and removed the et al. signal.

---

> > ### Comment · Reviewer_Epey · 2023-11-22
> > **Comment**
> >
> > Thank you for updating the paper. I feel the clarity and organization have improved. Hence, I increased the score for "presentation" to "3 good". I plan to maintain my overall score since I already have a positive score.

---

> > > ### Author Response · Authors · 2023-11-22
> > >
> > > Thank the reviewer's efforts to check our responses and we appreciate the opportunity to relieve the reviewer's concerns. Thank you!

---

### Official Review · Reviewer_hEe7 · 2023-11-06

**Soundness:** 2 fair
**Presentation:** 2 fair
**Contribution:** 3 good
**Rating:** 6
**Confidence:** 3

**Summary:**

The paper addresses the crucial issue of learning from noisy labels, emphasizing the significance of the transition matrix in modeling the relationship between noisy and true labels. Rather than concentrating solely on learning this matrix, the authors propose a new approach that leverages it, drawing inspiration from reweighting and resampling concepts. They introduce a Dirichlet-based weighting method, which assigns individual weights to each sample drawn from a Dirichlet distribution. This distribution is parameterized using a base measure informed by the transition matrix. Empirical results indicate that this weighting method outperforms forward risk minimization and direct reweighing techniques.

**Strengths:**

Originality: The concept of imposing a Dirichlet prior on individual sample weights is well-founded. This approach naturally leverages the base measure to integrate the transition matrix, creating an informative prior. Additionally, the ability to control the concentration parameter $\alpha$ enables the fine-tuning of weight properties, including variance-based regularization and addressing noise-related issues. The practical implementation of this method is straightforward, as demonstrated in Algorithm 1.

Significance: The significance of this work is underscored by the comparison of the proposed reweighing method with forward risk minimization and an existing reweighting method based on the likelihood ratio. This comparative analysis is conducted across four datasets, and the results demonstrate the promise and potential impact of the proposed approach.

Clarity: Overall, the paper effectively conveys its central idea. However, there is room for improvement in terms of notation clarity.

**Weaknesses:**

The comparative analysis of RENT is limited to FL and RW, utilizing transition matrices obtained through various methods. A more comprehensive evaluation, including a broader range of noise-label learning techniques, such as the approach proposed by Lin et al. in 2022, would provide a more comprehensive assessment of RENT's performance.

**Questions:**

* In table 2, there are a couple of settings, where RENT does not perform as well as the other two methods, can the authors discuss the underlying reasons?
* The author said that RENT performs better when estimated T differs from True T, the reviewer wondered if one can plot the performance gap against the differences between the estimated T and True T.

---

> ### Author Response · Authors · 2023-11-18
>
> **W1: Comparison with Lin et al. 2022**
>
> -	Thank you for the reviewer’s suggestion. We reproduce and report the experiment result of Lin et al., which is VRNL (the following table). Since it can be applied to various transition matrix estimation methods orthogonally, we report the experimental results as we did in Table 1 on the main paper. We provide additional details of the experiment at Appendix F.2
> | Base   | Risk        | SN 20%         | SN 50%         | ASN 20%        | ASN 40%        | SN 20%         | SN 50%         | ASN 20%        | ASN 40%        |
> |--------|-------------|----------------|----------------|----------------|----------------|----------------|----------------|----------------|----------------|
> | Forward| w/ FL       | 73.8$\pm$0.3   | 58.8$\pm$0.3   | 79.2$\pm$0.6   | 74.2$\pm$0.5   | 30.7$\pm$2.8   | 15.5$\pm$0.4   | 34.2$\pm$1.2   | 25.8$\pm$1.4   |
> | Forward| w/ VRNL     | 76.9$\pm$0.4   | 64.8$\pm$2.3   | 54.2$\pm$9.2   | 59.3$\pm$12.7  | 34.1$\pm$3.4   | 18.4$\pm$2.8   | 35.5$\pm$2.4   | 27.2$\pm$1.6   |
> | Forward| w/ RENT     | **78.7$\pm$0.3** | **69.0$\pm$0.1** | **82.0$\pm$0.5** | **77.8$\pm$0.5** | **38.9$\pm$1.2** | **28.9$\pm$1.1** | **38.4$\pm$0.7** | **30.4$\pm$0.3** |
> | DualT  | w/ FL       | 79.9$\pm$0.5   | 71.8$\pm$0.3   | 82.9$\pm$0.2   | 77.7$\pm$0.6   | 35.2$\pm$0.4   | 23.4$\pm$1.0   | 38.3$\pm$0.4   | 28.4$\pm$2.6   |
> | DualT  | w/ VRNL     | 81.2$\pm$0.3   | 73.7$\pm$0.8   | **83.5$\pm$0.1** | 78.1$\pm$2.2   | 38.2$\pm$1.4   | 25.2$\pm$3.7   | **40.0$\pm$1.2** | **34.4$\pm$1.3** |
> | DualT  | w/ RENT     | **82.0$\pm$0.2** | **74.6$\pm$0.4** | 83.3$\pm$0.1   | **80.0$\pm$0.9** | **39.8$\pm$0.9** | **27.1$\pm$1.9** | 39.8$\pm$0.7   | 34.0$\pm$0.4   |
> | TV         | w/ FL       | 74.0$\pm$0.5   | 50.4$\pm$0.6   | 78.1$\pm$1.3   | 71.6$\pm$0.3   | **34.5$\pm$1.4** | 21.0$\pm$1.4   | 33.9$\pm$3.6   | **28.7$\pm$0.8** |
> | TV         | w/ VRNL     | 76.0$\pm$0.6   | 51.3$\pm$0.7   | 78.8$\pm$0.3   | 58.5$\pm$9.2   | 28.5$\pm$1.2   | **22.9$\pm$2.8** | 29.9$\pm$1.0   | 26.4$\pm$2.3   |
> | TV         | w/ RENT     | **78.8$\pm$0.8** | **62.5$\pm$1.8** | **81.0$\pm$0.4** | **74.0$\pm$0.5** | 34.0$\pm$0.9 | 20.0$\pm$0.6   | **34.0$\pm$0.2** | 25.5$\pm$0.4   |
> | VolMinNet  | w/ FL       | 74.1$\pm$0.2   | 46.1$\pm$2.7   | 78.8$\pm$0.5   | 69.5$\pm$0.3   | 29.1$\pm$1.5   | 25.4$\pm$0.8   | 22.6$\pm$1.3   | 14.0$\pm$0.9   |
> | VolMinNet  | w/ VRNL     | 76.3$\pm$0.9   | 50.3$\pm$1.4   | 72.3$\pm$9.0   | 67.4$\pm$5.3   | 28.1$\pm$0.6   | 26.6$\pm$2.2   | 19.5$\pm$3.7   | 14.7$\pm$1.4   |
> | VolMinNet  | w/ RENT     | **79.4$\pm$0.3** | **62.6$\pm$1.3** | **80.8$\pm$0.5** | **74.0$\pm$0.4** | **35.8$\pm$0.9** | **29.3$\pm$0.5** | **36.1$\pm$0.7** | **31.0$\pm$0.8** |
> | Cycle      | w/ FL       | 81.6$\pm$0.5   | $-$            | 82.8$\pm$0.4   | 54.3$\pm$0.3   | 39.9$\pm$2.8   | $-$            | 39.4$\pm$0.2   | 31.3$\pm$1.2   |
> | Cycle      | w/ VRNL     | 82.4$\pm$0.6   | $-$            | **83.0$\pm$0.4** | 54.3$\pm$0.4   | **41.9$\pm$2.1** | $-$            | **42.1$\pm$1.6** | **32.5$\pm$1.2** |
> | Cycle      | w/ RENT     | **82.5$\pm$0.2** | **70.4$\pm$0.3** | 81.5$\pm$0.1   | **70.2$\pm$0.7** | 40.7$\pm$0.4 | **32.4$\pm$0.4** | 40.7$\pm$0.7 | 32.2$\pm$0.6 |
> | True $T$   | w/ FL       | 76.7$\pm$0.2   | 57.4$\pm$1.3   | 75.0$\pm$11.9  | 70.7$\pm$8.6   | 34.3$\pm$0.5   | 22.0$\pm$1.5   | **35.8$\pm$0.5** | **31.9$\pm$1.0** |
> | True $T$   | w/ VRNL     | 79.3$\pm$0.6   | 63.8$\pm$1.3   | 49.2$\pm$4.4   | 47.7$\pm$6.2   | **36.6$\pm$1.1** | **25.5$\pm$0.7** | 26.2$\pm$3.3   | 22.5$\pm$5.7   |
> | True $T$   | w/ RENT     | **79.8$\pm$0.2** | **66.8$\pm$0.6** | **82.4$\pm$0.4** | **78.4$\pm$0.3** | 36.1$\pm$1.1 | 24.0$\pm$0.3   | 34.4$\pm$0.9 | 27.2$\pm$0.6   |
>
> - Since there is no official code, we reproduced the code based on the paper explanation.
>
> - VRNL improves the original $T$ estimation methods, and RENT tends to show better performances than VRNL on CIFAR10, and similar performances on CIFAR 100 (15 win, 9 lose cases over 24 settings). We suppose the estimation gap of per-sample weights may be large with regard to CIFAR100, so it caused lower performance in some cases.
>
> - By the way, we think our method can also be integrated with VRNL. Specifically, this integration can be defined as utilizing the transition matrix as RENT (risk function part) and add variance of the risk as VRNL does (regularization). We may show the experiment results of such combination by the end of the rebuttal.

---

> ### Author Response · Authors · 2023-11-18
>
> **Q1: Performance in Table 2**
>
> -	First note that the noise ratio of Aggregate is as small as 9.03% (from the original paper[1]).
>
> -	We reported the performance comparison of each transition matrix utilization method changing the noise ratio. Figure 11 in Appendix F.2 of the updated manuscript shows this relation. As we can see in the figure, the performance gap between red lines and others are subtle when the noise ratio is small. Having said that, the performance gap between FL and RENT on Aggre setting is not statistically significant.
>
> -	Still, RENT shows larger gap over two other methods under high noise ratio settings.
>
> **Q2: Performance against the T estimation gap**
>
> -	We interpreted what the reviewer said is related to the result of table 1, however, if our understanding for the reviewer’s suggestion is wrong, please let us know.
>
> -	Thanks for the reviewer’s suggestion, and please check Appendix F.2 (True T is not the best? paragraph) on the revised manuscript for the related figures.
>
> -	First, we reported the performances of RENT with several T estimation methods over the differences between the resulting estimated T and true T. Figure 12 shows the results.
>
> -	At first glance, the model performance does not seem to have much correlation with T estimation error. We conjecture that this is due to two reasons: (1) since TV, VolMinNet and Cycle includes regularization terms for updating a classifier, those terms may have affected the classifier training process and (2) For TV, VolMinNet and Cycle, the transition matrix changes during training procedure, so the transition matrix estimation gap would have reduced while training (according to their original paper, the estimation error seems to decrease with training process). Therefore, the transition matrix gap of those algorithms would have been larger than the reported transition matrix estimation gap.
>
> -	We thought we need to compare performances under same risk function (including any regularization terms) and other learning procedures to analyze the impact of T estimation error to the model performance. Therefore, we generated error-injected transition matrix arbitrarily and reported the model performances as Figure 13 in Appendix F.2 of the revised manuscript. For making arbitrary wrong transition matrix, we subtracted $\epsilon$ to diagonal terms and added $\epsilon/$(the number of classes-1) to others.
>
> -	Interestingly, we found out that the performance is not the best when the transition matrix estimation error is 0, showing the possible direction of improving RENT.
>
> [1] Wei, J., Zhu, Z., Cheng, H., Liu, T., Niu, G., & Liu, Y. (2021). Learning with noisy labels revisited: A study using real-world human annotations. arXiv preprint arXiv:2110.12088.

---

> ### Author Response · Authors · 2023-11-21
>
> We uploaded the experimental results of the integration between VRNL and RENT in the revised manuscript, as we promised previously.
>
> Please take a look.
>
> Thanks!

---

> ### Author Response · Authors · 2023-11-22
> **reminder**
>
> We humbly remind the reviewer that we have uploaded responses to the reviewer's comments.
>
> Could the reviewer please provide feedback, as the discussion session ends in approximately 27 hours?
>
> We are eager to engage in further discussion with you!

---

### Author Response · Authors · 2023-11-18

Thanks for the reviewers for their sincere works. We provide responses to each reviewer's concern and update the manuscript to reflect the reviewer's comments. We set the changed part of the script as blue.  Please check our response and feel free to comment.

---

### Author Response · Authors · 2023-11-21

Dear Reviewers:

We updated the manuscript to add one more experiment in the appendix F.2.
To avoid confusion, this revision is reflected as violet color.

Best Regards,

---

### Meta-Review · Area_Chair_nnXs · 2023-12-09

**Metareview:**

In this paper, the authors propose to utilize a transition matrix that accounts for errors in the labeling to improve classification accuracy. They put forward a theoretical framework and an algorithm to improve the classification accuracy when considering the transition matrix. During the discussion phase, the reviewers and authors engaged, and many of the reviewers were satisfied with the answer and increased their scores.

The paper does not consider the problem of estimating the transition matrix, which is a minor limitation, as they indicate that other algorithms exist for it. The proposed algorithm minimally improves other existing algorithms.

**Justification For Why Not Higher Score:**

The paper is a fine contribution to a niche problem. It does not deserve to be highlighted.

**Justification For Why Not Lower Score:**

The paper could be rejected. The advantages of estimating and using the transition matrix are minor and unless the number of errors in labeling is huge these methods provide little gains. It is a niche problem with small practical application in today's machine learning and it could be dropped from the conference. The meta review can be adapted if the paper is rejected.

---

### Decision · Program_Chairs · 2024-01-16

Accept (poster)